# Efficient Adaptive Federated Optimization

## Abstract

Adaptive optimization plays a pivotal role in federated learning, where simultaneous server and client-side adaptivity have been shown to be essential for maximizing its performance. However, the scalability of jointly adaptive systems is often constrained by limited resources in communication and memory. In this paper, we introduce a class of efficient adaptive algorithms, named $\texttt{FedAda}^2$, designed specifically for large-scale, cross-device federated environments. $\texttt{FedAda}^2$ optimizes communication efficiency by avoiding the transfer of preconditioners between the server and clients. At the same time, it leverages memory-efficient adaptive optimizers on the client-side to reduce on-device memory consumption. Theoretically, we demonstrate that $\texttt{FedAda}^2$ achieves the same convergence rates for general, non-convex objectives as its more resource-intensive counterparts that directly integrate joint adaptivity. Empirically, we showcase the benefits of joint adaptivity and the effectiveness of $\texttt{FedAda}^2$ on both image and text datasets.

## 1 Introduction

Federated learning is a distributed learning paradigm which aims to train statistical models across multiple clients while minimizing raw data exposure (McMahan et al., 2017; Li et al., 2020a; Wang et al., 2021a). In vanilla federated learning, a central server orchestrates the training process by distributing the global model to a subsample of thousands or even millions of clients. These clients collaboratively perform local stochastic gradient descent while drawing from their private data streams. After several epochs have elapsed, each client communicates their aggregate updates to the server, which averages this information to make an informed adjustment to the global model. This algorithm, using non-adaptive weight updates, is called *FedAvg* (McMahan et al., 2017). A recent trend is to investigate utilizing adaptive optimizers to support federated learning (Reddi et al., 2021). Adaptivity can be employed in either the server-side or the client-side, where joint adaptivity (consisting of global *and* local adaptive updates) has been shown to play a pivotal role in accelerating convergence and enhancing accuracy (Wang et al., 2021b).

Nevertheless, efficiency challenges remain for the successful deployment of jointly adaptive algorithms in practice, especially in cross-device federated settings (Kairouz et al., 2021). The server, which collects pseudogradients pushed by participating clients, consolidates a global approximation of the preconditioners for adaptive model updates. Typically, the server sends the preconditioners back to the clients to precondition local adaptive updates. However, this can lead to significant communication overhead that detracts from the advantages offered by adaptivity (Wang et al., 2022). Furthermore, dynamically varying client resource limitations restrict the reliability of client-side adaptive optimizers in practice, especially when additional memory is required for handling local preconditioners during each client model update.

In this work, we propose a class of efficient jointly adaptive distributed training algorithms, called $\texttt{FedAda}^2$, to mitigate the aforementioned communication and memory restrictions while retaining the benefits of adaptivity. $\texttt{FedAda}^2$ maintains an identical communication complexity as the vanilla FedAvg algorithm. Instead of transmitting global server-side preconditioners from the server to the selected clients, we propose the simple strategy of allowing each client to initialize local preconditioners from constants (such as zero), without any extra communication of preconditioners. In addition, when running local updates, we adopt existing memory-efficient optimizers that factorize the gradient statistics to reduced dimensions to save on-device memory. We prove that for the general, non-convex setting, $\texttt{FedAda}^2$ achieves the same convergence rate as prior adaptive federated optimizers (e.g., Reddi et al. (2021)). In this paper, we demonstrate that jointly adaptive federated

learning, as well as adaptive client-side optimization, are practicable in real-world settings while sidestepping localized memory restrictions and communication bottlenecks.

**Contributions.** Our contributions are summarized as follows.

- Motivated by the importance of joint server- and client-side adaptivity both empirically and theoretically, we propose a framework `FedAda`$^2$ to avoid extra communication cost and reduce on-device memory while retaining the benefits of joint adaptive optimization (Section 4).
- We provide convergence analyses for a class of `FedAda`$^2$ algorithms instantiated with different server- and client-side adaptive methods and memory-efficient local optimizers (Section 5). To the very best of our knowledge, there are no known convergence results on joint adaptive federated optimization in the general convex or non-convex settings.
- Empirically, we show that `FedAda`$^2$, without transmitting preconditioners and employing on-device preconditioner compression, matches the performance of its more expensive counterparts, and outperforms baselines without joint adaptivity on both image and text datasets (Section 6).

## 2 RELATED WORK

We now provide a brief overview of related work in adaptive federated learning and memory-efficient[1] preconditioning.

**Adaptive Federated Optimization.** Adaptive optimization preconditions the gradients to enhance optimization efficacy, dynamically adjusting the learning rate for each model parameter (e.g., Duchi et al., 2011; Kingma & Ba, 2015; Reddi et al., 2018). Recent developments in federated learning have leveraged adaptive methods for server and client model parameter updates. Frameworks such as FedAdam (Reddi et al., 2021) and FederatedAGM (Tong et al., 2020) focus primarily on server-side adaptivity while using a constant learning rate for client updates. Additionally, FedCAMS (Wang et al., 2022) delves into communication-efficient adaptive optimization by implementing error feedback compression to manage client updates while maintaining adaptivity solely on the server side. Conversely, methodologies such as FedLALR (Sun et al., 2023), Local AdaAlter (Xie et al., 2019), and Local AMSGrad (Chen et al., 2020) have adopted client-side adaptivity exclusively. These approaches involve transmitting both client preconditioners and model parameters for global aggregation in the server. Moreover, some frameworks have embraced joint adaptivity. Local Adaptive FedOPT (Wang et al., 2021b) implements joint adaptivity while incorporating an additional client correction term. These terms, along with transmitted client pseudogradients, are aggregated on the server to construct a global preconditioner used to synthesize the subsequent model update. Alternatively, frameworks such as MIME (Karimireddy et al., 2021; Jin et al., 2022) transmit additional optimizer state information aggregated in the server to mimic adaptive updates in centralized settings, while maintaining frozen-state optimizers on the client-side. In contrast with all these approaches, `FedAda`$^2$ avoids the transmission of any local/global preconditioners and optimizer states entirely, maintaining precisely identical communication complexity as vanilla FedAvg despite leveraging joint adaptivity. We include further discussions in Appendix G.5.

**Memory-Efficient Adaptive Optimizers.** The implementation of local adaptive methods substantially increases client memory requirements, as it necessitates the maintenance of local preconditioners. For some models, it has been noted that the gradients combined with optimizer states consume significantly more memory than the actual model parameters themselves (Raffel et al., 2020). Memory-efficient adaptive optimizers have been extensively studied in prior literature. Algorithms such as Adafactor (Shazeer & Stern, 2018) address memory reduction by tracking moving averages of the reduction sums of squared gradients along a singular tensor axis, attaining a low-rank projection of the exponentially smoothed preconditioners. GaLore (Zhao et al., 2024) targets the low-rank assumption of the gradient tensor, which reduces memory of both gradients and preconditioners. Shampoo (Gupta et al., 2018) collapses gradient statistics into separate preconditioning matrices for

---

[1]There are various notions of 'efficiency' of adaptive methods in the context of the federated learning, two of them being communication efficiency and client memory efficiency. Our contribution specifically targets reducing communication and memory costs incurred by *local preconditioners*, which is complementary with works that reduce communication by repeated local updates or model weight/pseudogradient compression (e.g., FedCAMS (Wang et al., 2022)) and may, in theory, even be combined.

each tensor dimension, which is extended via extreme tensoring (Chen et al., 2019). In this paper, we focus on SM3 (Anil et al., 2019) in our implementation and experiments due to its empirical performance; however, our theoretical framework covers a broad class of memory-efficient optimizers applied on the client-side (Section 5 and Appendix D).

# 3 IMPORTANCE OF CLIENT-SIDE ADAPTIVITY

In this section, we motivate our work by providing a theoretical description of how leveraging client-side adaptivity improves distributed learning, which is later validated in experiments (Section 6). Our analyses are motivated by prior works that uncover critical conditions under which centralized SGD can diverge, specifically in settings involving heavy-tailed gradient noise (Zhang et al., 2020). After analyzing the importance of client-side adaptivity, we propose efficient FL frameworks to mitigate the heightened resources induced by adaptive local optimizers in Section 4, which is `FedAda`[2]. We begin by providing a definition of heavy-tailed noise following previous literature.

**Definition 1.** *A random variable $\xi \sim \mathcal{D}$ follows a **heavy-tailed** distribution if the $\alpha$-moment is infinite for $\alpha \geq 2$. In other words, we say that the stochastic gradient noise $g(x) - \nabla f(x)$ is heavy-tailed if $\mathbb{E}\left[\|g(x) - \nabla f(x)\|^{\alpha}\right]$ is bounded for $\alpha \in (0, 2)$ and unbounded for $\alpha \geq 2$, where $g(x)$ is the stochastic gradient under some model parameter $x$, and $\nabla f(x)$ the full gradient.*

We may now present the following proposition.

**Proposition 2.** *There exists a federated learning problem with heavy-tailed client-side gradient noise such that the following arguments hold:*

*(i) For vanilla FedAvg, given any client sampling strategy, if the probability $p_i^t$ of client $i$ with heavy-tailed gradient noise being sampled at communication round $t$ is non-zero, then $\mathbb{E}\|\nabla f(x_{t+1})\|^2 = \infty$ for any nontrivial learning rate schedule $\eta_\ell^t > 0$ and global parameter $x_{t+1}$.*

*(ii) Under an appropriate learning rate schedule, FedAvg with local adaptivity (i.e., via client-side AdaGrad) bounds the error in expectation as*

$$\lim_{t \to \infty} \mathbb{E}\|x_t - x^*\| \leq \frac{2\sqrt{3}}{1 - \hat{\varepsilon}} \quad \text{for some} \quad \hat{\varepsilon} \approx 0,$$

*where $x^*$ is the global optimum.*

A detailed proof is given by construction on a quadratic objective in Appendix A. We show that even a single client with heavy-tailed gradient noise is able to instantaneously propagate their volatility to the global model, which severely destabilizes distributed learning in expectation. Unfortunately, recent works have observed heavy-tailed gradient noise empirically, especially within model architectures utilizing attention mechanisms, including transformer-based models (Zhang et al., 2020; Devlin et al., 2018; Brown et al., 2020; Dosovitskiy et al., 2021; Nguyen et al., 2019; Simsekli et al., 2019; 2020). Proposition 2 (ii) suggests that client-side adaptivity has the potential to stabilize local model updates pushed from diverse and large-scale distributed sources, if communication bottlenecks and memory efficiency can be addressed.

The construction of the federated problem in Proposition 2 draws gradient noise from the Student $t$-distribution which is heavy-tailed depending on the parameter regime, whose moments are relatively controlled nevertheless. We may exacerbate the severity of gradient stochasticity by inserting a singular client with Cauchy-distributed noise, while enforcing all other clients to follow non-heavy-tailed Gaussian gradient noise. We further detail this setting in Proposition 10, Appendix A.

## 3.1 DEEP REMORSE OF FEDAVG AND SGD

So far, we have examined toy problems in which heavy-tailed gradient noise is guaranteed to destabilize distributed training in expectation. We now prove that this is an instantiation of a more general phenomenon in federated learning where a family of online $\mu$-strongly convex global objectives collapses to the identical failure mode. To our knowledge, this provable limitation of distributed training resultant from the heavy-tailed noise of a singular client has not previously been established within the literature. The proofs of all results are given in the appendix.

**Definition 3.** *A learning algorithm $\mathcal{A}$ is **deeply remorseful** if it incurs infinite or undefined regret in expectation. If $\mathcal{A}$ is guaranteed to instantly incur such regret due to sampling even a single client with a heavy-tailed gradient noise distribution, then we say $\mathcal{A}$ is **resentful** of heavy-tailed noise.*

**Theorem 4.** *Let the global objectives $f_t(x)$ of a distributed training problem satisfy $\mu$-strong convexity for $t = 1, \ldots, T$. Assume that the participation probability of a client with a heavy-tailed stochastic gradient noise distribution is non-zero. Then, FedAvg becomes a deeply remorseful algorithm and is resentful of heavy-tailed noise. Furthermore, if the probability of the heavy-tailed client being sampled at step $t$ is nontrivial, then the variance of the global objective at $t + 1$ satisfies $\mathbb{E}\|f_{t+1}(x_{t+1})\|^2 = \infty$.*

In federated learning, we typically have $f_t(x) \equiv f(x)$ for all $t = 1, \ldots, T$ (i.e., the objective functions are the same across all rounds). Proposition 2 intuits that inserting local adaptivity successfully breaks the generality of remorse and heavy-tailed resent for FedAvg. A high-level overview is that client-side AdaGrad clips the local updates of each iteration, which mollifies the impact of stochasticity in perturbing the weight updates. This gives Proposition 5, which is formulated loosely without utilizing any advantages provided by local adaptivity except for clipping. Given that adaptive methods inherently include an implicit soft clipping mechanism due to the effects of preconditioning, we consider them to be preferable to clipped SGD for large-scale applications as they also offer the benefits of adaptivity. This preference holds, provided that the memory and computational constraints of the clients can be adequately managed.

**Proposition 5.** *Introducing client-side adaptivity via AdaGrad for the setting in Theorem 4 produces a non-remorseful and a non-resentful algorithm.*

The benefits of client-side adaptivity have also been shown in previous works (e.g., Zhou et al. (2024); Wang et al. (2021b)). We note that Proposition 5 can be straightforwardly extended to jointly adaptive methods as well as for $f_t \in C(\mathbb{R}^d)$ not necessarily convex. An advantage of federated learning is that when done tactfully, the large supply of clients enable the trainer to draw from a virtually unlimited stream of computational power. The downside is that the global model may be strongly influenced by the various gradient distributions induced by the private client data shards. In this paper, we focus specifically on joint adaptive optimization as a countermeasure to stabilize learning. In Section 4, we propose $\texttt{FedAda}^2$, which utilizes joint adaptivity in an efficient and scalable manner for distributed or federated training.

# 4 FEDADA$^2$: EFFICIENT JOINT SERVER- AND CLIENT-SIDE ADAPTIVITY

In federated learning, a typical server-side objective is formed by taking an average of all client objectives $F_i(x)$ for $i \in [N]$ and $x \in \mathbb{R}^d$:

$$f(x) = \frac{1}{N} \sum_{i=1}^{N} F_i(x). \tag{1}$$

In the case of unbalanced client data sizes or sampling probabilities, the objective becomes $\sum_{i=1}^{N} p_i F_i(x)$ on the right hand side where $p_i$ is proportional to the local data size of client $i$, or the sampling probability. With a slight abuse of notation, we denote $F_i(x) = \mathbb{E}_{z \sim \mathcal{D}_i}[F_i(x, z)]$ where $F_i(x, z)$ is the stochastically realized local objective and $\mathcal{D}_i$ is the data distribution of client $i$. The convergence analysis developed in Section 5 holds when $\mathcal{D}_i$ is taken to be the local population distribution, as well as when $\mathcal{D}_i$ is the local empirical distribution. For analytical purposes, we assume that the global objective does not diverge to negative infinity and admits a minimzer $x^*$.

One determining property of cross-device federated settings is that the clients are not able to store or maintain 'states' across communication rounds (Kairouz et al., 2021). To realize joint adaptivity in federated systems in a stateless way, one natural baseline is to estimate (pseudo)gradient statistics on the server (i.e., maintaining server-side preconditioners or global preconditioners) and transmit them to all participating clients at every communication round. Then each selected client performs local adaptive steps with preconditioners starting from the global ones. This approach enables clients to utilize global preconditioner information to make informed adjustments to their respective local models. However, transmitting (pseudo)gradient statistics, such as the second moment, at each round significantly increases the communication cost. In addition, running adaptive updates locally based on the local data introduces memory overheads. Next, we discuss two main techniques we use for efficient federated adaptive optimization with convergence guarantees.

**Zero Local Preconditioner Initialization.** To enhance the feasibility of jointly adaptive federated learning in cross-device settings, we first address extra major communication bottlenecks brought by transmitting global preconditioners from the server to a subset of clients. We propose a simple strategy of uniformly initializing local preconditioners to zero (or some constant vector) at the beginning of each training round, thus eliminating the need for preconditioner transmission.

To describe the process in more detail, assume Adagrad (with momentum) as the server-side optimizer (Reddi et al., 2021) for illustration purposes. We have the following server update rule (SU) for $-\Delta_i^t$ the accumulated pseudogradient from client $i$ at step $t$,

$$
\text{Server Update:}
\begin{cases}
\Delta_t = \frac{1}{|\mathcal{S}^t|} \sum_{i \in \mathcal{S}^t} \Delta_i^t, & \widetilde{m}_t = \widetilde{\beta}_1 \widetilde{m}_{t-1} + (1 - \widetilde{\beta}_1)\Delta_t, \\
\widetilde{v}_t = \widetilde{v}_{t-1} + \Delta_t^2, & x_t = x_{t-1} + \eta \frac{\widetilde{m}_t}{\sqrt{\widetilde{v}_t} + \tau}.
\end{cases}
\tag{SU}
$$

Here, $\widetilde{v}_t$ is the sum of squared server-side pseudogradient $-\Delta_t$, and $\widetilde{\beta}_1$ is the momentum coefficient controlling the moving average $\widetilde{m}_t$ of $-\Delta_t$. The set $\mathcal{S}_t \subset [N]$ gives the index of all participating clients at round $t$, and $\tau$ is a constant. An extension to the case when Adam is selected as the server optimizer is given in Appendix C.2. After obtaining an updated global preconditioner $\widetilde{v}_t$ at each communication round, in $\text{FedAda}^2$, the server does not communicate $\widetilde{v}_t$ to the participating clients; instead, each client only receives $x_t$ and initializes the local preconditioners from zero. Empirically, we demonstrate this simple strategy does not degrade the performance relative to the alternative of transmitting global preconditioners, while being communication efficient for adaptive methods beyond AdaGrad (Section 6.1). In addition to communication reduction, this approach enables the use of different optimizers on the server and clients, as the server and client can maintain independent gradient statistics estimates. We discuss the theoretical guarantees/implications of this general framework in Section 5.1 and Appendix D.

**Addressing Client-Side Resource Constraints.** To accommodate local memory restrictions, we employ existing memory-efficient optimizers for all clients. Our framework allows any such optimizer to be used, including a heterogeneous mixture within each communication round. We provide a convergence guarantee for a very broad class of optimizer strategies in Theorem 6. We note that in order for convergence to be guaranteed, the memory-efficient optimizer must satisfy the conditions of Theorem 25, which are non-restrictive[2]. The $\text{FedAda}^2$ framework is summarized in Algorithm 1 below, presented in a simplified form. Local statistics or global statistics refer to those used to construct preconditioners (e.g., first or second moment).

---

**Algorithm 1** $\text{FedAda}^2$: Efficient Jointly Adaptive Optimization Framework (Simplified)

---

**Require:** Init model $x_0$, total number of clients $N$, total rounds $T$
1: **for** $t = 1, \ldots, T$ **do**
2:     Sample subset $\mathcal{S}^t \subset [N]$ of clients using any sampling scheme
3:     **for** each client $i \in S_l^t$ (in parallel) **do**
4:         $x_{i,0}^t \leftarrow x_{t-1}$
5:         **(Main Ingredient 1) Zero Local Preconditioner Initialization:** local_statistics $\leftarrow 0$
6:         **for** $k = 1, \ldots, K$ **do**
7:             Draw gradient $g_{i,k}^t \sim \mathcal{D}_{i,\text{grad}}(x_{i,k-1}^t)$
8:             **(Main Ingredient 2)** $x_{i,k}^t \leftarrow \text{Efficient\_Adaptive\_Optim.}(x_{i,k-1}^t, g_{i,k}^t, \text{local\_statistics})$
9:         **end for**
10:        $\Delta_i^t = x_{i,K}^t - x_{t-1}$
11:     **end for**
12:     $x^t \leftarrow \text{Adaptive\_Optim.}(\{\Delta_i^t\}_{i \in S_l^t}, \text{global\_statistics})$ (for example, Eq. (SU))
13: **end for**

---

During implementation, we have chosen to instantiate $\text{FedAda}^2$ with SM3 (Anil et al., 2019) adaptations of Adam and Adagrad as the memory-efficient local optimizers (Appendix B) due to its strong empirical performance. Intuitively, SM3 exploits natural activation patterns observed in

---

[2]It can easily be shown that Adam, AdaGrad, SGD, as well as their memory-efficient counterparts (Anil et al., 2019) for the first two, all satisfy the optimizer conditions for guaranteed convergence.

model gradients to efficiently synthesize a low-rank approximation of the preconditioner. It maintains the statistics in the granularity of parameter groups instead of individual coordinates. Our analyses in Section 5 hold for a class of memory-efficient local optimizers.

## 5 CONVERGENCE ANALYSES

One of the challenges in proving the convergence bound for jointly adaptive systems lies in handling local adaptivity with applying multiple updates locally. Furthermore, server adaptivity actively interferes and complicates the analysis. To address these issues, we assume access to full batch client gradients which are bounded. To proceed with the convergence analysis, we make the following assumptions where the $\ell_2$ norm is taken by default.

**Assumption 1 ($L$-smoothness).** The local objectives are $L$-smooth and satisfy $\|\nabla F_i(x) - \nabla F_i(y)\| \leq L\|x - y\|$ for all $x, y \in \mathcal{X}$ and $i \in [N]$.

**Assumption 2 (Bounded Gradients).** The local objective gradient is bounded by $\left|[\nabla F_i(x, z)]_j\right| \leq G$ for $j \in [d]$, $i \in [N]$, and $z \sim \mathcal{D}_i$.

These assumptions are standard within the literature and have been used in previous works (Reddi et al., 2021; Xie et al., 2020; Wang et al., 2020; Li et al., 2020b). We note that Assumption 2 implies $|\nabla F_i(x)| \leq G$ for $x \in \mathcal{X}$ via Jensen and integrating over $z \sim \mathcal{D}_i$. In particular, this delineates an $\widetilde{L}$-Lipschitz family of client objectives given that the arguments are $\eta_\ell \varepsilon_s$-bounded away from each other,

$$\|\nabla F_i(x) - \nabla F_j(y)\| \leq \widetilde{L}\|x - y\| := \frac{2\sqrt{d}G}{\eta_\ell \varepsilon_s}\|x - y\|$$

for $i, j \in [N]$ and $\|x - y\| \geq \eta_\ell \varepsilon_s$. Here, $\varepsilon_s$ is an epsilon smoothing term that activates on the client side. This quantity is used in a gradient clipping step in FedAda$^2$ (full version Algorithm 5), where if the local gradient update is negligibly small in magnitude, the gradient is autonomously clipped to 0. $\eta_\ell > 0$ is the local learning rate, and in particular, we note that $\widetilde{L} = \Theta(\eta_\ell^{-1})$. By taking $\varepsilon_s \to 0$, our algorithm recovers federated algorithms that do not utilize local gradient clipping. The definition of $\varepsilon_s$ is for analysis purposes; in experiments, we take $\varepsilon_s$ to be a negligible value so that $m_k$ is not 0.

We now provide a convergence bound for the general, non-convex case under local gradient descent and partial client participation. The full theorem statement is provided in Appendix D as Theorem 25. The SM3 instantiation of FedAda$^2$, as well as the generalization to the case where we use Adam as the server/client optimizers are provided in Appendices C.1 and C.2.

**Theorem 6** (Simplified). *Under Assumptions 1 and 2 as well as some non-restrictive optimizer update conditions (given in Theorem 25), for any choice of initialization $x_0$, Algorithm 1 deterministically satisfies*

$$\min_{t \in [T]} \|\nabla f(x_{t-1})\|^2 \leq \frac{\Psi_1 + \Psi_2 + \Psi_3 + \Psi_4 + \Psi_5}{\Psi_6}$$

*where asymptotically,*

$$\psi_1 = \Theta(1), \ \psi_2 = \eta^2 \eta_\ell^2 T, \ \psi_3 = \eta \eta_\ell^2 T, \ \psi_4 = \eta \eta_\ell \log(1 + T\eta_\ell^2)$$

*and*

$$\psi_5 = \begin{cases} \eta^3 \eta_\ell^3 T & \text{if } \mathcal{O}(\eta_\ell) \leq \mathcal{O}(1) \\ \eta^3 \eta_\ell T & \text{if } \Theta(\eta_\ell) > \Omega(1) \end{cases}, \quad \psi_6 = \begin{cases} \eta \eta_\ell T & \text{if } \mathcal{O}(T\eta_\ell^2) \leq \mathcal{O}(1) \\ \eta \sqrt{T} & \text{if } \Theta(T\eta_\ell^2) > \Omega(1) \end{cases}.$$

We defer the detailed proofs to Appendix C, D. We make no other assumptions on local or global learning rates to extract the most general use of Theorem 6. We have the following two corollaries.

**Corollary 7.** *Any of the following conditions are sufficient to ensure convergence of Algorithm 1:*

$$(A): \quad \eta_\ell \leq \mathcal{O}(T^{-\frac{1}{2}}) \quad \text{for} \quad \Omega(T^{-1}) < \eta \eta_\ell < \mathcal{O}(1),$$

$$(B): \quad \eta_\ell = \Theta(T^{-\frac{49}{100}}) \quad \text{for} \quad \Omega(T^{-\frac{1}{2}}) < \eta < \mathcal{O}(T^{\frac{12}{25}}).$$

**Corollary 8.** *Algorithm 1 converges at rate $\mathcal{O}(T^{-1/2})$.*

In particular, $\eta_\ell$ must necessarily decay to establish convergence in Theorem 6. However, striking a balance between local and global learning rates provably allows for greater than $\Omega(T^{1/3})$ divergence in the server learning rate without nullifying the desirable convergence property. This theoretically demonstrates the enhanced resilience of adaptive client-side federated learning algorithms to mitigate suboptimal choices of server learning rates.

## 5.1 DISCUSSION OF CONVERGENCE BOUND

There have been several recent works exploring adaptivity and communication efficiency in federated learning. The convergence rate in Corollary 8 matches the state of the art for federated non-convex optimization methods (Reddi et al., 2021; Wang et al., 2022; Tong et al., 2020; Sun et al., 2023; Xie et al., 2019; Chen et al., 2020). However, to the best of our knowledge, there are no known convergence results of jointly adaptive federated optimization that explicitly support several popular methods including Adam and AdaGrad.

**Generality of `FedAda`$^2$: Federated Blended Optimization.** The gradient descent setting used in the analysis of Theorem 6 is conceptually equivalent to accessing oracle client workers capable of drawing their entire localized empirical data stream. While this constraint is a limitation of our theory, it enables us to derive stronger results and induce additional adaptive frameworks for which our analysis generalizes. For instance, our bound deterministically guarantees asymptotic stabilization of the minimum gradient, regardless of initialization or client subsampling procedure. In Appendix D, we present the `FedAda`$^2$ framework under its most general, technical form, which we also call Federated Blended Optimization (Algorithm 5).

Blended optimization distributes local optimizer strategies during the subsampling process, which are formalized as functions that take as input the availability of client resources and outputs hyperparameters such as delay step size $z$ or choice of optimizer (Adam, AdaGrad, SGD, etc). These may be chosen to streamline model training based on a variety of factors, such as straggler mitigation or low availability of local resources. In particular, this framework permits the deployment of different adaptive optimizers per device for each round, enhancing the utility of communication-efficient frameworks that do not retain preconditioners between clients or between the server and client. This flexibility is especially beneficial in scenarios where there are inconsistencies between server and client adaptive optimizer choices.

## 6 EMPIRICAL EVALUATION

In this section, we empirically demonstrate the performance of `FedAda`$^2$ compared with several baselines that are either non-adaptive or adaptive but inefficient. We first present our main results by comparing different instantiations of `FedAda`$^2$ with more expensive jointly adaptive baselines and non-jointly adaptive methods in Section 6.1. We then investigate the effects of hyperparameters in more detail in Section 6.2. We repeat every run for 20 times under different random seeds for statistical significance, and report 95% confidence intervals as shaded error regions in all plots.

**Evaluation Setup.** We explore the impact of adaptivity on both text and image datasets, i.e., StackOverflow (Exchange, 2021), CIFAR-100 (Krizhevsky, 2009), and GLD-23K (Weyand et al., 2020). In StackOverflow, each client is a single user posting on the StackOverflow website. Due to the sensitivity nature of the data in federated networks, we evaluate `FedAda`$^2$ in both private and non-private settings with a logistic regression model. For images, we finetune vision transformers (ViT-S Sharir et al. (2021)) pretrained on the ImageNet-21K dataset (Ridnik et al., 2021) on the GLD-23K subset of the Google Landmarks dataset (Weyand et al., 2020), which represents a domain shift onto natural user-split pictorial data. We use the same model on the CIFAR100 dataset (Krizhevsky, 2009), where we partition the data using LDA (Blei et al., 2003) with $\alpha = 0.001$. Details for federated dataset statistics, learning tasks, and hyperparameter tuning are provided in Appendix H.

**Description of Baselines.** Throughout this section, we compare with the following baselines. FedAvg is the vanilla FL algorithm introduced in McMahan et al. (2017), without any additional momentum for the server-side aggregation. FedAdaGrad or FedAdam are two examples of server-only adaptive federated optimization methods (Reddi et al., 2021), where the server-side model updates are performed by an adaptive optimizer (e.g., AdaGrad/Adam) instead of vanilla averaging. 'Direct

Joint Adaptivity' (named *Direct Joint Adap.* in the captions) indicates a jointly adaptive training regimen, where server-side preconditioners are transmitted to clients at every communication round. For instance, we may denote one such setup as 'AdaGrad-AdaGrad', where server-side AdaGrad preconditioners are transmitted to the client-side AdaGrad optimizers as initialization. Removing server-side preconditioner transmission and using zero initialization of client-side preconditioners results in the 'Joint Adaptivity without Preconditioner Communication' (named *Joint Adap. w/o Precond. Commu.* in the captions) baseline, which is communication-efficient. Further compressing the local preconditioners using SM3 (Anil et al., 2019) to account for client memory resource limitations gives `FedAda`$^2$. Therefore, the baselines and `FedAda`$^2$ may be viewed as naturally motivated variations via the addition of adaptive updates and memory-efficient optimizers.

## 6.1 EMPIRICAL PERFORMANCE OF FEDADA$^2$

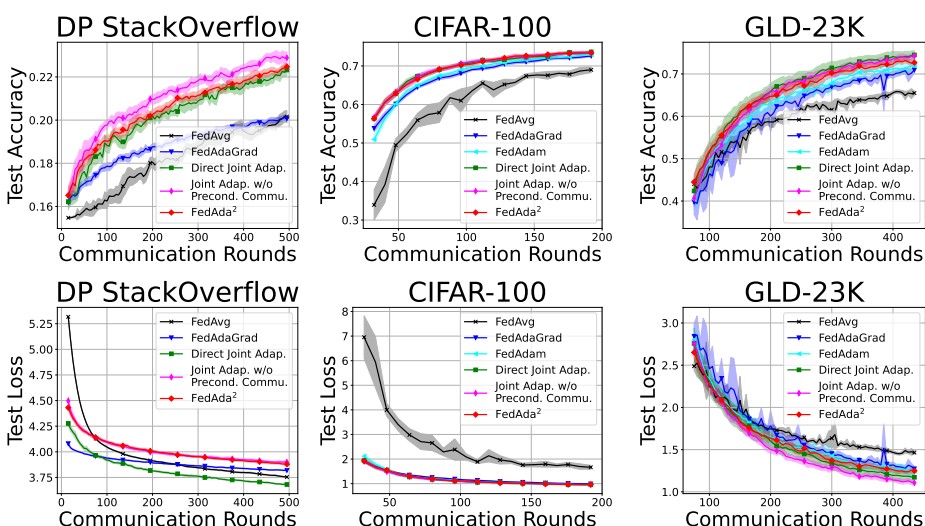

Figure 1: (Top) Test accuracies on StackOverflow, CIFAR-100, and GLD-23K datasets. For StackOverflow, we evaluate the performance of `FedAda`$^2$ and baselines under differential privacy (DP) constraints. If not otherwise specified, StackOverflow uses AdaGrad for adaptivity, while CIFAR-100 and GLD-23K use Adam. We see that jointly adaptive algorithms demonstrate improved performance over FedAvg and server-only adaptive systems. Further, not transmitting the global preconditioner does not degrade performance, and `FedAda`$^2$ preserves the benefits of joint adaptivity while maintaining efficiency. (Bottom) Corresponding test losses for the three datasets of `FedAda`$^2$ and benchmarks. We also note that on the StackOverflow dataset, there is a mismatch between best-performing methods in terms of test accuracies and losses.

**Results of `FedAda`$^2$ under Differential Privacy (DP).** DP is a mathematical framework that can quantify the degree to which sensitive information about individual data points may be purposely obscured during model training, providing rigorous privacy measurement (Abadi et al., 2016; Mironov, 2017; Dwork et al., 2006). For the StackOverflow dataset, we investigate the setting of noise multiplier $\sigma = 1$, which provides a privacy budget of $(\varepsilon, \delta) = (13.1, 0.0025)$ with optimal Rényi-Differential Privacy (RDP) (Mironov, 2017) order 2.0 (Appendix H.1). As mentioned in the beginning of this section, we use AdaGrad to be both server-side and client-side adaptive methods. Notably, we see in our experiments that the proposed technique of initializing client-side preconditioners from zero can even outperform direct joint adaptivity in this setting, where the latter approach transmits the server preconditioner to the client for local updates at every round. Further compressing client-side adaptive preconditioning via `FedAda`$^2$ reduces the performance slightly, but still performs the best among the FedAvg, FedAdaGrad, Direct Joint Adaptivity (AdaGrad-AdaGrad) baselines. In Figure 2, we further demonstrate communication-efficiency of `FedAda`$^2$ by evaluating convergence versus the number of actual transmitted bits.

**`FedAda`$^2$ for Finetuning Vision Transformer Models.** We investigate the performance of finetuning vision transformer models (ViT-S Sharir et al. (2021)) on image data. For all runs on the CI-

FAR100 and GLD-23K datasets, we use Adam as the optimizer everywhere, except for the baseline of FedAdaGrad. For CIFAR-100 (Figure 1 (middle)), direct joint adaptive and server-only adaptive methods (FedAdam and FedAdam) converge faster and achieve higher accuracy than FedAvg. Methods using joint adaptivity (including $\texttt{FedAda}^2$) convergence faster than FedAdam. While 'Direct Joint Adap.' achieves similar performance to $\texttt{FedAda}^2$, $\texttt{FedAda}^2$ is much more memory and communication efficient. Similar trends are observed on GLD-23K (the right column). Furthermore, as a side, we propose to incorporate delayed preconditioner updates (Gupta et al., 2018) on the client-side as an optional step to potentially reduce communication (explained in Appendix B) and show that $\texttt{FedAda}^2$ is robust to delayed local preconditioner updates as well (Appendix I.2).

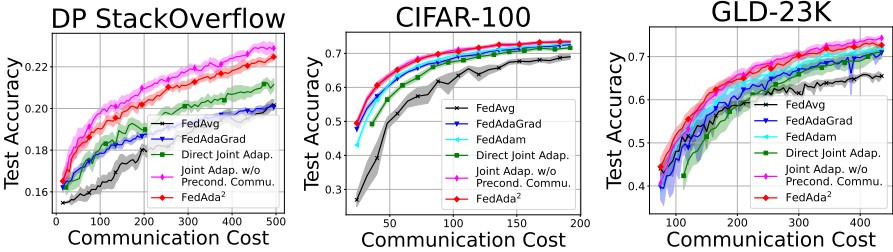

Figure 2: Test accuracies against actual communication cost (total transmitted bits normalized to that of FedAvg) for $\texttt{FedAda}^2$ and baseline methods, using the same settings as in Figure 1. When compared based on communication cost, both 'Joint Adaptivity without Preconditioner Transmission' and $\texttt{FedAda}^2$ demonstrate the fastest convergence.

**Results of Additional Adaptive Setups.** Algorithm 1 provides a general framework, and in Figure 1, we focus on symmetric server-client optimizer configurations (e.g., Adam-Adam, AdaGrad-AdaGrad). In Appendix I.1, Figure 7, we generalize this setting to examine the performance of *asymmetric* server-client adaptivity setups under both jointly adaptive baselines and $\texttt{FedAda}^2$. Our results show that in the Joint Adaptivity w/o Preconditioner Transmission baseline, employing an unbalanced preconditioner (e.g., transmitting the server-side Adam preconditioner to client-side AdaGrad), does not significantly impact performance across a hyperparameter sweep. Similarly, $\texttt{FedAda}^2$ demonstrates robust training dynamics across various adaptivity instantiations, highlighting its effectiveness in enabling efficient jointly adaptive optimization.

## 6.2 EFFECTS OF VARYING CONFIGURATIONS

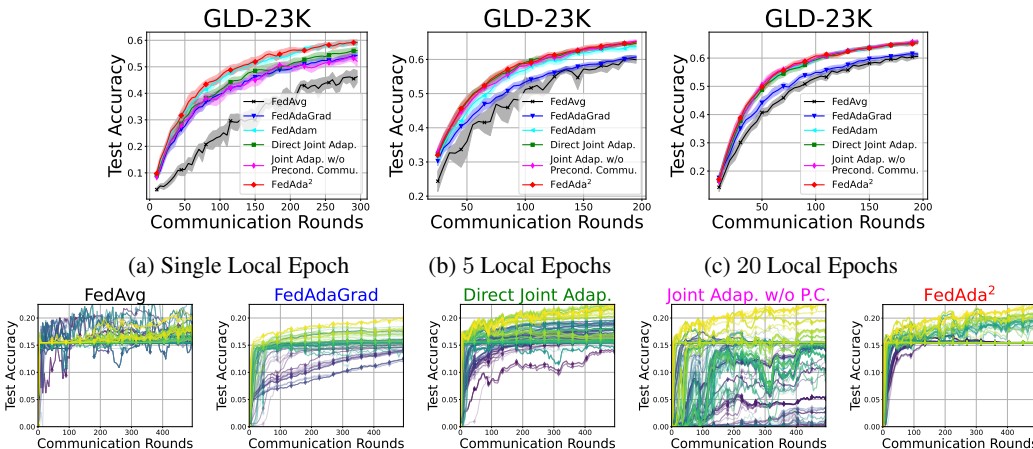

Figure 3: (Top) Algorithm testing performance comparision under varying client resource limitations (i.e., number of local epochs). When resources are constrained, $\texttt{FedAda}^2$ converges the fastest, followed closely by FedAdam. Interestingly, the relative performance advantage of $\texttt{FedAda}^2$ becomes less significant as the number of local epochs increases. (Bottom) We plot all test accuracies obtained during the hyperparameter sweeps detailed in Appendix H.1, with fixed client subsampling random seed. The runs are ranked hierarchically from the lowest to the highest final test loss, with the colors transitioning from lighter to darker shades accordingly.

**Dynamics of `FedAda`$^2$ under a Varying Number of Local Epochs.** In Figure 3 (top), we study the transfer learning setting of a vision model under a highly constrained, moderate, and sufficient client computation budget, corresponding to running 1, 5, and 20 local epochs on the clients. We see that when the number of epochs is low (Figure 3 (a)), `FedAda`$^2$ achieves the best performance, closely followed by FedAdam. Interestingly, as the clients' computational budget increases, the relative performance advantage of `FedAda`$^2$ diminishes. In such scenarios, jointly adaptive benchmarks outperform FedAdam, although the margin is not substantial.

**Sensitivity to Hyperparameters.** In Figure 3 (bottom), we plot test accuracies over the hyperparameter sweeps detailed in Appendix H for `FedAda`$^2$ and all baselines. Server-only adaptivity stabilizes the performance of FedAvg, and direct joint adaptivity further enhances these stabilized accuracies. However, eliminating server preconditioner transmission destabilizes the accuracy, resulting in significantly poorer performance for the worst losses, while retaining the best performing losses. Surprisingly, approximating the preconditioners in a memory-efficient manner using SM3 restabilizes the losses, which we hypothesize is due to the denoising effect of projections during SM3 compression. Interestingly, in the DP setting, zero initialization and compressing gradient statistics (`FedAda`$^2$) achieves even better performance than direct joint adaptivity, when test accuracies over best-performing hyperparameters are averaged over 20 random seeds for convergence (Figure 1, top).

**Summary.** For DP StackOverflow and CIFAR-100 experiments, a natural yet expensive implementation of joint client- and server-side adaptivity with transmitted global preconditioners surpasses the performance of FedAvg and server-only adaptivity. However, full preconditioner transmission incurs significant communication costs, as noted in Section 1. Additionally, the adaptive optimizer substantially increases the memory demand on the client due to the maintenance of auxiliary second-order statistics used to synthesize model updates in every local iteration, which motivates the development of efficient adaptive frameworks. In our empirical evaluations, we consistently found that initializing local preconditioners from zero did not underperform direct joint adaptivity (full server-side preconditioner transmission) after optimal hyperparameter tuning. The performance of joint adaptivity under differential privacy is notable, where this compromise to reduce communication cost even achieved better test performance than the more expensive baseline with full preconditioner transmission. In addition, when evaluating convergence in terms of the actual communicated bits (communication rounds times number of bits per round), `FedAda`$^2$ significantly outperforms direct joint adaptivity (Figure 2), saving significant communication bandwidth. In general, we observe that `FedAda`$^2$ retains the competitive advantage of joint adaptivity while being communication- and memory-efficient. Empirically, avoiding preconditioner transmission and leveraging client-side preconditioner approximations (i.e., `FedAda`$^2$) does not substantively harm the performance of its more expensive variants, and can even surpass the performance of direct joint adaptivity in certain settings (e.g., StackOverflow and GLD-23K under constrained client resources).

## 7 CONCLUSION AND FUTURE WORK

In this work, we introduce `FedAda`$^2$, a class of jointly adaptive algorithms designed to enhance scalability and performance in large-scale, cross-device federated environments. `FedAda`$^2$ is conceptually simple and straightforward to implement. In particular, we show that joint adaptivity is practicable while sidestepping communication bottlenecks and localized memory restrictions. By optimizing communication efficiency and employing localized memory-efficient adaptive optimizers, `FedAda`$^2$ significantly reduces the overhead associated with transferring preconditioners and extra on-device memory cost without degrading model performance. Our empirical results demonstrate the practical benefits of `FedAda`$^2$ in real-world federated learning scenarios. Future research could explore extensions of `FedAda`$^2$ (Section 5.1, Appendix D) to study the training dynamics under alternative, potentially client-specific local optimizer instantiations.

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

## A    IMPORTANCE OF CLIENT-SIDE APDAPTIVITY

**Overview of Student's $t$-distribution.**    For the convenience of the reader, we provide a brief summary of basic properties of the Student's $t$-distribution. Intuitively, the $t$-distribution can be understood as an approximation of the Gaussian with heavier tails. The density is given by

$$f_\nu(t) = \frac{\Gamma\left(\frac{\nu+1}{2}\right)}{\sqrt{\pi\nu}\Gamma\left(\frac{\nu}{2}\right)}\left(1 + \frac{t^2}{\nu}\right)^{-(\nu+1)/2}$$

where $\nu \in \mathbb{R}_{>0}$ is the degree of freedom (or normality parameter), and $\Gamma$ is the gamma function. We recover the normalized Gaussian as the degree of freedom tends to infinity. The first moment is $0$ for $\nu > 1$, and the second moment satisfies $\nu/(\nu - 2)$ for $\nu > 2$ while being infinite for $1 < \nu \leq 2$, where the heavy-tails are most pronounced. Following the convention of Zhang et al. (2020), we refer to a distribution as being heavy-tailed if the second moment is infinite.

The following proposition showcases the utility of local adaptivity in federated learning.

**Proposition 9.** *There exists a federated optimization problem with heavy-tailed client noise which satisfies the following under FedAvg (where appropriate learning rate schedules are chosen for (ii-iv)):*

*(i) Given any client sampling strategy, if the probability $p_i^t$ of client $i$ with heavy-tailed gradient noise being sampled at step $t$ is non-zero, then $\mathbb{E}\|\nabla f(x_{t+1})\|^2 = \infty$ for any nontrivial learning rate schedule $\eta_\ell^t > 0$.*

*(ii) Local adaptivity via client-side AdaGrad bounds the error in expectation as*

$$\lim_{t\to\infty} \mathbb{E}\|x_t - x^*\| \leq \frac{2\sqrt{3}}{1 - \hat\varepsilon} \quad \text{for some} \quad \hat\varepsilon \approx 0,$$

*where $x^*$ is the global optimum.*

*(iii) Furthermore, local adaptivity implicitly constructs a critical Lyapunov stable region which stabilizes the gradient variance via the following inequality which holds once any learned weight enters the region:*

$$\min_{t\in\{1,\ldots,T\}} \mathbb{E}\|\nabla f(x_t)\|^2 \leq \mathcal{O}\left(\frac{1}{T}\right).$$

*(iv) The global gradient variance of the federated problem with heavy-tailed client noise is fully stabilized via*

$$\mathbb{E}[\|\nabla f(x_t)\|^2] \leq 2\|x_0\|^2 + 2\left(\int_1^\infty \frac{1}{x^2}\,\mathrm{d}x\right)^2 \quad \text{for} \quad \forall t \in \{1,\ldots,T\}.$$

This proposition demonstrates that even a single client with heavy-tailed gradient noise is able to instantaneously propagate their volatility to the global model, which destabilizes federated training in expectation. However, recent work (Zhang et al., 2020) has shown that heavy-tailed gradient distributions appear frequently in language model applications, and more generally within model architectures utilizing any kind of attention mechanism, including transformers. To our knowledge, this provable failure mode of distributed training resultant from the unbiased, yet heavy-tailed noise of a singular client has not previously been reported within the literature.

**Proof of (i).**    Let the local stochastic objectives be given by $F_i(x, \xi_i) = x^2/2 + \xi_i x$ where gradient noise follows a $t$-distribution with $i + 1$ degrees of freedom, $\xi_i \sim t_{i+1}$ for $\forall i \in \{1,\ldots,N\}$. This construction is chosen to materialize the setting in which only a singular client suffers from heavy-tailed noise ($i = 1$). Minibatches are sampled with replacement, which ensures that gradient noise in each client epoch are independent amongst and in between any two (possibly identical) clients, and further identically distributed conditional on the client ID $i$. Clearly, the global objective is

$$f(x) = \frac{1}{N}\sum_{i=1}^N \mathbb{E}_{\xi_i}[f_i(x, \xi_i)] = \frac{1}{N}\mathbb{E}\left[\frac{N}{2}x^2 + \sum_{i=1}^N \xi_i x\right] = \frac{1}{2}x^2.$$

For global step $t$, we subsample clients $\mathcal{S}^t$ following any sampling strategy, where $\mathcal{C}^t$ is the collection of all possible multisets $\mathcal{S}_r^t$ whose elements indicate (possibly repeated) client selection, with associated probabilities $p_C^t(r) > 0$ of realization for $r \in [|\mathcal{C}^t|]$. Assume that $1 \in \mathcal{S}_m^t$ for some $m$.

Then, FedAvg updates may be written

$$x_{t+1} = x_t - \frac{\eta_\ell}{|\mathcal{S}^t|} \sum_{i \in \mathcal{S}^t} \sum_{\ell=1}^K g_{i,\ell}^t$$

which gives the squared length of the global gradient under expectation as

$$\mathbb{E}_t \|\nabla f(x_{t+1})\|^2 = \mathbb{E}_t \left\| x_t - \frac{\eta_\ell}{|\mathcal{S}^t|} \sum_{i \in \mathcal{S}^t} \sum_{\ell=1}^K \left( \nabla f(x_{i,\ell-1}^t) + \xi_{i,\ell-1}^t \right) \right\|^2$$

$$= \mathbb{E}_{\xi|t} \mathbb{E}_{\mathcal{S}^t|\xi,t} \left\| x_t - \frac{\eta_\ell}{|\mathcal{S}^t|} \sum_{i \in \mathcal{S}^t} \sum_{\ell=1}^K \left( \nabla f(x_{i,\ell-1}^t) + \xi_{i,\ell-1}^t \right) \right\|^2$$

$$= \sum_{r=1}^{|\mathcal{C}^t|} \mathbb{E}_{\xi|t} p_C^t(r) \left\| x_t - \frac{\eta_\ell}{|\mathcal{S}_r^t|} \sum_{i \in \mathcal{S}_r^t} \sum_{\ell=1}^K \left( \nabla f(x_{i,\ell-1}^t) + \xi_{i,\ell-1}^t \right) \right\|^2$$

$$\geq p_C^t(m) \mathbb{E}_{\xi|t} \left\| x_t - \frac{\eta_\ell}{|\mathcal{S}_m^t|} \sum_{i \in \mathcal{S}_m^t} \sum_{\ell=1}^K \left( x_{i,\ell-1}^t + \xi_{i,\ell-1}^t \right) \right\|^2$$

where in the second equality we have conditioned on local gradient noise $\xi$ and stochastic realizations up to timestep $t$, using the law of iterated expectations. Recursively unravelling $x_{i,\ell-1}^t$ in terms of sampled noise and $x_{i,0}^t = x_t$ gives

$$x_{i,\ell-1}^t = x_{i,\ell-2}^t - \eta_\ell g_{i,\ell-2}^t = x_{i,0}^t - \eta_\ell \sum_{p=0}^{\ell-2} g_{i,p}^t$$

$$= x_{i,0}^t - \eta_\ell \left( \sum_{p=0}^{\ell-2} \nabla f(x_{i,p}^t) + \xi_{i,p}^t \right)$$

$$= x_{i,0}^t - \eta_\ell \left( \sum_{p=0}^{\ell-2} x_{i,p}^t + \xi_{i,p}^t \right)$$

$$= a_t x_t - \sum_{p=0}^{\ell-2} a_{i,p}^t \xi_{i,p}^t$$

where $a_t, a_{i,p}^t \in \mathbb{Q}[\eta_\ell]$ are polynomial functions of the learning rate with rational coefficients. Therefore, we have for $b_{i,p}^t \in \mathbb{Q}[\eta_\ell]$

$$p_C^t(m) \mathbb{E}_{\xi|t} \left\| x_t - \frac{\eta_\ell}{|\mathcal{S}_m^t|} \sum_{i \in \mathcal{S}_m^t} \sum_{\ell=1}^K \left( a_t x_t - \sum_{p=0}^{\ell-2} a_{i,p}^t \xi_{i,p}^t + \xi_{i,\ell-1}^t \right) \right\|^2$$

$$= p_C^t(m) \mathbb{E}_{\xi|t} \left\| \left( 1 - \frac{\eta_\ell}{|\mathcal{S}_m^t|} \sum_{i \in \mathcal{S}_m^t} \sum_{\ell=1}^K a_t \right) x_t + \frac{\eta_\ell}{|\mathcal{S}_m^t|} \sum_{i \in \mathcal{S}_m^t} \sum_{\ell=1}^K \left( \sum_{p=0}^{\ell-2} a_{i,p}^t \xi_{i,p}^t + \xi_{i,\ell-1}^t \right) \right\|^2$$

$$= p_C^t(m) \mathbb{E}_{\xi|t} \left\| \left( 1 - \frac{\eta_\ell}{|\mathcal{S}_m^t|} \sum_{i \in \mathcal{S}_m^t} \sum_{\ell=1}^K a_t \right) x_t \right\|^2 + \frac{\eta_\ell^2 p_C^t(m)}{|\mathcal{S}_m^t|^2} \mathbb{E}_{\xi|t} \left\| \sum_{i \in \mathcal{S}_m^t} \left( \sum_{p=0}^{K-2} b_{i,p}^t \xi_{i,p}^t + \xi_{i,K-1}^t \right) \right\|^2$$

$$\geq \frac{\eta_\ell^2 p_C^t(m) \mathbb{E} \left\| \xi_{1,K-1}^t \right\|^2}{|\mathcal{S}_m^t|^2} = \infty,$$

where we have used that $\xi_{i,\ell}^t \sim t_{i+1}$ independently with mean 0, for all permissible $i$, $\ell$, and $t$.

**Proof of (ii).** We specialize to the setting with client-side AdaGrad with $K = 1$. Assume that clients $S^t$ have been selected to participate in the round, which gives the update as

$$x_{t+1} = x_t - \frac{\eta_\ell}{|\mathcal{S}^t|} \sum_{i \in \mathcal{S}^t} \sum_{\ell=1}^{K} \frac{g_{i,\ell}^t}{\|g_{i,\ell}^t\| + \varepsilon} \tag{2}$$

$$= x_t - \frac{\eta_\ell}{|\mathcal{S}^t|} \sum_{i \in \mathcal{S}^t} \frac{\nabla f(x_{i,0}^t) + \xi_{i,1}^t}{\|\nabla f(x_{i,0}^t) + \xi_{i,1}^t\| + \varepsilon}$$

$$= x_t \left( 1 - \frac{\eta_\ell}{|\mathcal{S}^t|} \sum_{i \in \mathcal{S}^t} \frac{1}{\|x_t + \xi_i\| + \varepsilon} \right) - \frac{\eta_\ell}{|\mathcal{S}^t|} \sum_{i \in \mathcal{S}^t} \frac{\xi_i}{\|x_t + \xi_i\| + \varepsilon}$$

where we have gradually simplified notation. Noting that

$$\int \frac{1}{\|x_t + \xi_i\| + \varepsilon} \, p(\xi_i) \, \mathrm{d}\xi_i \le \frac{1}{\varepsilon},$$

setting $\eta_\ell \le \varepsilon$ gives

$$\|\nabla f(x_{t+1})\| = \|x_{t+1}\| \le \|x_t\| \cdot \left( 1 - \frac{\eta_\ell}{|\mathcal{S}^t|} \sum_{i \in \mathcal{S}^t} \frac{1}{\|x_t + \xi_i\| + \varepsilon} \right) + \frac{\eta_\ell}{|\mathcal{S}^t|} \sum_{i \in \mathcal{S}^t} \frac{\|\xi_i\|}{\|x_t + \xi_i\| + \varepsilon}. \tag{3}$$

Using $\mathbb{E}_t$ to denote expectation conditional over realizations up to step $t$, we have

$$\mathbb{E}_t \|x_{t+1}\| \le \|x_t\| \cdot \left( 1 - \frac{\eta_\ell}{|\mathcal{S}^t|} \mathbb{E}_t \left[ \sum_{i \in \mathcal{S}^t} \frac{1}{\|x_t + \xi_i\| + \varepsilon} \right] \right) + \frac{\eta_\ell}{|\mathcal{S}^t|} \sum_{i \in \mathcal{S}^t} \mathbb{E}_t \left[ \frac{\|\xi_i\|}{\|x_t + \xi_i\| + \varepsilon} \right].$$

To further bound the right hand side, consider the functional

$$I_i(\varepsilon) := \int \frac{1}{\|x_t + \xi_i\| + \varepsilon} \, p_{i+1}(\xi_i) \, \mathrm{d}\xi_i,$$

where clearly

$$I_i(0) \ge \int_{-x_t^-}^{-x_t^+} \frac{1}{\|x_t + \xi_i\|} \, p_{i+1}(\xi_i) \, \mathrm{d}\xi_i \approx \int_{0^-}^{0^+} \frac{p_{i+1}(-x_t)}{|x|} \, \mathrm{d}x = \infty$$

and $I_i(1) < 1$. By continuity and strict decay of $I_i(\varepsilon)$, there exists $1 \gg \hat{\varepsilon}_i > 0$ and $\varepsilon_i \in (0,1]$ such that for all $i \in [N]$, we have $1 > I_i(\varepsilon) \ge 1 - \hat{\varepsilon}_i$ for $\varepsilon \in [\varepsilon_i, 1]$. Taking $\varepsilon \in [\max_{i \in [N]} \varepsilon_i, 1]$ and $\hat{\varepsilon} := \max_{i \in [N]} \hat{\varepsilon}_i$, we thus obtain

$$\mathbb{E}_t \|x_{t+1}\| \le \|x_t\| \cdot (1 - \eta_\ell(1 - \hat{\varepsilon})) + \frac{\eta_\ell}{|\mathcal{S}^t|} \sum_{i \in \mathcal{S}^t} \mathbb{E}_t \left[ \frac{\|\xi_i\|}{\|x_t + \xi_i\| + \varepsilon} \right]. \tag{4}$$

To bound the remaining term, it is easy to show that $\|\xi_i\| p_{i+1}(\xi_i)$ is symmetric around the origin $O$, and strictly increases from 0 to $(3/2 + 2/(i+1))^{-1/2}$ while strictly decreasing afterwards. Defining the even extension of

$$h_{i+1}(\xi_i) = \begin{cases} -\frac{x}{(3/2 + 2/(i+1))^{-1/2}} + \sup_{\xi_i \in \mathbb{R}} \|\xi_i\| p_{i+1}(\xi_i) + \epsilon & \text{for } 0 \le \xi_i \le \left( \frac{3}{2} + \frac{2}{i+1} \right)^{-\frac{1}{2}}, \\ \|\xi_i\| p_{i+1}(\xi_i) & \text{for } \xi_i > \left( \frac{3}{2} + \frac{2}{i+1} \right)^{-\frac{1}{2}} \end{cases}$$

to be $h_{i+1}(\xi_i)$ for small $1 \gg \epsilon > 0$, we note that $1/(\|x_t + \xi_i\| + \varepsilon)$ analogously is symmetric around $\xi_i = -x_t$ while decaying with respect to the argument $\|x_t + \xi_i\|$. As $h_{i+1}(\xi_i)$ is symmetric around $O$ and decays moving to the left and right of $O$, by matching monotonicity and maxima with $1/(\|x_t + \xi_i\| + \varepsilon)$, we conclude that the left hand side of (5) is maximized for $x_t = 0$:

$$\mathbb{E}_t \left[ \frac{\|\xi_i\|}{\|x_t + \xi_i\| + \varepsilon} \right] \le \int \frac{h_{i+1}(\xi_i)}{\|\xi_i\| + \varepsilon} \, \mathrm{d}\xi_i = B_i. \tag{5}$$

Asymptotically as $\xi_i \to \infty$, we have

$$\frac{h_{i+1}(\xi_i)}{\|\xi_i\| + \varepsilon} \lesssim p_{i+1}(\xi_i),$$

which gives that $B_i < \infty$. Letting $B := \max_{i \in [N]} B_i$ and scheduling the learning rate $\eta_\ell^t = 1/((t + t_0)(1 - \hat{\varepsilon}))$ where $t_0$ is the smallest positive integer satisfying $\eta_\ell^t < \varepsilon$ for all $t$, we thus conclude

$$\mathbb{E}\|x_{t+1}\| \leq \frac{t + t_0 - 1}{t + t_0}\mathbb{E}\|x_t\| + \frac{B}{(t + t_0)(1 - \hat{\varepsilon})}$$

$$\leq \frac{t + t_0 - 2}{t + t_0}\mathbb{E}\|x_{t-1}\| + \frac{2B}{(t + t_0)(1 - \hat{\varepsilon})}$$

$$\leq \cdots \leq \frac{t_0 - 1}{t + t_0}\mathbb{E}\|x_0\| + \frac{(t + 1)B}{(t + t_0)(1 - \hat{\varepsilon})}$$

$$\leq \mathcal{O}\left(\frac{1}{t}\right) + \frac{B}{1 - \hat{\varepsilon}}.$$

As this bound holds for any choice of client subsample $S^t$, we are done. It is easy to show by straightforward integration that $B < 2\sqrt{3}$.

**Proof of (iii).** Our strategy is to locate a 1-shot stabilization regime of the gradient norm that is formed via client adaptivity, which may be viewed as a Lyapunov stable region of the optimum $x^*$. From (3) and Jensen,

$$\|x_{t+1}\|^2 \leq 2\|x_t\|^2 \cdot \left(1 - \frac{\eta_\ell}{|\mathcal{S}^t|}\sum_{i \in \mathcal{S}^t}\frac{1}{\|x_t + \xi_i\| + \varepsilon}\right)^2 + \frac{2\eta_\ell^2}{|\mathcal{S}^t|^2}\left(\sum_{i \in \mathcal{S}^t}\frac{\|\xi_i\|}{\|x_t + \xi_i\| + \varepsilon}\right)^2$$

$$\leq 2\|x_t\|^2 \cdot \left(1 - \frac{\eta_\ell}{|\mathcal{S}^t|}\sum_{i \in \mathcal{S}^t}\frac{1}{\|x_t + \xi_i\| + \varepsilon}\right)^2 + \frac{2\eta_\ell^2}{|\mathcal{S}^t|}\sum_{i \in \mathcal{S}^t}\left(\frac{\|\xi_i\|}{\|x_t + \xi_i\| + \varepsilon}\right)^2.$$

We now impose $\eta_\ell \leq 2\varepsilon$, while letting $\|x_t\| < \delta$ for some $\delta \in \mathbb{R}_{>0}$. Taking expectations gives

$$\mathbb{E}_t\|x_{t+1}\|^2 \leq 2\|x_t\|^2 + \frac{2\eta_\ell^2}{|\mathcal{S}^t|}\sum_{i \in \mathcal{S}^t}\mathbb{E}_t\left(\frac{\|\xi_i\|}{\|x_t + \xi_i\| + \varepsilon}\right)^2,$$

and by similar arguments to the proof of **(ii)**, the summands of the second term are bounded uniformly by $\widetilde{B}$ which yields

$$\mathbb{E}\|x_{t+1}\|^2 \leq 2\delta^2 + 2\eta_\ell^2\widetilde{B}.$$

Setting $\delta, \eta_\ell^t \leq \mathcal{O}(1/\sqrt{T})$ immediately gives the desired inequality.

**Proof of (iv).** An advantage of client-side adaptive optimization is the autonomous normalization and clipping of the stochastic gradients. Let $\eta_\ell^t := 1/t^2$. Telescoping (2) gives

$$x_{T+1} = x_0 - \sum_{t=1}^{T}\frac{\eta_\ell^t}{|\mathcal{S}^t|}\sum_{i \in \mathcal{S}^t}\sum_{\ell=1}^{K}\frac{g_{i,\ell}^t}{\|g_{i,\ell}^t\| + \varepsilon},$$

which implies

$$\|x_{T+1} - x_0\| = \left\|\sum_{t=1}^{T}\frac{\eta_\ell^t}{|\mathcal{S}^t|}\sum_{i \in \mathcal{S}^t}\sum_{\ell=1}^{K}\frac{g_{i,\ell}^t}{\|g_{i,\ell}^t\| + \varepsilon}\right\|$$

$$\implies \left|\|x_{T+1}\| - \|x_0\|\right| \leq \left\|\sum_{t=1}^{T}\frac{\eta_\ell^t}{|\mathcal{S}^t|}\sum_{i \in \mathcal{S}^t}\sum_{\ell=1}^{K}\frac{g_{i,\ell}^t}{\|g_{i,\ell}^t\| + \varepsilon}\right\|$$

$$\implies \|x_{T+1}\| \leq \|x_0\| + \left\|\sum_{t=1}^{T}\frac{\eta_\ell^t}{|\mathcal{S}^t|}\sum_{i \in \mathcal{S}^t}\sum_{\ell=1}^{K}\frac{g_{i,\ell}^t}{\|g_{i,\ell}^t\| + \varepsilon}\right\|$$

$$\implies \mathbb{E}\|x_{T+1}\|^2 \leq 2\|x_0\|^2 + 2\mathbb{E}\left\|\sum_{t=1}^{T}\frac{\eta_\ell^t}{|\mathcal{S}^t|}\sum_{i \in \mathcal{S}^t}\sum_{\ell=1}^{K}\frac{g_{i,\ell}^t}{\|g_{i,\ell}^t\| + \varepsilon}\right\|^2.$$

Substituting the learning rate schedule gives

$$\mathbb{E} \left\| \sum_{t=1}^{T} \frac{\eta_\ell^t}{|\mathcal{S}^t|} \sum_{i \in \mathcal{S}^t} \sum_{\ell=1}^{K} \frac{g_{i,\ell}^t}{\|g_{i,\ell}^t\| + \varepsilon} \right\|^2 \leq \mathbb{E} \left\| \sum_{t=1}^{T} K \eta_\ell^t \right\|^2$$

$$\leq \mathbb{E} \left\| K \int_1^\infty \frac{1}{x^2} \, \mathrm{d}x \right\|^2.$$

Therefore, we conclude that for any $t$,

$$\mathbb{E}\|x_t\|^2 \leq 2\|x_0\|^2 + 2K^2 \left( \int_1^\infty \frac{1}{x^2} \, \mathrm{d}x \right)^2.$$

## A.1 EXACERBATION OF SINGULAR CLIENT NOISE

**Overview of Cauchy–Lorentz distribution**   For the convenience of the reader, we provide a brief description of the Cauchy distribution $\mathcal{CL}(x_0, \gamma)$. The density is given by

$$f(x; x_0, \gamma) = \frac{1}{\pi \gamma \left[ 1 + \left( \frac{x - x_0}{\gamma} \right)^2 \right]} = \frac{1}{\pi} \left[ \frac{\gamma}{(x - x_0)^2 + \gamma^2} \right],$$

where $x_0$ is the location parameter and $\gamma > 0$ the scale parameter. Note that the Cauchy distribution is an example of "worst case gradient noise" that a federated problem may encounter in its clients. That is, the tails are so heavy that the distribution, despite being symmetric around the origin $O$, does not admit a mean due to being non-(Lebesgue) integrable. In particular, this indicates that the law of large numbers cannot be applied due to uncontrolled stochasticity, which lethally destabilizes pure stochastic gradient descent. Despite this limitation, we provide an example demonstrating that local adaptivity can be utilized to successfully mollify extreme client noise even in this "worst case" setting.

**Proposition 10.** *There exists a generalized federated optimization problem which satisfies the following under FedAvg:*

*(i) Given any client sampling strategy without replacement, if the probability $p_i^t$ of client $i$ with heavy-tailed gradient noise being sampled at each step $t$ is non-zero, then $\mathbb{E}\|\nabla f(x_{t+1})\| = \infty$ or $\mathbb{E}\|\nabla f(x_t)\| = \infty$ for any $t \in \mathbb{Z}_{\geq 1}$ and nontrivial learning rate $\eta_\ell^t > 0$.*

*(ii) Under local adaptivity via client-side AdaGrad, we have bounded gradient length as*

$$\lim_{t \to \infty} \mathbb{E}\|\nabla f(x_t)\| \leq \frac{2}{1 - \hat{\varepsilon}} \quad \text{for some} \quad \hat{\varepsilon} \approx 0.$$

**Proof of (i).**   We provide a similar construction as in the proof of Theorem 9. Let all local stochastic objectives be given by $F_i(x, \xi_i) = x^2/2 + \xi_i x$ where client gradient noise mostly models a Gaussian, $\xi_i \sim \mathcal{N}(0, \sigma_i^2)$ for $\forall i \in \{2, \ldots, N\}$ and $\sigma_i \in \mathbb{R}$. For the first client, we let $\xi_1 \sim \mathcal{CL}(0, \gamma)$ for any $\gamma \in (0, 1/3)$. We sample minibatches with replacement, but clients are selected without replacement. In this case, we must consider a generalized version of the federated objective as strictly speaking, the deterministic local objective

$$\mathbb{E}_{\xi_1}[F_1(x, \xi_1)] = \frac{1}{2}x^2 + x \int \xi_1 \, \mathrm{d}\xi_1$$

does not exist due to extreme stochasticity. That is, even though $\mathcal{CL}(0, \gamma)$ is symmetric around $O$, $\mathbb{E}_{\xi_1}[\xi_1]$ is not Lebesgue integrable. Most importantly, this implies that the law of large numbers cannot be applied. Note that such a construction dislocates this example from the vast majority of convergence results, as most assume bounded variance or controlled gradient noise which sidesteps the consideration of the kind of stochasticity that we explore here entirely. To proceed with the analysis, we use symmetry to define the reasonable objective

$$\mathbb{E}[F_1(x, \xi_1)] = \frac{1}{2}x^2$$

which is consistent with the desired population objective that is distributed across all other clients, though with less noise. As before, we have the convex global objective $f(x) = x^2/2$. Note that it can be shown that the empirical mean of the Cauchy distribution follows the Cauchy distribution, that is, the CL-distribution is stable.

As the general case has been handled in Theorem 9 (i), we specialize to $K = 1$. To simplify notation, assume that participating clients have been selected as $\mathcal{S}^t$, where client 1 participates. Then, the FedAvg update may be written

$$x_{t+1} = x_t - \frac{\eta_\ell}{|\mathcal{S}^t|} \sum_{i \in \mathcal{S}^t} g_{i,1}^t$$

which gives the length of the global gradient under expectation as

$$\mathbb{E}\|\nabla f(x_{t+1})\| = \mathbb{E}\left\| x_t - \frac{\eta_\ell}{|\mathcal{S}^t|} \sum_{i \in \mathcal{S}^t} \left(\nabla f(x_{i,0}^t) + \xi_{i,1}^t\right) \right\|$$

$$\geq \mathbb{E}\left\| \frac{\eta_\ell}{|\mathcal{S}^t|} \xi_{1,1}^t \right\| - \mathbb{E}\left\| \left(1 - \frac{\eta_\ell}{|\mathcal{S}^t|}\right) x_t - \frac{\eta_\ell}{|\mathcal{S}^t|} \sum_{i \in \mathcal{S}^t \setminus \{1\}} \left(\nabla f(x_{i,0}^t) + \xi_{i,1}^t\right) \right\|$$

$$\geq \mathbb{E}\left\| \frac{\eta_\ell}{|\mathcal{S}^t|} \xi_{1,1}^t \right\| - \mathbb{E}\left\| (1 - \eta_\ell) x_t - \frac{\eta_\ell}{|\mathcal{S}^t|} \sum_{i \in \mathcal{S}^t \setminus \{1\}} \xi_{i,1}^t \right\|$$

$$\geq \mathbb{E}\left\| \frac{\eta_\ell}{|\mathcal{S}^t|} \xi_{1,1}^t \right\| - \mathbb{E}\|(1 - \eta_\ell) x_t\| - \frac{\eta_\ell}{|\mathcal{S}^t|} \sum_{i \in \mathcal{S}^t \setminus \{1\}} \mathbb{E}\left\| \xi_{i,1}^t \right\|$$

Note that we allow $\eta_\ell = 1$. As $\mathbb{E}\left\| \xi_{i,1}^t \right\| < \infty$ for $i \in \{2, \ldots, N\}$, we thus have

$$\mathbb{E}\|\nabla f(x_{t+1})\| + |1 - \eta_\ell| \mathbb{E}\|\nabla f(x_t)\| \geq \infty$$

which gives the desired result.

**Proof of (ii).** As we intervened only on gradient noise while preserving client objectives, an analogous proof strategy used in Theorem 9 (ii) carries through. The only difference is the value of $B$, which may be computed as being upper bounded by 2 for $\gamma < 1/3$.

### A.2 FEDAVG AND STOCHASTIC GRADIENT DESCENT ARE DEEPLY REMORSEFUL

In Appendix A, we have provided two localized examples of how heavy-tailed gradient noise can destabilize distributed training. In this subsection, we prove that this is an instantiation of a more general phenomenon in which federated learning with a $\mu$-strongly convex global objective collapses to an analogous failure mode. We begin by motivating a precise definition of heavy-tailed noise previously reported in the literature (Zhang et al., 2020) for completeness.

**Definition 11.** *A random variable $\xi \sim \mathcal{D}$ follows a **heavy-tailed** distribution if the $\alpha$-moment is infinite for $\alpha \geq 2$.*

Intuitively, this expresses that the $\alpha$-moment is not sparsely supported outside a compact interval. That is, $\int_{\|\xi\| > R} \|\xi\|^\alpha p(\xi) \, d\xi < \infty$ indicates a dense support integrating to infinity in the closed ball $\mathcal{B}_0(R)$, and a light tail for $\mathcal{B}_0(R)^c$. Definition 1 enforces that the noise must not decay rapidly outside said compact ball, i.e. that light tails must be excluded. This follows from the observation that $\int_{\|\xi\| > R} \|\xi\|^\alpha p(\xi) \, d\xi = \infty$ for all $\alpha \geq 2$ and any $R \geq 0$ because $\int_{\|\xi\| \leq R} \|\xi\|^\alpha p(\xi) \, d\xi \leq R^\alpha < \infty$ via continuity and the extremal value theorem. By equivalence of norms on $\mathbb{R}^d$ and hence their preserved continuity, we analogously have for $\|\cdot\|_\infty$ the supremum norm,

$$\int_{\|\xi\|_\infty > R} c^\alpha \|\xi\|_2^\alpha \, p(\xi) \, d\xi \geq \int_{\|\xi\|_\infty > R} \|\xi\|_\infty^\alpha \, p(\xi) \, d\xi = \infty$$

for some $c > 0$. To proceed with the analysis, we impose an integrability condition on the mean, which gives $\mathbb{E}[\xi] = \mu \in \mathbb{R}^d$.

**Problem Setup.** The local objectives are determined by $F_i(x) = \mathbb{E}_z[F_i(x, z)]$, where $z$ integrates over the randomness in the stochastic objective. The gradient noise $\xi$ is additively modeled via a possibly uncentered random variable with $\mathbb{E}(\xi) = \mu$. Minibatches are sampled with replacement, implying that gradient noise in each client epoch are independent amongst and in between any two possibly identical clients. We analyze the case where noise is identically distributed conditional on client ID $i$. The global objective is given as the expected client objective under the uniform sampling prior, $f(x) = \sum_{i \in [N]} F_i(x)/N$.

We now present the following definition.

**Definition 12.** *A learning algorithm $\mathcal{A}$ is **deeply remorseful** if it incurs infinite or undefined regret in expectation. If $\mathcal{A}$ is guaranteed to instantly incur such regret due to sampling even a single client with a heavy-tailed stochastic gradient distribution, then we say $\mathcal{A}$ is **resentful** of heavy-tailed noise.*

We are now ready to prove the following theorem.

**Theorem 13.** *Let the global objectives $f_t(x)$ of a distributed training problem satisfy $\mu$-strong convexity for $t = 1, \ldots, T$. Assume that the participation probability of a client with a heavy-tailed stochastic gradient distribution is non-zero. Then, FedAvg becomes a deeply remorseful algorithm and is resentful of heavy-tailed noise. Furthermore, if the probability of the heavy-tailed client being sampled at step $t$ is nontrivial, then the variance of the global objective at $t + 1$ satisfies $\mathbb{E}\|f_{t+1}(x_{t+1})\|^2 = \infty$.*

*Proof.* Assuming that a heavy-tailed client may be subsampled at step $t$ with non-zero probability, let us show that the regret

$$R(T) := \sum_{t=1}^{T} f_t(x_t) - \sum_{t=1}^{T} f_t(x^*)$$

is infinite under expectation, assuming it is well-defined. Here, $x^*$ is taken to be the argument uniformly minimizing the materialized global objectives up to step $T$, $x^* := \arg\min_x \sum_{t=1}^{T} f_t(x)$. For notational simplicity, we carry out the analysis conditioned on the event that the heavy-tailed client has been subsampled. We aim to show that $\mathbb{E}[f_{t+1}(x_{t+1})] - f_{t+1}(x^*) = \infty$ where $x^*$ is arbitrarily fixed and $f_{t+1}$ satisfies $\mu$-strong convexity. Clearly,

$$f_{t+1}(x_{t+1}) \geq f_{t+1}(x_t) - \left\langle \nabla f_{t+1}(x_t), \frac{\eta_\ell}{|\mathcal{S}^t|} \sum_{i \in \mathcal{S}^t} \sum_{\ell=1}^{K} g_{i,\ell}^t \right\rangle + \frac{\mu \eta_\ell^2}{2|\mathcal{S}^t|^2} \left\| \sum_{i \in \mathcal{S}^t} \sum_{\ell=1}^{K} g_{i,\ell}^t \right\|^2$$

$$\geq f_{t+1}(x_t) - \left\langle \nabla f_{t+1}(x_t), \frac{\eta_\ell}{|\mathcal{S}^t|} \sum_{i \in \mathcal{S}^t} \sum_{\ell=1}^{K} \left( \nabla f(x_{i,\ell-1}^t) + \xi_{i,\ell-1}^t \right) \right\rangle$$

$$+ \frac{\mu \eta_\ell^2}{2|\mathcal{S}^t|^2} \left\| \sum_{i \in \mathcal{S}^t} \sum_{\ell=1}^{K} \left( \nabla f(x_{i,\ell-1}^t) + \xi_{i,\ell-1}^t \right) \right\|^2.$$

Denoting $\mathbb{E}_{t^+}[\cdot]$ to be the expectation conditional over all stochastic realizations up to step $t$ and $\ell = K - 1$, we have

$$\mathbb{E}_{t^+}[f_{t+1}(x_{t+1})] \geq f_{t+1}(x_t) - \left\langle \nabla f_{t+1}(x_t), \frac{\eta_\ell}{|\mathcal{S}^t|} \sum_{i \in \mathcal{S}^t} \left( \left( \sum_{\ell=1}^{K-1} \nabla f(x_{i,\ell-1}^t) + \xi_{i,\ell-1}^t \right) + \nabla f(x_{i,K-1}^t) \right) \right\rangle$$

$$- \left\langle \nabla f_{t+1}(x_t), \frac{\eta_\ell}{|\mathcal{S}^t|} \sum_{i \in \mathcal{S}^t} \mathbb{E}_{t^+} \left[ \xi_{i,K-1}^t \right] \right\rangle + \frac{\mu \eta_\ell^2}{2|\mathcal{S}^t|^2} \mathbb{E}_{t^+} \left\| \sum_{i \in \mathcal{S}^t} \sum_{\ell=1}^{K} \left( \nabla f(x_{i,\ell-1}^t) + \xi_{i,\ell-1}^t \right) \right\|^2. \quad (6)$$

As the means of all gradient noise are finite (typically centered at 0), it suffices to show that

$$\mathbb{E}_{t^+} \left\| \sum_{i \in \mathcal{S}^t} \sum_{\ell=1}^{K} \left( \nabla f(x_{i,\ell-1}^t) + \xi_{i,\ell-1}^t \right) \right\|^2 = \infty.$$

However, this is clear as expanding the norm gives

$$
\mathbb{E}_{t^+}\left\|\sum_{i\in\mathcal{S}^t}\sum_{\ell=1}^{K}\left(\nabla f(x_{i,\ell-1}^t)+\xi_{i,\ell-1}^t\right)\right\|^2 = \left\|\sum_{i\in\mathcal{S}^t}\sum_{\ell=1}^{K-1}\left(\nabla f(x_{i,\ell-1}^t)+\xi_{i,\ell-1}^t\right)+\sum_{i\in\mathcal{S}^t}\nabla f(x_{i,K-1}^t)\right\|^2
$$

$$
+2\left\langle\sum_{i\in\mathcal{S}^t}\sum_{\ell=1}^{K-1}\left(\nabla f(x_{i,\ell-1}^t)+\xi_{i,\ell-1}^t\right)+\sum_{i\in\mathcal{S}^t}\nabla f(x_{i,K-1}^t),\sum_{i\in\mathcal{S}^t}\mathbb{E}_{t^+}[\xi_{i,K-1}^t]\right\rangle+\sum_{i\in\mathcal{S}^t}\mathbb{E}\|\xi_{i,K-1}^t\|^2,
$$

where in the final line we used the independence of the noise random variables. As there exists $i\in\mathcal{S}^t$ that satisfies heavy-tailed noise, we obtain

$$
\mathbb{E}_{t^+}[f_{t+1}(x_{t+1})]\geq\infty.
$$

Taking expectations on both sides gives that $\mathbb{E}[f_{t+1}(x_{t+1})]\geq\infty$ under the law of iterated expectations, assuming that the expectation is well-defined. Thus, FedAvg is deeply resentful of the influence of heavy-tailed noise.

Now, we change perspectives and write the general form of (6) as

$$
f_{t+1}(y)\geq f_{t+1}(x)+\langle\nabla f_{t+1}(x),y-x\rangle+\frac{\mu}{2}\|y-x\|^2
$$

$$
= f_{t+1}(x)+\sum_{j=1}^{d}(\nabla f_{t+1}(x))_j(y_j-x_j)+\frac{\mu}{2}\sum_{j=1}^{d}(y_j-x_j)^2.
$$

For any arbitrarily fixed $x$, there exists $\tilde{a}_{t+1,j}>0$, $R_j>0$, and $\tilde{b}_{t+1,j}<0$ such that

$$
\tilde{f}_{t+1,j}(y_j)=\begin{cases}\tilde{a}_{t+1,j}(y_j-R_j) & \text{for} \quad y_j>R_j,\\ 0 & \text{for} \quad |y_j|\leq R_j,\\ \tilde{b}_{t+1,j}(y_j+R_j) & \text{for} \quad y_j<-R_j,\end{cases}\tag{7}
$$

and

$$
0\leq\tilde{f}_{t+1,j}(y_j)\leq\frac{f_{t+1}(x)}{d}+(\nabla f_{t+1}(x))_j(y_j-x_j)+\frac{\mu}{2}(y_j-x_j)^2
$$

for $|y_j|>R_j$. Without loss of generality, we may substitute $\tilde{a}_{t+1,j}\leftarrow\tilde{a}=\min_j\tilde{a}_{t+1,j}$, $\tilde{b}_{t+1,j}\leftarrow\tilde{b}=\max_j\tilde{b}_{t+1,j}$, and $R_j\leftarrow R:=\max_{j\in[d]}R_j$. We thus have

$$
\mathbb{E}_{t^+}[\|f_{t+1}(x_{t+1})\|^2]\geq\mathbb{E}_{t^+}\left[\chi\{x_{t+1}\in B_R^\infty(0)^c\}\|f_{t+1}(x_{t+1})\|^2\right]
$$

where $\chi$ is the indicator and $B_R^\infty(0)$ is the closed ball in $\mathbb{R}^d$ under the infinity norm centered at 0. As $f_{t+1}(y)\geq\sum_{j=1}^{d}\tilde{f}_{t+1,j}(y_j)$ for $y\in B_R^\infty(0)^c$,

$$
\mathbb{E}_{t^+}[\|f_{t+1}(x_{t+1})\|^2]\geq\mathbb{E}_{t^+}[\chi\{x_{t+1}\in B_R^\infty(0)^c\}\|\sum_{j=1}^{d}\tilde{f}_{t+1,j}(x_{t+1})\|^2]
$$

$$
\geq\mathbb{E}_{t^+}[\chi\{x_{t+1}\in B_R^\infty(0)^c\}\|\sum_{j=1}^{d}\tilde{f}_{t+1,j}\left(\frac{\eta_\ell}{|\mathcal{S}^t|}\sum_{i\in\mathcal{S}^t}\sum_{\ell=1}^{K}\left(\nabla f(x_{i,\ell-1}^t)_j+(\xi_{i,\ell-1}^t)_j\right)\right)\|^2].
$$

The integrand on the final line is non-negatively lower bounded given $x_{t+1} \in B_R^\infty(0)^c$ by

$$\left( c \sum_{j=1}^d \left| \frac{\eta_\ell}{|\mathcal{S}^t|} \sum_{i \in \mathcal{S}^t} \left( \left( \sum_{\ell=1}^{K-1} \nabla f(x_{i,\ell-1}^t) + \xi_{i,\ell-1}^t \right) + \nabla f(x_{i,K-1}^t) \right)_j + \frac{\eta_\ell}{|\mathcal{S}^t|} \sum_{i \in \mathcal{S}^t} (\xi_{i,K-1}^t)_j \pm R_j \right| \right)^2$$

$$\geq \sum_{j=1}^d c^2 \left| \frac{\eta_\ell}{|\mathcal{S}^t|} \sum_{i \in \mathcal{S}^t} \left( \left( \sum_{\ell=1}^{K-1} \nabla f(x_{i,\ell-1}^t) + \xi_{i,\ell-1}^t \right) + \nabla f(x_{i,K-1}^t) \right)_j + \frac{\eta_\ell}{|\mathcal{S}^t|} \sum_{i \in \mathcal{S}^t} (\xi_{i,K-1}^t)_j \pm R_j \right|^2$$

$$\geq \sum_{j=1}^d c^2 \left| \frac{\eta_\ell}{|\mathcal{S}^t|} \sum_{i \in \mathcal{S}^t} \left( \left( \sum_{\ell=1}^{K-1} \nabla f(x_{i,\ell-1}^t) + \xi_{i,\ell-1}^t \right) + \nabla f(x_{i,K-1}^t) \right)_j \pm R_j \right|^2$$

$$+ 2 \sum_{j=1}^d c^2 \left\langle \frac{\eta_\ell}{|\mathcal{S}^t|} \sum_{i \in \mathcal{S}^t} \left( \left( \sum_{\ell=1}^{K-1} \nabla f(x_{i,\ell-1}^t) + \xi_{i,\ell-1}^t \right) + \nabla f(x_{i,K-1}^t) \right)_j \pm R_j, \frac{\eta_\ell}{|\mathcal{S}^t|} \sum_{i \in \mathcal{S}^t} (\xi_{i,K-1}^t)_j \right\rangle$$

$$+ \sum_{j=1}^d \frac{c^2 \eta_\ell^2}{|\mathcal{S}^t|^2} \left( \sum_{i \in \mathcal{S}^t} (\xi_{i,K-1}^t)_j \right)^2$$

where $c = \min\{|\tilde{a}|, |\tilde{b}|\}$. The sign on $R_j$ is determined by the sign of the value $(x_{t+1})_j$ and equation (7).

Clearly, there exists compact intervals $[\bar{a}_{i,j}, \bar{b}_{i,j}]$ such that with non-zero probability, $(\xi_{i,K-1}^t)_j \in [\bar{a}_{i,j}, \bar{b}_{i,j}]$. For the setminus operation subtracting only one selection of client $i$ from the multiset $\mathcal{S}^t$ and $1 \in S^t$ being the heavy-tailed client, let $\hat{R}$ be equal to

$$\frac{|\mathcal{S}^t|}{\eta_\ell} \left( |R| + \max_{i,j} \left( \frac{\eta_\ell \max\{|\bar{a}_{i,j}|, |\bar{b}_{i,j}|\}}{|\mathcal{S}^t|} + \left| \frac{\eta_\ell}{|\mathcal{S}^t|} \sum_{\tilde{i} \in \mathcal{S}^t} \left( \left( \sum_{\ell=1}^{K-1} \nabla f(x_{i,\ell-1}^t) + \xi_{i,\ell-1}^t \right) + \nabla f(x_{i,K-1}^t) \right)_j \right| \right) \right).$$

Then as

$$\chi\{x_{t+1} \in B_R^\infty(0)^c\} \geq \chi\{x_{t+1} \in B_R^\infty(0)^c\} \Pi_{i \in S^t \setminus \{1\}} \chi\{(\xi_{i,K-1}^t)_j \in [\bar{a}_{i,j}, \bar{b}_{i,j}]\}$$

$$\geq \chi_j^+ := \chi\{|(\xi_{1,K-1}^t)_j| > \hat{R}\} \Pi_{i \in S^t \setminus \{1\}} \chi\{(\xi_{i,K-1}^t)_j \in [\bar{a}_{i,j}, \bar{b}_{i,j}]\},$$

we may conclude

$$\mathbb{E}_{t+}[\|f_{t+1}(x_{t+1})\|^2] \geq \mathbb{E}_{t+}\left[ \chi_j^+ \| \sum_{j=1}^d \tilde{f}_{t+1,j} \left( \frac{\eta_\ell}{|\mathcal{S}^t|} \sum_{i \in \mathcal{S}^t} \sum_{\ell=1}^K \left( \nabla f(x_{i,\ell-1}^t)_j + (\xi_{i,\ell-1}^t)_j \right) \right) \|^2 \right]$$

$$\geq \sum_{j=1}^d c^2 \mathbb{E}_{t+}[\chi_j^+] \left| \frac{\eta_\ell}{|\mathcal{S}^t|} \sum_{i \in \mathcal{S}^t} \left( \left( \sum_{\ell=1}^{K-1} \nabla f(x_{i,\ell-1}^t) + \xi_{i,\ell-1}^t \right) + \nabla f(x_{i,K-1}^t) \right)_j \pm R_j \right|^2$$

$$+ 2 \sum_{j=1}^d c^2 \left\langle \frac{\eta_\ell}{|\mathcal{S}^t|} \sum_{i \in \mathcal{S}^t} \left( \left( \sum_{\ell=1}^{K-1} \nabla f(x_{i,\ell-1}^t) + \xi_{i,\ell-1}^t \right) + \nabla f(x_{i,K-1}^t) \right)_j \pm R_j, \frac{\eta_\ell}{|\mathcal{S}^t|} \sum_{i \in \mathcal{S}^t} \mathbb{E}_{t+}[\chi_j^+ (\xi_{i,K-1}^t)_j] \right\rangle$$

$$+ \sum_{j=1}^d \frac{c^2 \eta_\ell^2}{|\mathcal{S}^t|^2} \mathbb{E}_{t+}\left[ \left( \sum_{i \in \mathcal{S}^t} (\xi_{i,K-1}^t)_j \right)^2 \right]$$

$$\geq C_1(t^+) + \sum_{j=1}^d \frac{c^2 \eta_\ell^2}{|\mathcal{S}^t|^2} \mathbb{E}_{t+}\left[ \chi_j^+ \left( \sum_{i \in \mathcal{S}^t} (\xi_{i,K-1}^t)_j \right)^2 \right]$$

Noting that

$$\mathbb{E}_{t+}[\chi_j^+ (\xi_{i,K-1}^t)_j] = \int_{\bar{a}_{i,j}}^{\bar{b}_{i,j}} (\xi_{i,K-1}^t)_j \, dp(\xi_{i,K-1}^t),$$

we deduce that the existence of $\mathbb{E}(\xi_{i,K-1}^t)_j \in \mathbb{R}$ (from all noise having finite mean) enforces that $\mathbb{E}_{t+}[\chi_j^+(\xi_{i,K-1}^t)_j]$ must also exist and be finite. Thus, $C_1(t^+)$ is finite and well-defined given $t^+$. It remains to analyze the final term

$$\sum_{j=1}^{d} \mathbb{E}_{t+}\left[\chi_j^+\left(\sum_{i\in\mathcal{S}^t}(\xi_{i,K-1}^t)_j\right)^2\right] = \sum_{j=1}^{d}\mathbb{E}_{t+}\left[\chi_j^+\sum_{i\in\mathcal{S}^t}(\xi_{i,K-1}^t)_j^2\right] + 2\mathbb{E}_{t+}\left[\chi_j^+\sum_{i_1<i_2}(\xi_{i_1,K-1}^t)_j(\xi_{i_2,K-1}^t)_j\right]$$

$$= \sum_{j=1}^{d}\sum_{i\in\mathcal{S}^t}\mathbb{E}_{t+}\left[\chi_j^+(\xi_{i,K-1}^t)_j^2\right] + 2\sum_{i_1<i_2}\mathbb{E}_{t+}\left[\chi_j^+(\xi_{i_1,K-1}^t)_j\right]\mathbb{E}_{t+}\left[\chi_j^+(\xi_{i_2,K-1}^t)_j\right]$$

where we used the independence of $\xi_{i,\ell}^t$ which is preserved across coordinate projections. Finally, note that for $C_2 := \min_{j\in[d]}\Pi_{i\in S^t\setminus\{1\}}\mathbb{P}((\xi_{i,K-1}^t)_j \in [\bar{a}_{i,j}, \bar{b}_{i,j}]) \neq 0$, we have

$$\sum_{j=1}^{d}\sum_{i\in\mathcal{S}^t}\mathbb{E}_{t+}\left[\chi_j^+(\xi_{i,K-1}^t)_j^2\right] \geq C_2\sum_{j=1}^{d}\int_{|(\xi_{1,K-1}^t)_j|>\hat{R}}\|(\xi_{1,K-1}^t)_j\|^2\,\mathrm{d}p(\xi_{1,K-1}^t)$$

$$\geq C_2\int_{\|(\xi_{1,K-1}^t)\|_\infty>\hat{R}}\|\xi_{1,K-1}^t\|^2\,\mathrm{d}p(\xi_{1,K-1}^t) = \infty.$$

Thus, we have as before

$$\mathbb{E}_{t+}[\|f_{t+1}(x_{t+1})\|^2] \geq \infty.$$

As the variance is well-defined, we conclude that $\mathbb{E}[\|f_{t+1}(x_{t+1})\|^2] = \infty$ under the tower law of expectation. □

For federated learning, we typically have $f_t(x) \equiv f(x)$ for all $t = 1, \ldots, T$. We saw from Proposition 9 that inserting local adaptivity successfully breaks the generality of remorse and heavy-tailed resent for FedAvg. A high-level, intuitive overview is that client-side AdaGrad clips the local updates of each iteration, which mollifies the impact of stochasticity in perturbing the weight updates. We present the following proposition, formulated loosely without utilizing any advantages provided via local adaptivity except for clipping which leaves room for far sharper generalization. For this reason, we view local adaptive methods to be more desirable than clipped SGD in large-scale applications, if memory and computation constraints of the clients can be addressed.

**Proposition 14.** *Let $f_t \in C(\mathbb{R}^d)$ for $t = 1, \ldots, T$ for $f_t$ not necessarily convex. Introducing client-side adaptivity via AdaGrad into the setting in Theorem 4 produces a non-remorseful and a non-resentful algorithm.*

*Proof.* By Jensen, we have that $\|\mathbb{E}f(x_t)\| \leq \mathbb{E}\|f(x_t)\|$. Thus, it is enough to show $\mathbb{E}\|f(x_t)\| < \infty$ which guarantees that the $t$-th regret update $\mathbb{E}[f_t(x_t)] - f_t(x^*)$ is finite for any $x^*$ arbitrarily fixed. However, this is immediate as $x_t \in B_{Kt}(x_0)$, where $K$ is the number of local iterations prior to server synchronization. Thus, by the extremal value theorem, there exists an $M \in \mathbb{R}_{\geq 0}$ such that

$$0 \leq \mathbb{E}\|f(x_t)\| \leq \mathbb{E}[M] < \infty.$$

Similarly, we may also show that the variance $\mathbb{E}\|f(x_t)\|^2 < \infty$. □

## B   DETAILED FEDADA$^2$ ALGORITHM DESCRIPTION

In the main text, we have opted to describe the intuitions behind SM3, due to its technical implementation. In this appendix section, we give a more through walk-through of our algorithm details for any interested readers wishing to reproduce our proof strategies or implementations.

**Addressing Client-Side Resource Constraints.**   In this paper, we specifically focus on SM3 (Anil et al., 2019) adaptations of Adam and Adagrad. Intuitively, SM3 exploits natural activation patterns observed in model gradients to accumulate approximate parameter-wise statistics for preconditioning. More precisely, the gradient information in each coordinate element $\{1, \ldots, d\}$ is blanketed by a cover $\{S_1, \ldots, S_q\}$ satisfying $\bigcup_{b=1}^{q} S_b = \{1, \ldots, d\}$ for which an auxiliary $\mu_k(b)$ is assigned

---

**Algorithm 2** Adaptive server and client-side ADAGRAD with SM3 ($\texttt{FedAda}^2$)

---

**Require:** A full cover $\{S_1, \ldots, S_q\} \subset \mathcal{P}([d])$ where $\bigcup_{b=1}^q S_b = \{1, \ldots, d\}$
    Update delay step size $z \in \mathbb{Z}_{\geq 1}$, initializations $x_0, \widetilde{v}_0 \geq \tau^2$ and $\widetilde{m}_0 \leftarrow 0$
    Local epsilon smoothing terms $\varepsilon_s, \varepsilon > 0$, global smoothing term $\tau > 0$
    Global decay parameter $\widetilde{\beta}_1 \in [0, 1)$
1: **for** $t = 1, \ldots, T$ **do**
2:    Sample subset $\mathcal{S}^t \subset [N]$ of clients using any sampling scheme
3:    **for** each client $i \in \mathcal{S}^t$ (in parallel) **do**
4:       Initialize $v_0 \geq 0$ (default value $v_0 \leftarrow 0$), $x_{i,0}^t \leftarrow x_{t-1}$
5:       **for** $k = 1, \ldots, K$ **do**
6:         Draw stochastic gradient $g_{i,k}^t \sim \mathcal{D}_{i,\mathrm{grad}}(x_{i,k-1}^t)$ with mean $\nabla F_i(x_{i,k-1}^t) \in \mathbb{R}^d$
7:         $m_k \leftarrow g_{i,k}^t$, $\mu_k(b) \leftarrow 0$ for $\forall b \in \{1, \ldots, q\}$
8:         **for** $j = 1, \ldots, d$ **do**
9:           Approximate Preconditioner (SM3)
10:        **end for**
11:       **if** $0 < \|m_k / (\sqrt{v_k} + \varepsilon)\| < \varepsilon_s$, **do** $m_k \leftarrow 0$
12:       $x_{i,k}^t \leftarrow x_{i,k-1}^t - \eta_\ell \cdot m_k / (\sqrt{v_k} + \varepsilon)$
13:       **end for**
14:       $\Delta_i^t = x_{i,K}^t - x_{t-1}$
15:    **end for**
16:    Server Update (SU)
17: **end for**

---

for each $b \in [q]$. The $\mu_k(b)$ then act to form $v_k$ as a coordinate ascent upper bound to the squared gradient sum $\sum_{\ell=1}^k (g_{i,\ell}^t)^2$ as SM3 iterates over each $j \in [d]$.

As an optional add-on, utilizing the staleness of gradients to construct preconditioners has previously been suggested as a strategy to accelerate adaptive optimization without hurting the performance (Gupta et al., 2018; Li et al., 2023). Therefore, we may optionally further mollify the burden of client-side adaptive optimizers by enforcing delayed preconditioner updates (Appendix I.2). This is given by the following SM3 update rule (SM3) which incorporates delay step $z$,

$$\text{SM3 Update:} \begin{cases} v_k(j) \leftarrow \min_{b:S_b \ni j} \mu_{k-1}(b) + \left(g_{i,k}^t(j)\right)^2 & \text{for } \frac{k-1}{z} \in \mathbb{Z} \\ \mu_k(b) \leftarrow \max\{\mu_k(b), v_k(j)\}, \text{for } \forall b : S_b \ni j & \\ v_k(j) \leftarrow v_{k-1}(j) & \text{otherwise} \end{cases} \tag{SM3}$$

where $k$ is the index of local iteration (starting from 1). These methodologies are consolidated into $\texttt{FedAda}^2$, Algorithm 2. For simplicity, we describe the variant in which both the client and server employ AdaGrad as the adaptive optimizers. However, we present other instantiations of $\texttt{FedAda}^2$ with different adaptive methods in Appendix D and I.1.

We now present a description of SM3-I/II with delayed preconditioner updates as Algorithms 3 and 4. SM3-II capitalizes on a tighter approximation of the second moment, and empirically demonstrates better results. We have opted to implement a smoothing term $\varepsilon$ instead of treating any zero denominator as zero as done in the original work. In this paper, we provide the analysis for SM3-II which generalizes the analysis for SM3-I.

## C   DETAILED PROOFS

To enhance clarity, we present several lemmas before giving the proof of Theorem 20. Note that Lemma 15 is written in broadcasting notation, where the scalars in the right hand side have $\mathbf{1} \in \mathbb{R}^d$ implicitly multiplied and the inequality holds coordinatewise. For notational convenience, we will view $\Phi_1^K, \Phi_2^K$ as vectors.

---

**Algorithm 3** Delayed preconditioner SM3-I

---

**Require:** Client learning rate $\eta_\ell$, step delay $z \in \mathbb{Z}_{\geq 1}$, and $\varepsilon$-smoothing term $\varepsilon > 0$

**Require:** A full cover $\{S_1, \ldots, S_k\} \subset \mathcal{P}([d])$ where $\bigcup_{\ell=1}^k S_\ell = \{1, \ldots, d\}$

  1: **Initialize:** $x_1 = 0$ and $\mu_0(r) = 0$ for $\forall r \in \{1, \ldots, k\}$

  2: **for** $t = 1, \ldots, K$ **do**

  3:    $g_t \leftarrow \nabla \ell(x_t)$

  4:    **if** $(t-1)/z \in \mathbb{Z}$ **then**

  5:      **for** $r = 1, \ldots, k$ **do**

  6:        $\mu_t(r) \leftarrow \mu_{t-1}(r) + \max_{j \in S_r} g_t^2(j)$

  7:      **end for**

  8:    **end if**

  9:    **for** $j = 1, \ldots, d$ **do**

10:      $\nu_t(j) \leftarrow \min_{r:S_r \ni j} \mu_t(r)$ (minimum taken over all $r$ such that $j \in S_r$)

11:      $x_{t+1}(j) \leftarrow x_t(j) - \frac{\eta_\ell g_t(j)}{\sqrt{\nu_t(j)+\varepsilon}}$

12:    **end for**

13: **end for**

---

**Algorithm 4** Delayed preconditioner SM3-II

---

**Require:** Client learning rate $\eta_\ell$, step delay $z \in \mathbb{Z}_{\geq 1}$, and $\varepsilon$-smoothing term $\varepsilon > 0$

**Require:** A full cover $\{S_1, \ldots, S_k\} \subset \mathcal{P}([d])$ where $\bigcup_{\ell=1}^k S_\ell = \{1, \ldots, d\}$

  1: **Initialize:** $x_1 = 0$ and $\mu'_0(r) = 0$ for $\forall r \in \{1, \ldots, k\}$

  2: **for** $t = 1, \ldots, K$ **do**

  3:    $g_t \leftarrow \nabla \ell(x_t)$

  4:    $\mu'_t(r) \leftarrow 0$ for $\forall r \in [k]$

  5:    **for** $j = 1, \ldots, d$ **do**

  6:      **if** $(t-1)/z \in \mathbb{Z}$ **then**

  7:        $\nu'_t(j) \leftarrow \min_{r:S_r \ni j} \mu'_{t-1}(r) + g_t^2(j)$

  8:        **for all** $r : S_r \ni j$ **do**

  9:          set $\mu'_t(r) \leftarrow \max\{\mu'_t(r), \nu'_t(j)\}$

10:        **end for**

11:      **else**

12:        $\nu'_t(j) \leftarrow \nu'_{t-1}(j)$

13:      **end if**

14:      $x_{t+1}(j) \leftarrow x_t(j) - \frac{\eta_\ell g_t(j)}{\sqrt{\nu'_t(j)+\varepsilon}}$

15:    **end for**

16: **end for**

---

**Lemma 15.** *Under Algorithm 2, $|\Delta_i^t|$ is bounded by*

$$|\Delta_i^t| \leq \Phi_1^K := \eta_\ell \left( \sqrt{\left\lceil \frac{K}{z} \right\rceil} \cdot \log^{\frac{1}{2}} \left( 1 + \frac{\left\lceil \frac{K}{z} \right\rceil G^2}{\varepsilon^2} \right) + \frac{\eta_\ell (K - \left\lceil \frac{K}{z} \right\rceil) G}{\sqrt{v_0} + \varepsilon} \right).$$

*Proof.* Forming a bound for the pseudogradients is not trivial due to delayed preconditioner updates. We begin by noting that delayed gradient updates are initiated at local timesteps $k = nz + 1$ for $n \in \mathbb{Z}_{\geq 0}$. We now split cases $k/z \notin \mathbb{Z}$ and $k/z \in \mathbb{Z}$. In the first case, there exists $n \in \mathbb{Z}_{\geq 0}$ such that $nz + 1 \leq k < (n+1)z$, and the latest preconditioner update by client step $k$ is given at timestep $(\lceil k/z \rceil - 1)z + 1 = \lfloor k/z \rfloor z + 1$. In the second case, if $z \neq 1$, then step $k$ is just one step shy of a preconditioner update. The latest update is therefore held at step $(\lceil k/z \rceil - 1)z + 1$ which is no longer identical to $\lfloor k/z \rfloor z + 1$.

With this observation, it is easy to show by induction that

$$v_k(j) \geq v_0(j) + \sum_{\ell=1}^{\lceil \frac{k}{z} \rceil} \left( g_{i,(\ell-1)z+1}^t(j) \right)^2 \quad \text{for} \quad j \in \{1, \ldots, d\} \quad \text{and} \quad k \in \{1, \ldots, K\}.$$

Recall that $\Delta_t = 1/|\mathcal{S}^t| \sum_{i \in \mathcal{S}^t} \Delta_i^t$ and $\Delta_i^t = x_{i,K}^t - x_{i,0}^t$. By telescoping for $K$ local steps and the definition of gradient updates in AdaSquare-SM3, we obtain

$$|\Delta_i^t| = \left| \sum_{p=1}^K \eta_\ell \frac{m_p}{\sqrt{v_p} + \varepsilon} \right| \leq \eta_\ell \sum_{p=1}^K \frac{|g_{i,p}^t|}{\sqrt{v_0 + \sum_{r=1}^{\lceil \frac{p}{z} \rceil} (g_{i,(r-1)z+1}^t)^2} + \varepsilon}$$

For $\mathcal{F} = \{0, 1, \ldots, \lceil K/z \rceil - 1\}z + 1$, we thus have that

$$|\Delta_i^t| \leq \eta_\ell \sum_{p \in \mathcal{F}} \frac{|g_{i,p}^t|}{\sqrt{v_0 + \sum_{r=1}^{\lceil \frac{p}{z} \rceil} (g_{i,(r-1)z+1}^t)^2} + \varepsilon}$$

$$+ \eta_\ell \sum_{p \in [K] \setminus \mathcal{F}} \frac{|g_{i,p}^t|}{\sqrt{v_0 + \sum_{r=1}^{\lceil \frac{p}{z} \rceil} (g_{i,(r-1)z+1}^t)^2} + \varepsilon}.$$

To obtain a deterministic bound, we cannot ignore the worst-case stochastic realization that $g_{i,(r-1)z+1}^t = 0$ for $\forall r \in [\lceil \frac{p}{z} \rceil], p \in [K] \setminus \mathcal{F}$. Therefore, we form the upper bound (where $\sum_1^0 := 0$ by definition)

$$|\Delta_i^t| \leq \eta_\ell \underbrace{\sum_{p \in \mathcal{F}} \frac{|g_{i,p}^t|}{\sqrt{v_0 + |g_{i,p}^t|^2 + \sum_{r=1}^{\lceil \frac{p}{z} \rceil - 1} (g_{i,(r-1)z+1}^t)^2} + \varepsilon}}_{T_1} + \frac{\eta_\ell}{\sqrt{v_0} + \varepsilon} \left( \sum_{p \in [K] \setminus \mathcal{F}} |g_{i,p}^t| \right) \quad (8)$$

$$\leq \eta_\ell T_1 + \frac{\eta_\ell (K - \lceil \frac{K}{z} \rceil) G}{\sqrt{v_0} + \varepsilon}.$$

As $0$ is trivially bounded by any non-negative upper bound, we may without loss of generality assume that $g_{i,(r-1)z+1}^t \neq 0$ for at least one $r \in [\lceil \frac{p}{z} \rceil]$. We further bound $T_1$ as follows:

$$T_1 \leq \sum_{p \in \mathcal{F}} \frac{|g_{i,p}^t|}{\sqrt{|g_{i,p}^t|^2 + \sum_{r=1}^{\lceil \frac{p}{z} \rceil - 1} (g_{i,(r-1)z+1}^t)^2} + \varepsilon} \leq \sum_{p \in \mathcal{F}} \sqrt{\frac{|g_{i,p}^t|^2}{\varepsilon^2 + \sum_{r \in [p] \cap \mathcal{F}} |g_{i,r}^t|^2}}$$

$$\leq \sqrt{|\mathcal{F}|} \sqrt{\left( \sum_{p \in \mathcal{F}} \frac{|g_{i,p}^t|^2}{\varepsilon^2 + \sum_{r \in [p] \cap \mathcal{F}} |g_{i,r}^t|^2} \right)}$$

$$\leq \sqrt{\left\lceil \frac{K}{z} \right\rceil} \cdot \log^{\frac{1}{2}} \left( 1 + \sum_{p \in \mathcal{F}} \frac{|g_{i,p}^t|^2}{\varepsilon^2} \right)$$

Note the use of Cauchy Schwartz in the third inequality. A detailed proof of the log inequality used in the third line may be found as part of the proof of Theorem 20, equation (13) which uses similar techniques. By Assumption 2, we are done. □

The server-side pseudogradient updates may also be bounded as follows.

**Lemma 16.** *Under Algorithm 2, each server step size is bounded in absolute value by*

$$\Phi_2^K := \min\left\{\eta\sqrt{(1-\widetilde{\beta}_1)(1-\widetilde{\beta}_1^{2t})}, \frac{\eta}{\tau}\Phi_1^K\right\}.$$

*Proof.* Without loss of generality, we may let $\tau = 0$ when forming the first upper bound for expository purposes.

$$\eta\frac{|\widetilde{m}_t|}{\sqrt{\widetilde{v}_t}+\tau} \leq \frac{\eta(1-\widetilde{\beta}_1)\sum_{\ell=1}^t \widetilde{\beta}_1^{t-\ell}|\Delta_\ell|}{\sqrt{\sum_{\ell=1}^t \Delta_\ell^2 + \tau^2}+\tau}$$

$$\leq \frac{\eta(1-\widetilde{\beta}_1)\left(\sum_{\ell=1}^t \widetilde{\beta}_1^{t-\ell}|\Delta_\ell|\right)\sqrt{\sum_{\ell=1}^t \widetilde{\beta}_1^{2t-2\ell}}}{\sqrt{\sum_{\ell=1}^t \Delta_\ell^2}\sqrt{\sum_{\ell=1}^t \widetilde{\beta}_1^{2t-2\ell}}}$$

$$\leq \eta\sqrt{1-\widetilde{\beta}_1}\sqrt{1-\widetilde{\beta}_1^2}\sqrt{\sum_{\ell=1}^t \widetilde{\beta}_1^{2t-2\ell}}$$

$$= \eta\sqrt{1-\widetilde{\beta}_1}\sqrt{1-\widetilde{\beta}_1^{2t}}.$$

Note that the final inequality is obtained using Cauchy-Schwartz, while the second bound in the lemma statement follows from the first inequality and Lemma 15. □

Finally, we form a loose upper bound for the gradient variance.

**Lemma 17.** *For $k \in \{1, \ldots, K\}$, the uncentered variance estimate $v_k$ as well as $\mu_k$ in Algorithm 2 are bounded by*

$$(B1): \quad 0 \leq \mu_k(b) \leq dkG^2 \quad for \quad and \quad b \in \{1, \ldots, q\},$$
$$(B2): \quad 0 \leq v_k(j) \leq dkG^2 \quad for \quad j \in \{1, \ldots, d\}.$$

*Proof.* Non-negativity of the variance estimates $v_k$ is trivial and implies the non-negativity of $\mu_k$, thus we focus on the upper bound for which we use dual induction. The case $k = 1$ is satisfied by zero initialization. Assuming the inequality holds for $k \leftarrow k-1$, we have for each $j$

$$v_k(j) = \min_{b:S_b\ni j}\mu_{k-1}(b) + \left(g_{i,k}^t(j)\right)^2 \leq d(k-1)G^2 + G^2 \leq dkG^2.$$

Now, $\mu_k$ is initialized to zero at the start of each step $k$ and its entries are increased while broadcasting over each coordinate $j \in \{1, \ldots, d\}$ by

$$\mu_k(b) \leftarrow \max\{\mu_k(b), v_k(j)\} \quad for \quad \forall b : j \in S_b.$$

For $j = 1$, it is clear that

$$\mu_k(b) \leftarrow v_k(j) \leq dkG^2 \quad for \quad \forall b \in \{1, \ldots, q\}.$$

For $j \geq 2$, inductively, we have

$$\mu_k(b) \leftarrow \max\{\mu_k(b), v_k(j)\} \leq dkG^2$$

as both arguments of the maximum function are upper bounded by $dkG^2$. This completes the proof.
□

## C.1 PRECOMPACT CONVERGENCE ANALYSIS

We aim to analyze the convergence of learning algorithms under the general, non-convex setting. However, extremely popular and well known adaptive optimizers such as Adam whose efficacy is strongly supported by empirical evidence have been shown to fail to converge even for convex settings (Reddi et al., 2018). Therefore, recent works have investigated the asymptotic stabilization of gradients, instead of requiring strict convergence to local or global optima of the objective (Reddi et al., 2021; Wang et al., 2022; Tong et al., 2020; Sun et al., 2023; Xie et al., 2019; Chen et al., 2020; Zhang et al., 2020). Such convergence bounds are of the form $\min_t \|\nabla f(x_t)\| \leq \mathcal{O}(T^{-\alpha})$, and are interpreted via the following lemma:

**Lemma 18.** *For $x_t$ the $t$-step parameters of any objective $f(x)$ learned by an algorithm, let $\min_{1 \leq t \leq T} \|\nabla f(x_t)\| \leq \mathcal{O}(T^{-\alpha})$ for $\alpha > 0$. Then, there exists a learning algorithm which outputs parameters $\{\widetilde{x}_1, \widetilde{x}_2, \ldots\}$ such that $\|\nabla f(\tilde{x}_t)\| \to 0$ as $t \to \infty$.*

*Proof.* Assuming otherwise gives that $\|\nabla f(x_t)\|$ is $\varepsilon$-bounded away from $0$ for some $\varepsilon > 0$, for any parameter $x_t$ realized by the algorithm. Clearly, $\min_{1 \leq t \leq T} \|\nabla F(x_t)\| \to 0$ as $T \to \infty$ gives a contradiction. More constructively, note that $\forall \varepsilon > 0$, $\exists \widetilde{T}(\varepsilon) \in \mathbb{N}$ such that $T \geq \widetilde{T}(\varepsilon) \implies \min_{1 \leq t \leq T} \|\nabla f(x_t)\| < \varepsilon$. Letting $\varepsilon = 1/n$ for $n \in \mathbb{N}$ and $T_n := \widetilde{T}(1/n)$, we have that there exists $t_n \in [T_n]$ such that $\|\nabla f(x_{t_n})\| < 1/n$. Letting $\widetilde{x}_i := x_{t_i}$ extracts the desired parameter sequence. $\square$

This notion of convergence can be formalized as *precompact convergence* which is consistent with sequence properties of precompact normed sets. In this paper, we explicitly formalize the conventions used in prior works, and take the term convergence to mean precompact convergence unless stated otherwise.

**Definition 19** (Precompact convergence). *A sequence $\{y_n\}_{n \in \mathbb{N}}$ in a normed space $\mathcal{Y}$ is said to converge precompactly to $y \in \mathcal{Y}$ if there exists $\varphi : \mathbb{N} \to \mathbb{N}$ such that $y_{\varphi(n)} \to y$.*

Our goal is to develop principled federated algorithms whose global gradients are guaranteed to converge precompactly to $0$ regardless of parameter initialization, in the general, non-convex setting. Note that precompact convergence must allow for convergence to each element $y_n$ of the sequence. Now, we are ready to present the following theorem.

**Theorem 20.** *In Algorithm 2, we have that*

$$\min_{t \in [T]} \|\nabla f(x_{t-1})\|^2 \leq \frac{\Psi_1 + \Psi_2 + \Psi_3 + \Psi_4 + \Psi_5}{\Psi_6},$$

*where*

$$\Psi_1 = f(x_0) - f(x^*),$$

$$\Psi_2 = \frac{\eta^2 L T d \|\Phi_1^K\|^2}{\tau^2},$$

$$\Psi_3 = \frac{(1 - \widetilde{\beta}_1^T) \eta \eta_\ell K \widetilde{L} T \|\Phi_1^K\|^2}{\widetilde{\alpha}_1 \tau (\sqrt{v_0} + \varepsilon)^2},$$

$$\Psi_4 = \frac{(1 - \widetilde{\beta}_1) \eta \eta_\ell K L T c(\widetilde{\beta}_1) \|\Phi_2^K\|^2}{\widetilde{\alpha}_1 \tau (\sqrt{v_0} + \varepsilon)^2},$$

$$\Psi_5 = \frac{\eta d \|\Phi_1^K\| G \left(1 - \widetilde{\beta}_1 + \log\left(1 + \frac{T \|\Phi_1^K\|^2}{\tau^2}\right)\right)}{\tau},$$

$$\Psi_6 = \frac{3(1 - \widetilde{\beta}_1) \eta \widetilde{\gamma}_1 T}{4 \left(\sqrt{T \|\Phi_1^K\|^2 + \widetilde{v}_0} + \tau\right)}.$$

*Here, the constant $c$ is defined with respect to $\widetilde{\beta}_1$ as*

$$c(\widetilde{\beta}_1) := \sum_{u=0}^{\widetilde{u}_0(\widetilde{\beta}_1)} \widetilde{\beta}_1^u u^2 + \int_{\widetilde{u}_0(\widetilde{\beta}_1)}^{\infty} \frac{1}{x^2} \mathrm{d}x \quad \text{for} \quad \widetilde{u}_0(\widetilde{\beta}_1) = \inf\{u \in \mathbb{N} : \widetilde{\beta}_1^v v^2 < \frac{1}{v^2} \text{ for } \forall v \geq u\}$$

*and the intermediary $\widetilde{\gamma}_1, \widetilde{\alpha}_1$ values are defined as*

$$\widetilde{\gamma}_1 := \eta_\ell \frac{K}{\sqrt{v_0 + dKG^2 + \varepsilon}}, \quad \widetilde{\alpha}_1 := \frac{1}{2\sqrt{v_0 + dKG^2 + 2\varepsilon}}.$$

*Proof.* To enhance readability, we use both coordinatewise and broadcasting notation, where a $[\cdot]_j$ subscript is attached for the $j$-th coordinate. In particular, the arguments are detailed mostly in the latter notation as it significantly clarifies the intuitions behind the proof. By $L$-smoothness, we have

$$f(x_t) \leq f(x_{t-1}) + \langle \nabla f(x_{t-1}), x_t - x_{t-1} \rangle + \frac{L}{2} \|x_t - x_{t-1}\|^2$$

$$= f(x_{t-1}) + \eta \left\langle \nabla f(x_{t-1}), \frac{\widetilde{\beta}_1^t \widetilde{m}_0 + (1 - \widetilde{\beta}_1) \sum_{r=1}^t \widetilde{\beta}_1^{t-r} \Delta_r}{\sqrt{\widetilde{v}_t} + \tau} \right\rangle + \frac{\eta^2 L}{2} \left\| \frac{\widetilde{\beta}_1^t \widetilde{m}_0 + (1 - \widetilde{\beta}_1) \sum_{r=1}^t \widetilde{\beta}_1^{t-r} \Delta_r}{\sqrt{\widetilde{v}_t} + \tau} \right\|^2$$

$$= f(x_{t-1}) + \eta T_{0,0} + (1 - \widetilde{\beta}_1) \eta \sum_{r=1}^t T_{0,r} + \frac{\eta^2 L}{2} \left\| \frac{\widetilde{\beta}_1^t \widetilde{m}_0 + (1 - \widetilde{\beta}_1) \sum_{r=1}^t \widetilde{\beta}_1^{t-r} \Delta_r}{\sqrt{\widetilde{v}_t} + \tau} \right\|^2 \quad (9)$$

where for $r \in [t]$,

$$T_{0,r} = \widetilde{\beta}_1^{t-r} \left\langle \nabla f(x_{t-1}), \frac{\Delta_r}{\sqrt{\widetilde{v}_t} + \tau} \right\rangle \quad \text{and} \quad T_{0,0} = \left\langle \nabla f(x_{t-1}), \frac{\widetilde{\beta}_1^t \widetilde{m}_0}{\sqrt{\widetilde{v}_t} + \tau} \right\rangle. \quad (10)$$

Note that $T_{0,0}$ can only decay exponentially as training progresses, as $\sqrt{\widetilde{v}_t}$ is monotonically increasing with respect to $t$ and $\nabla f(x_{t-1})$ is coordinatewise bounded by $G$. We decompose $T_{0,r}$ further by

$$T_{0,r} = \underbrace{\widetilde{\beta}_1^{t-r} \left\langle \nabla f(x_{t-1}), \frac{\Delta_r}{\sqrt{\widetilde{v}_t} + \tau} - \frac{\Delta_r}{\sqrt{\widetilde{v}_{t-1}} + \tau} \right\rangle}_{T_{1,r}} + \underbrace{\widetilde{\beta}_1^{t-r} \left\langle \nabla f(x_{t-1}), \frac{\Delta_r}{\sqrt{\widetilde{v}_{t-1}} + \tau} \right\rangle}_{T_{2,r}}.$$

A bound for $T_{1,r}$ can be obtained as:

$$T_{1,r} = \widetilde{\beta}_1^{t-r} \left\langle \nabla f(x_{t-1}), \frac{\Delta_r (\sqrt{\widetilde{v}_{t-1}} - \sqrt{\widetilde{v}_t})}{(\sqrt{\widetilde{v}_t} + \tau)(\sqrt{\widetilde{v}_{t-1}} + \tau)} \right\rangle$$

$$= \widetilde{\beta}_1^{t-r} \left\langle \nabla f(x_{t-1}), \frac{-\Delta_r \Delta_t^2}{(\sqrt{\widetilde{v}_t} + \tau)(\sqrt{\widetilde{v}_{t-1}} + \tau)(\sqrt{\widetilde{v}_{t-1}} + \sqrt{\widetilde{v}_t})} \right\rangle$$

$$\leq \widetilde{\beta}_1^{t-r} \left\langle |\nabla f(x_{t-1})|, \frac{|\Delta_r| \Delta_t^2}{(\widetilde{v}_t + \tau^2)(\sqrt{\widetilde{v}_{t-1}} + \tau)} \right\rangle$$

$$\leq \widetilde{\beta}_1^{t-r} \sum_{j=1}^d G \left[ \frac{|\Delta_r| \Delta_t^2}{(\widetilde{v}_t + \tau^2)(\sqrt{\widetilde{v}_{t-1}} + \tau)} \right]_j$$

$$\leq \frac{\|\Phi_1^K\| G \widetilde{\beta}_1^{t-r}}{\tau} \sum_{j=1}^d \left[ \frac{\Delta_t^2}{\widetilde{v}_t} \right]_j.$$

Lemma 30 is used to obtain the final inequality. For $T_{2,r}$, we apply a further decomposition for $\gamma_r > 0$ allowed to be arbitrary within a compact interval $\epsilon \eta_\ell$-bounded away from 0,

$$T_{2,r} = \underbrace{\widetilde{\beta}_1^{t-r} \left\langle \frac{\nabla f(x_{t-1})}{\sqrt{\widetilde{v}_{t-1}} + \tau}, \Delta_r + \gamma_r \nabla f(x_{t-1}) \right\rangle}_{T_{2,r}^1} - \gamma_r \widetilde{\beta}_1^{t-r} \left\| \frac{\nabla f(x_{t-1})}{\sqrt{\sqrt{\widetilde{v}_{t-1}} + \tau}} \right\|^2.$$

For expository purposes, we present the case in which local gradient clipping is not triggered. The analysis directly generalizes to the setting where clipping activates. Unraveling the definition of $\Delta_r$

gives

$$\Delta_r = \frac{-\eta_\ell}{|\mathcal{S}^r|} \sum_{i \in \mathcal{S}^r} \sum_{p=1}^{K} \frac{g_{i,p}^r}{\sqrt{v_{i,p}^r} + \varepsilon},$$

which intuits the following value

$$\gamma_r := \frac{\eta_\ell}{|\mathcal{S}^r|} \sum_{i \in \mathcal{S}^r} \sum_{p=1}^{K} \frac{1}{\sqrt{v_{i,p}^r} + \varepsilon}.$$

We have by Assumption 2 and Lemma 17 that

$$\gamma_r \in [\widetilde{\gamma}_1, \widetilde{\gamma}_2] := \left[ \eta_\ell \sum_{p=1}^{K} \frac{1}{\sqrt{v_0 + dKG^2} + \varepsilon}, \frac{\eta_\ell K}{\sqrt{v_0} + \varepsilon} \right].$$

Expanding $T_{2,r}^1$ for $\alpha_r > 0$ to be fixed,

$$\widetilde{\beta}_1^{t-r} \left\langle \frac{\nabla f(x_{t-1})}{\sqrt{\widetilde{v}_{t-1}} + \tau}, \Delta_r + \gamma_r \nabla f(x_{t-1}) \right\rangle$$

$$= \frac{\widetilde{\beta}_1^{t-r}}{|\mathcal{S}^r|} \sum_{i \in \mathcal{S}^r} \sum_{p=1}^{K} \left\langle \frac{\nabla f(x_{t-1})}{\sqrt{\widetilde{v}_{t-1}} + \tau}, \frac{\eta_\ell \left( \nabla f(x_{t-1}) - g_{i,p}^r \right)}{\sqrt{v_p} + \varepsilon} \right\rangle$$

$$\leq \frac{\eta_\ell \widetilde{\beta}_1^{t-r} \alpha_r K}{2|\mathcal{S}^r|} \sum_{i \in \mathcal{S}^r} \left\| \frac{\nabla f(x_{t-1})}{\sqrt{\sqrt{\widetilde{v}_{t-1}} + \tau}} \right\|^2$$

$$+ \frac{\eta_\ell \widetilde{\beta}_1^{t-r}}{2|\mathcal{S}^r| \alpha_r} \sum_{i \in \mathcal{S}^r} \sum_{p=1}^{K} \left\| \frac{\left( \nabla f(x_{t-1}) - \nabla F_i(x_{i,p-1}^r) \right)}{\sqrt{\sqrt{\widetilde{v}_{t-1}} + \tau} \left( \sqrt{v_p} + \varepsilon \right)} \right\|^2$$

$$\leq \frac{\eta_\ell \widetilde{\beta}_1^{t-r} \alpha_r K}{2} \left\| \frac{\nabla f(x_{t-1})}{\sqrt{\sqrt{\widetilde{v}_{t-1}} + \tau}} \right\|^2$$

$$+ \frac{\eta_\ell \widetilde{\beta}_1^{t-r}}{2|\mathcal{S}^r| \alpha_r \tau (\sqrt{v_0} + \varepsilon)^2} \sum_{i \in \mathcal{S}^r} \sum_{p=1}^{K} \left\| \nabla f(x_{t-1}) - \nabla F_i(x_{i,p-1}^r) \right\|^2.$$

where in the first inequality we drew the deterministic gradient instead of accessing the stochastic sample via full gradient descent. The first term is controlled by setting

$$\alpha_r = \frac{\gamma_r}{2\eta_\ell K} \in [\widetilde{\alpha}_1, \widetilde{\alpha}_2] := \left[ \frac{1}{2\sqrt{v_0 + dKG^2} + 2\varepsilon}, \frac{1}{2\sqrt{v_0} + 2\varepsilon} \right].$$

We aim to bound the second term via majorization and telescoping arguments. We have by $L$-smoothness, Lemmas 15, 16, and Assumption 2 that

$$
\begin{aligned}
\left\|\nabla f(x_{t-1}) - \nabla F_i(x_{i,p-1}^r)\right\|^2 &\leq \frac{1}{N} \sum_{i' \in [N]} \left\|\left(\nabla F_{i'}(x_{t-1}) - \nabla F_i(x_{i,p-1}^r)\right)\right\|^2 \\
&= \frac{1}{N} \sum_{i' \in [N]} \left\|\left(\nabla F_{i'}(x_{t-1}) - \nabla F_{i'}(x_{r-1}) + \nabla F_{i'}(x_{r-1}) - \nabla F_i(x_{i,p-1}^r)\right)\right\|^2 \\
&\leq \frac{2}{N} \sum_{i' \in [N]} \left(\|\nabla F_{i'}(x_{t-1}) - \nabla F_{i'}(x_{r-1})\|^2 + \left\|\nabla F_{i'}(x_{r-1}) - \nabla F_i(x_{i,p-1}^r)\right\|^2\right) \\
&\leq \frac{2L}{N} \sum_{i' \in [N]} \|x_{t-1} - x_{r-1}\|^2 + \frac{2\widetilde{L}}{N} \sum_{i' \in [N]} \|x_{i,p-1}^r - x_{i,0}^r\|^2 \\
&= 2L \|x_{t-1} - x_{r-1}\|^2 + 2\widetilde{L} \left\|x_{i,p-1}^r - x_{i,0}^r\right\|^2 \\
&\leq 2L(t-r) \sum_{o=r}^{t-1} \|x_o - x_{o-1}\|^2 + 2\widetilde{L}\|\Phi_1^p\|^2 \\
&\leq 2L(t-r)^2\|\Phi_2^K\|^2 + 2\widetilde{L}\|\Phi_1^K\|^2.
\end{aligned}
$$

Note that the first inequality was obtained by Jensen, while the third inequality uses that the client weights $x_{i,0}^r$ are synchronized to the global weights $x_{r-1}$ for $\forall i \in [N]$ at the start of training. Now, we have

$$
\begin{aligned}
&\frac{\eta_\ell \widetilde{\beta}_1^{t-r}}{2|\mathcal{S}^r|\alpha_r\tau(\sqrt{v_0} + \varepsilon)^2} \sum_{i \in \mathcal{S}^r} \sum_{p=1}^K \left(2L(t-r)^2\|\Phi_2^K\|^2 + 2\widetilde{L}\|\Phi_1^K\|^2\right) \\
&\leq \frac{\eta_\ell \widetilde{\beta}_1^{t-r} K L(t-r)^2\|\Phi_2^K\|^2}{\alpha_r\tau(\sqrt{v_0} + \varepsilon)^2} + \frac{\eta_\ell \widetilde{\beta}_1^{t-r} \widetilde{L}K\|\Phi_1^K\|^2}{\alpha_r\tau(\sqrt{v_0} + \varepsilon)^2}.
\end{aligned}
$$

Collecting terms gathered thus far gives

$$
\begin{aligned}
(1 - \widetilde{\beta}_1)\eta \sum_{r=1}^t T_{0,r} &\leq (1 - \widetilde{\beta}_1)\eta \sum_{r=1}^t \left(\frac{\|\Phi_1^K\|G\widetilde{\beta}_1^{t-r}}{\tau} \sum_{j=1}^d \left[\frac{\Delta_t^2}{\widetilde{v}_t}\right]_j - \frac{3\gamma_r\widetilde{\beta}_1^{t-r}}{4} \left\|\frac{\nabla f(x_{t-1})}{\sqrt{\sqrt{\widetilde{v}_{t-1}} + \tau}}\right\|^2\right) \\
&\quad + (1 - \widetilde{\beta}_1)\eta \sum_{r=1}^t \left(\frac{\eta_\ell \widetilde{\beta}_1^{t-r} K L(t-r)^2\|\Phi_2^K\|^2}{\alpha_r\tau(\sqrt{v_0} + \varepsilon)^2} + \frac{\eta_\ell \widetilde{\beta}_1^{t-r} \widetilde{L}K\|\Phi_1^K\|^2}{\alpha_r\tau(\sqrt{v_0} + \varepsilon)^2}\right).
\end{aligned}
$$

Now, let us bound the final term in equation (9),

$$
\begin{aligned}
\left\|\frac{\widetilde{\beta}_1^t \widetilde{m}_0 + (1 - \widetilde{\beta}_1) \sum_{r=1}^t \widetilde{\beta}_1^{t-r}\Delta_r}{\sqrt{\widetilde{v}_t} + \tau}\right\|^2 &\leq 2\left\|\frac{\widetilde{\beta}_1^t \widetilde{m}_0}{\sqrt{\widetilde{v}_t} + \tau}\right\|^2 + 2\left\|\frac{(1 - \widetilde{\beta}_1) \sum_{r=1}^t \widetilde{\beta}_1^{t-r}\Delta_r}{\sqrt{\widetilde{v}_t} + \tau}\right\|^2 \\
&\leq 2\left\|\frac{\widetilde{\beta}_1^t \widetilde{m}_0}{\sqrt{\widetilde{v}_t} + \tau}\right\|^2 + 2\left\|\frac{(1 - \widetilde{\beta}_1) \sum_{r=1}^t \widetilde{\beta}_1^{t-r} \max_{r \in [t]} |\Delta_r|}{\sqrt{\widetilde{v}_t} + \tau}\right\|^2 \\
&\leq 2\left\|\frac{\widetilde{\beta}_1^t \widetilde{m}_0}{\sqrt{\widetilde{v}_t} + \tau}\right\|^2 + 2\left\|\frac{(1 - \widetilde{\beta}_1^t)}{\sqrt{\widetilde{v}_t} + \tau}\right\|^2 \|\Phi_1^K\|^2 \\
&\leq 2\left\|\frac{\widetilde{\beta}_1^t \widetilde{m}_0}{\sqrt{\widetilde{v}_t} + \tau}\right\|^2 + 2d\frac{\|\Phi_1^K\|^2}{\tau^2}.
\end{aligned}
$$

Substituting into equation (9) gives that

$$f(x_t) \leq f(x_{t-1}) + \eta T_{0,0} + \eta^2 L \left\| \frac{\widetilde{\beta}_1^t \widetilde{m}_0}{\sqrt{\widetilde{v}_t} + \tau} \right\|^2 + \frac{\eta^2 L d \|\Phi_1^K\|^2}{\tau^2} + (1 - \widetilde{\beta}_1)\eta \sum_{r=1}^{t} \left( \frac{\|\Phi_1^K\| \|G \widetilde{\beta}_1^{t-r}}{\tau} \sum_{j=1}^{d} \left[ \frac{\Delta_t^2}{\widetilde{v}_t} \right]_j \right)$$

$$+ (1 - \widetilde{\beta}_1)\eta \sum_{r=1}^{t} \left( \frac{\eta_\ell \widetilde{\beta}_1^{t-r} K L (t-r)^2 \|\Phi_2^K\|^2}{\alpha_r \tau (\sqrt{v_0} + \varepsilon)^2} + \frac{\eta_\ell \widetilde{\beta}_1^{t-r} \widetilde{L} K \|\Phi_1^K\|^2}{\alpha_r \tau (\sqrt{v_0} + \varepsilon)^2} \right)$$

$$+ (1 - \widetilde{\beta}_1)\eta \sum_{r=1}^{t} \left( -\frac{3\gamma_r \widetilde{\beta}_1^{t-r}}{4} \left\| \frac{\nabla f(x_{t-1})}{\sqrt{\sqrt{\widetilde{v}_{t-1}} + \tau}} \right\|^2 \right). \tag{11}$$

Note that the exponential decay caused by $\widetilde{\beta}_1$ in the third term will expectedly dominate the effect of first order moment initialization $\widetilde{m}_0$ as training progresses, and summation over $t \in [T]$ gives $\mathcal{O}(1)$. We initialize $\widetilde{m}_0 \leftarrow 0$ to further simplify the equations. We also further exacerbate the upper bound by substituting $\widetilde{\gamma}_1, \widetilde{\alpha}_1$ into $\gamma_r, \alpha_r$ respectively, which achieves independence from $r$. Telescoping equation (11) then gives

$$\frac{3(1 - \widetilde{\beta}_1)\eta \widetilde{\gamma}_1}{4} \sum_{t=1}^{T} \sum_{r=1}^{t} \widetilde{\beta}_1^{t-r} \left\| \frac{\nabla f(x_{t-1})}{\sqrt{\sqrt{\widetilde{v}_{t-1}} + \tau}} \right\|^2 \leq f(x_0) - f(x^*) + \frac{(1 - \widetilde{\beta}_1)\eta \|\Phi_1^K\| G}{\tau} \sum_{t=1}^{T} \sum_{r=1}^{t} \sum_{j=1}^{d} \widetilde{\beta}_1^{t-r} \left[ \frac{\Delta_t^2}{\widetilde{v}_t} \right]_j$$

$$+ \frac{\eta^2 L T d \|\Phi_1^K\|^2}{\tau^2} + \frac{(1 - \widetilde{\beta}_1)\eta \eta_\ell K}{\widetilde{\alpha}_1 \tau (\sqrt{v_0} + \varepsilon)^2} \sum_{t=1}^{T} \sum_{r=1}^{t} \left( L \widetilde{\beta}_1^{t-r} (t-r)^2 \|\Phi_2^K\|^2 + \widetilde{L} \widetilde{\beta}_1^{t-r} \|\Phi_1^K\|^2 \right). \tag{12}$$

To complete the proof, we aim to ease a logarithm out from the third term on the right hand side. For this purpose, we induce a recursion with a log bound

$$(1 - \widetilde{\beta}_1) \sum_{t=1}^{T} \sum_{r=1}^{t} \widetilde{\beta}_1^{t-r} \frac{\Delta_{t,j}^2}{\sum_{\ell=1}^{t} \Delta_{\ell,j}^2 + \tau^2} \leq \sum_{t=1}^{T} (1 - \widetilde{\beta}_1^t) \frac{\Delta_{t,j}^2}{\sum_{\ell=1}^{t} \Delta_{\ell,j}^2 + \tau^2}$$

$$\leq a_T + c_T \log(1 + b_T). \tag{13}$$

Setting $T = 1$ gives

$$(1 - \widetilde{\beta}_1) \frac{\Delta_{1,j}^2}{\Delta_{1,j}^2 + \tau^2} \leq a_1 + c_1 \log(1 + b_1),$$

and setting $a_T = 1 - \widetilde{\beta}_1$ satisfies this inequality (among other choices). Assuming formula (13) holds for $T$, let us explore the induction condition for $T + 1$, which is

$$\sum_{t=1}^{T} (1 - \widetilde{\beta}_1^t) \frac{\Delta_{t,j}^2}{\sum_{\ell=1}^{t} \Delta_{\ell,j}^2 + \tau^2} + (1 - \widetilde{\beta}_1^{T+1}) \frac{\Delta_{T+1,j}^2}{\sum_{\ell=1}^{T+1} \Delta_{\ell,j}^2 + \tau^2} \leq a_{T+1} + c_{T+1} \log(1 + b_{T+1}).$$

For simplicity, we impose that $c_t$ is a monotonically increasing non-negative sequence of $t$. We intend to contain the increase in the left hand side as $T$ grows in the log argument only, in the right hand side. Therefore, we select $a_{T+1} = a_T$. For a suitable choice of $b_{T+1}$ satisfying strong induction, it is enough to resolve

$$(1 - \widetilde{\beta}_1^{T+1}) \frac{\Delta_{T+1,j}^2}{\sum_{\ell=1}^{T+1} \Delta_{\ell,j}^2 + \tau^2} \leq c_{T+1} \log\left( \frac{1 + b_{T+1}}{1 + b_T} \right) = c_{T+1} \log\left( 1 + \frac{b_{T+1} - b_T}{1 + b_T} \right).$$

Here, we used monotonicity of $c_t$. Noting that $\log(1 + x) \geq x/(1+x)$, it is again enough to resolve

$$\frac{\Delta_{T+1,j}^2}{\sum_{\ell=1}^{T+1} \Delta_{\ell,j}^2 + \tau^2} \leq \frac{c_{T+1}(b_{T+1} - b_T)}{b_{T+1} + 1}$$

$$\iff \frac{\Delta_{T+1,j}^2}{\sum_{\ell=1}^{T+1} \Delta_{\ell,j}^2 + \tau^2} + c_{T+1} b_T \leq \left( c_{T+1} - \frac{\Delta_{T+1,j}^2}{\sum_{\ell=1}^{T+1} \Delta_{\ell,j}^2 + \tau^2} \right) b_{T+1}.$$

By positivity of $b_t$ for $t > 1$, a necessary condition is therefore that

$$c_{T+1} \geq \frac{\Delta_{T+1,j}^2}{\sum_{\ell=1}^{T+1} \Delta_{\ell,j}^2 + \tau^2}$$

In order to enhance the tightness of our bound, we choose the minimal permissible value $c_t = 1$ uniformly, which is attained as a suprema. In this setting, we are left with a recursion

$$\frac{\Delta_{T+1,j}^2}{\sum_{\ell=1}^{T+1} \Delta_{\ell,j}^2 + \tau^2} = \frac{b_{T+1} - b_T}{b_{T+1} + 1},$$

and collecting the terms in the form $b_{T+1} = b_T \omega_1(\Delta) + \omega_2(\Delta)$ would provide an optimal recursive bound given our simplifying assumptions, starting with $b_1 = 0$. A less optimal but simpler bound can be formed by selecting $b_{T+1} = b_T + \Delta_{T+1,j}^2/\tau^2$ for $b_1 = \Delta_{1,j}^2/\tau^2$. Therefore, we arrive at

$$(1 - \widetilde{\beta}_1) \sum_{t=1}^{T} \sum_{r=1}^{t} \widetilde{\beta}_1^{t-r} \frac{\Delta_{t,j}^2}{\sum_{\ell=1}^{t} \Delta_{\ell,j}^2 + \tau^2} \leq 1 - \widetilde{\beta}_1 + \log\left(1 + \sum_{\ell=1}^{T} \left(\frac{\Delta_{\ell,j}}{\tau}\right)^2\right)$$

$$\leq 1 - \widetilde{\beta}_1 + \log\left(1 + \frac{T\|\Phi_1^K\|^2}{\tau^2}\right). \tag{14}$$

The remaining term to be bounded in equation (12) is given

$$\frac{(1 - \widetilde{\beta}_1)\eta\eta_\ell KL}{\widetilde{\alpha}_1 \tau(\sqrt{v_0} + \varepsilon)^2} \sum_{t=1}^{T} \sum_{r=1}^{t} \left(\widetilde{\beta}_1^{t-r}(t-r)^2 \|\Phi_2^K\|^2\right).$$

The trick is to notice that the explosion of the series caused by double summation is culled selectively in reverse chronological order by the exponential, rendering the tail end asymptotically vacuous. Note that $(1 - \widetilde{\beta}_1)$ stabilizes the divergence as $\widetilde{\beta}_1 \to 1^-$ in the limit. By a change of variable $u = t - r$,

$$(1 - \widetilde{\beta}_1) \sum_{t=1}^{T} \sum_{r=1}^{t} \widetilde{\beta}_1^{t-r}(t-r)^2 = (1 - \widetilde{\beta}_1) \sum_{u=0}^{T-1} \widetilde{\beta}_1^u u^2 (T-u).$$

Defining

$$\widetilde{u}_0(\widetilde{\beta}_1) = \inf\{u \in \mathbb{N} : \widetilde{\beta}_1^v v^2 < \frac{1}{v^2} \text{ for } \forall v \geq u\},$$

let

$$c(\widetilde{\beta}_1) := \sum_{u=0}^{\widetilde{u}_0(\widetilde{\beta}_1)} \widetilde{\beta}_1^u u^2 + \int_{\widetilde{u}_0(\widetilde{\beta}_1)}^{\infty} \frac{1}{x^2} \mathrm{d}x.$$

Then, we claim that

$$(1 - \widetilde{\beta}_1) \sum_{t=1}^{T} \sum_{r=1}^{t} \widetilde{\beta}_1^{t-r}(t-r)^2 \leq (1 - \widetilde{\beta}_1)c(\widetilde{\beta}_1)T.$$

We prove this by induction. The case $T = 1$ is trivial. Now, assume the desired inequality holds until $T$. For $T + 1$, we want to show

$$(1 - \widetilde{\beta}_1) \sum_{u=0}^{T} \widetilde{\beta}_1^u u^2 (T - u + 1) \leq (1 - \widetilde{\beta}_1)c(\widetilde{\beta}_1)(T + 1)$$

$$\iff (1 - \widetilde{\beta}_1) \sum_{u=0}^{T-1} \widetilde{\beta}_1^u u^2 (T - u) + (1 - \widetilde{\beta}_1) \sum_{u=0}^{T} \widetilde{\beta}_1^u u^2 \leq (1 - \widetilde{\beta}_1)c(\widetilde{\beta}_1)(T + 1)$$

and thus by the inductive hypothesis it is enough to show

$$\sum_{u=0}^{T} \widetilde{\beta}_1^u u^2 \leq c(\widetilde{\beta}_1).$$

However, this is trivial by the definition of $c(\widetilde{\beta}_1)$. Upon substitution into equation (12) and noting that

$$\frac{3(1 - \widetilde{\beta}_1)\eta\widetilde{\gamma}_1}{4} \sum_{t=1}^{T} \sum_{r=1}^{t} \widetilde{\beta}_1^{t-r} \left\| \frac{\nabla f(x_{t-1})}{\sqrt{\sqrt{\widetilde{v}_{t-1}} + \tau}} \right\|^2 \geq \frac{3(1 - \widetilde{\beta}_1)\eta\widetilde{\gamma}_1 T}{4\left(\sqrt{T\|\Phi_1^K\|^2 + \widetilde{v}_0} + \tau\right)} \min_{t \in [T]} \|\nabla f(x_{t-1})\|^2$$

we simplify as

$$\frac{3(1 - \widetilde{\beta}_1)\eta\widetilde{\gamma}_1 T}{4\left(\sqrt{T\|\Phi_1^K\|^2 + \widetilde{v}_0} + \tau\right)} \min_{t \in [T]} \|\nabla f(x_{t-1})\|^2 \leq f(x_0) - f(x^*) + \frac{\eta^2 L T d\|\Phi_1^K\|^2}{\tau^2}$$

$$+ \frac{(1 - \widetilde{\beta}_1^T)\eta\eta_\ell K T \widetilde{L}\|\Phi_1^K\|^2}{\widetilde{\alpha}_1 \tau (v_0 + \varepsilon)^2} + \frac{(1 - \widetilde{\beta}_1)\eta\eta_\ell K T L c(\widetilde{\beta}_1)\|\Phi_2^K\|^2}{\widetilde{\alpha}_1 \tau (v_0 + \varepsilon)^2} \tag{15}$$

$$+ \frac{\eta d\|\Phi_1^K\| G \left(1 - \widetilde{\beta}_1 + \log\left(1 + \frac{T\|\Phi_1^K\|^2}{\tau^2}\right)\right)}{\tau}$$

Therefore, we immediately conclude that

$$\min_{t \in [T]} \|\nabla f(x_{t-1})\|^2 \leq \frac{\Psi_1 + \Psi_2 + \Psi_3 + \Psi_4 + \Psi_5}{\Psi_6},$$

where

$$\Psi_1 = f(x_0) - f(x^*),$$

$$\Psi_2 = \frac{\eta^2 L T d\|\Phi_1^K\|^2}{\tau^2},$$

$$\Psi_3 = \frac{(1 - \widetilde{\beta}_1^T)\eta\eta_\ell K \widetilde{L} T\|\Phi_1^K\|^2}{\widetilde{\alpha}_1 \tau (\sqrt{v_0} + \varepsilon)^2},$$

$$\Psi_4 = \frac{(1 - \widetilde{\beta}_1)\eta\eta_\ell K L T c(\widetilde{\beta}_1)\|\Phi_2^K\|^2}{\widetilde{\alpha}_1 \tau (\sqrt{v_0} + \varepsilon)^2},$$

$$\Psi_5 = \frac{\eta d\|\Phi_1^K\| G \left(1 - \widetilde{\beta}_1 + \log\left(1 + \frac{T\|\Phi_1^K\|^2}{\tau^2}\right)\right)}{\tau},$$

$$\Psi_6 = \frac{3(1 - \widetilde{\beta}_1)\eta\widetilde{\gamma}_1 T}{4\left(\sqrt{T\|\Phi_1^K\|^2 + \widetilde{v}_0} + \tau\right)}.$$

Here, the constant $c$ is defined with respect to $\widetilde{\beta}_1$ as

$$c(\widetilde{\beta}_1) := \sum_{u=0}^{\widetilde{u}_0(\widetilde{\beta}_1)} \widetilde{\beta}_1^u u^2 + \int_{\widetilde{u}_0(\widetilde{\beta}_1)}^{\infty} \frac{1}{x^2} \mathrm{d}x \quad \text{for} \quad \widetilde{u}_0(\widetilde{\beta}_1) = \inf\{u \in \mathbb{N} : \widetilde{\beta}_1^v v^2 < \frac{1}{v^2} \text{ for } \forall v \geq u\}$$

and the intermediary $\widetilde{\gamma}_1, \widetilde{\alpha}_1$ values are defined as

$$\widetilde{\gamma}_1 := \eta_\ell \frac{K}{\sqrt{v_0 + dKG^2} + \varepsilon}, \quad \widetilde{\alpha}_1 := \frac{1}{2\sqrt{v_0 + dKG^2} + 2\varepsilon}.$$

This concludes the proof. □

Note that we have also shown the following two useful lemmas:

**Lemma 21.** *For $\widetilde{\beta}_1 \in [0, 1)$ and $T \in \mathbb{Z}_{\geq 0}$, let*

$$\widetilde{u}_0(\widetilde{\beta}_1) = \inf\{u \in \mathbb{N} : \widetilde{\beta}_1^v v^2 < \frac{1}{v^2} \text{ for } \forall v \geq u\},$$

*and*

$$c(\widetilde{\beta}_1) := \sum_{u=0}^{\widetilde{u}_0(\widetilde{\beta}_1)} \widetilde{\beta}_1^u u^2 + \int_{\widetilde{u}_0(\widetilde{\beta}_1)}^{\infty} \frac{1}{x^2} \mathrm{d}x.$$

*Then, we have that*

$$\sum_{t=1}^{T}\sum_{r=1}^{t}\widetilde{\beta}_1^{t-r}(t-r)^2 \leq c(\widetilde{\beta}_1)T.$$

**Lemma 22.** *Let $\Delta_{\ell,j} \in \mathbb{R}$, $\widetilde{\beta}_1 \in [0,1)$, and $T \in \mathbb{Z}_{\geq 0}$. Then,*

$$(1 - \widetilde{\beta}_1)\sum_{t=1}^{T}\sum_{r=1}^{t}\widetilde{\beta}_1^{t-r}\frac{\Delta_{t,j}^2}{\sum_{\ell=1}^{t}\Delta_{\ell,j}^2 + \tau^2} \leq 1 - \widetilde{\beta}_1 + \log\left(1 + \frac{T\|\Phi_1^K\|^2}{\tau^2}\right).$$

We present the following corollary.

**Corollary 23.** *Any of the following conditions are sufficient to ensure convergence of Algorithm 2:*

$$(A): \quad \eta_\ell \leq \mathcal{O}(T^{-1/2}) \quad for \quad \Omega(T^{-1}) < \eta\eta_\ell < \mathcal{O}(1),$$
$$(B): \quad \eta_\ell = \Theta(T^{-\frac{49}{100}}) \quad for \quad \Omega(T^{-\frac{1}{2}}) < \eta < \mathcal{O}(T^{\frac{12}{25}}).$$

*Proof.* The proof is formed by comparing orders of $T$. Recall that $\widetilde{\gamma}_1 = \Theta(\eta_\ell)$ and $\widetilde{L} = \Theta(\eta_\ell^{-1})$. As $\Phi_1^K = \Theta(\eta_\ell)$ and $\Phi_2^K = \Theta(\min\{\eta, \eta\eta_\ell\})$, we have for $\eta = \Theta(T^{p_1})$ and $\eta_\ell = \Theta(T^{p_2})$,

$$\psi_1 = \Theta(1)$$
$$\psi_2 = \eta^2\eta_\ell^2 T$$
$$\psi_3 = \eta\eta_\ell^2 T$$
$$\psi_4 = \begin{cases} \eta^3\eta_\ell^3 T & \text{if } \mathcal{O}(\eta_\ell) \leq \mathcal{O}(1) \\ \eta^3\eta_\ell T & \text{if } \Theta(\eta_\ell) > \Omega(1) \end{cases}$$
$$\psi_5 = \eta\eta_\ell \log(1 + T\eta_\ell^2)$$
$$\psi_6 = \begin{cases} \eta\eta_\ell T & \text{if } \mathcal{O}(T\eta_\ell^2) \leq \mathcal{O}(1) \\ \eta\sqrt{T} & \text{if } \Theta(T\eta_\ell^2) > \Omega(1) \end{cases}.$$

If $\mathcal{O}(T\eta_\ell^2) \leq \mathcal{O}(1)$, then $\mathcal{O}(\eta_\ell) \leq \mathcal{O}(1)$ which implies

$$\frac{\psi_1}{\psi_6} : (\eta\eta_\ell T)^{-1} = \Theta\left(T^{-(p_1+p_2+1)}\right)$$

$$\frac{\psi_2}{\psi_6} : \eta\eta_\ell = \Theta\left(T^{p_1+p_2}\right)$$

$$\frac{\psi_3}{\psi_6} : \eta_\ell = \Theta\left(T^{p_2}\right)$$

$$\frac{\psi_4}{\psi_6} : \eta^2\eta_\ell^2 = \Theta\left(T^{2p_1+2p_2}\right)$$

$$\frac{\psi_5}{\psi_6} : \frac{\log(1 + T\eta_\ell^2)}{T} = \mathcal{O}(T^{-1})$$

This implies that we must have that $p_2 \leq -1/2$ and $-1 < p_1 + p_2 < 0$ for guaranteed convergence. Thus, $\eta_\ell \leq \mathcal{O}(T^{-1/2})$ such that $\Omega(T^{-1}) < \eta\eta_\ell < \mathcal{O}(1)$ is a sufficient condition. For instance, let $\eta_\ell = \Theta(T^{-1/2})$ and $\Omega(T^{-1/2}) < \eta < \mathcal{O}(T^{1/2})$.

Now, assume $\Theta(T\eta_\ell^2) > \Omega(1)$. If $\Theta(\eta_\ell) > \Omega(1)$, $\Psi_3/\Psi_6$ diverges. Therefore, let $\eta_\ell \le \mathcal{O}(1)$. We have

$$\frac{\psi_1}{\psi_6} : (\eta\sqrt{T})^{-1} = \Theta(T^{-p_1-\frac{1}{2}})$$

$$\frac{\psi_2}{\psi_6} : \eta\eta_\ell^2\sqrt{T} = \Theta(T^{p_1+2p_2+\frac{1}{2}})$$

$$\frac{\psi_3}{\psi_6} : \eta_\ell^2\sqrt{T} = \Theta(T^{2p_2+\frac{1}{2}})$$

$$\frac{\psi_4}{\psi_6} : \eta^2\eta_\ell^3\sqrt{T} = \Theta(T^{2p_1+3p_2+\frac{1}{2}})$$

$$\frac{\psi_5}{\psi_6} : \frac{\eta_\ell\log(1+T\eta_\ell^2)}{\sqrt{T}} < \mathcal{O}(T^{-\frac{1}{2}+p_2})$$

Therefore, it suffices to satisfy

$$-\frac{1}{2} < p_2 \le -\frac{1}{4}, \quad -\frac{1}{2} < p_1, \quad p_1 + 2p_2 < -\frac{1}{2}, \quad 2p_1 + 3p_2 < -\frac{1}{2}.$$

An example satisfying these conditions are

$$\eta_\ell = \Theta(T^{-\frac{49}{100}}), \quad \Omega(T^{-\frac{1}{2}}) < \eta < \mathcal{O}(T^{\frac{12}{25}}).$$

$\square$

Note that for all cases, $\eta_\ell$ must decay to establish convergence. However, striking a balance between local and global learning rates provably allows for greater than $\Omega(T^{1/3})$ divergence in the server learning rate without nullifying desirable convergence properties. This theoretically demonstrates the enhanced robustness properties of adaptive client-side federated learning algorithms to mitigate suboptimal choices of server learning rates.

**Corollary 24.** *Algorithm 2 converges at rate $\mathcal{O}(T^{-1/2})$.*

*Proof.* If $\mathcal{O}(T\eta_\ell^2) \le \mathcal{O}(1)$, then we juxtapose $\psi_1/\psi_6$ and $\psi_2/\psi_6$. It is clear that the minimax value of the respective powers are attained at $p_1 + p_2 = -1/2$, realized by $p_2 = -1/2$ and $p_1 = 0$. In this case, clearly $\Theta(\psi_i/\psi_6) \le \mathcal{O}(T^{-1/2})$ for $1 \le i \le 5$. If $\Theta(T\eta_\ell^2) > \Omega(1)$, then our strategy should be to minimize $p_2$ due to positive coefficients in the powers $\psi_i/\psi_6$. Thus, let $p_2 = -1/2 + \varepsilon$ for $1 \gg \varepsilon > 0$. Then, the order of decay in $\psi_2/\psi_6$ is $p_1 - 1/2 + 2\varepsilon$, which is once again matched against $-p_1 - 1/2$, the power of $\psi_1/\psi_6$. Taking the limit $\varepsilon \to 0^+$, minimax$\{p_1 - 1/2, -p_1 - 1/2\}$ for the range $-1/2 < p_1$ is attained at $p_1 = 0$. This sets the maximal decay rate to $\mathcal{O}(T^{-1/2})$ for the second case. $\square$

## C.2 EXTENSION TO ADAM

The extension to the case where Adam is selected as the optimizer for the server, or for both the server and client is straightforward. We present the latter as it generalizes the former analysis. As in Lemma 15, we have the following bound for the compressed SM3 estimates of the second moment,

$$v_k(j) \ge v_0(j) + \sum_{\ell=1}^{\lceil \frac{k}{z} \rceil} \left(g_{i,(\ell-1)z+1}^t(j)\right)^2 \quad \text{for} \quad j \in \{1, \dots, d\} \quad \text{and} \quad k \in \{1, \dots, K\},$$

which allows bounds to be established for the local and global pseudogradients following analogous logic as Lemmas 16, 28. As before, we arrive at equation (10) where due to exponential moving averaging on the server side, we have

$$\widetilde{v}_t = \widetilde{\beta}_2^t\widetilde{v}_0 + (1 - \widetilde{\beta}_2)\sum_{\ell=1}^t \widetilde{\beta}_2^{t-r}\Delta_\ell.$$

Now, decompose $T_{0,r}$ as

$$T_{0,r} = \underbrace{\widetilde{\beta}_1^{t-r} \left\langle \nabla f(x_{t-1}), \frac{\Delta_r}{\sqrt{\widetilde{v}_t} + \tau} - \frac{\Delta_r}{\sqrt{\widetilde{\beta}_2 \widetilde{v}_{t-1}} + \tau} \right\rangle}_{T_{1,r}} + \underbrace{\widetilde{\beta}_1^{t-r} \left\langle \nabla f(x_{t-1}), \frac{\Delta_r}{\sqrt{\widetilde{\beta}_2 \widetilde{v}_{t-1}} + \tau} \right\rangle}_{T_{2,r}},$$

where $T_{1,r}$ may be bounded via

$$T_{1,r} = \widetilde{\beta}_1^{t-r} \left\langle \nabla f(x_{t-1}), \frac{\Delta_r(\sqrt{\widetilde{\beta}_2 \widetilde{v}_{t-1}} - \sqrt{\widetilde{v}_t})}{(\sqrt{\widetilde{v}_t} + \tau)(\sqrt{\widetilde{\beta}_2 \widetilde{v}_{t-1}} + \tau)} \right\rangle$$

$$= \widetilde{\beta}_1^{t-r} \left\langle \nabla f(x_{t-1}), \frac{-\Delta_r \Delta_t^2(1 - \widetilde{\beta}_2)}{(\sqrt{\widetilde{v}_t} + \tau)(\sqrt{\widetilde{\beta}_2 \widetilde{v}_{t-1}} + \tau)(\sqrt{\widetilde{\beta}_2 \widetilde{v}_{t-1}} + \sqrt{\widetilde{v}_t})} \right\rangle$$

$$\leq \frac{\|\Phi_1^K\| G \widetilde{\beta}_1^{t-r}(1 - \widetilde{\beta}_2)}{\tau} \sum_{j=1}^{d} \left[ \frac{\Delta_t^2}{\widetilde{v}_t} \right]_j.$$

Due to the exponential decay parameter in the first pseudogradient moment, we have

$$\eta \sum_{t=1}^{T} \sum_{r=1}^{t} \frac{\|\Phi_1^K\| G \widetilde{\beta}_1^{t-r}(1 - \widetilde{\beta}_2)}{\tau} \sum_{j=1}^{d} \left[ \frac{\Delta_t^2}{\widetilde{v}_t} \right]_j \leq \eta \sum_{t=1}^{T} \sum_{r=1}^{t} \frac{\|\Phi_1^K\|^3 G \widetilde{\beta}_1^{t-r}(1 - \widetilde{\beta}_2)}{\tau^2}$$

$$\leq \frac{\eta \|\Phi_1^K\|^3 G T(1 - \widetilde{\beta}_2)}{\tau^2}.$$

An analogue of the arguments made in the proof of Theorem 6 with appropriate modifications, e.g.,

$$\gamma_r := \frac{\eta_\ell}{|\mathcal{S}^r|} \sum_{i \in \mathcal{S}^r} \sum_{p=1}^{K} \frac{(1 - \beta_1) \sum_{\ell=1}^{p} \beta_1^{p-\ell}}{\sqrt{(1 - \beta_2) \sum_{\ell=1}^{\lceil \frac{p}{z} \rceil} \beta_2^{\lceil \frac{p}{z} \rceil - \ell} (g_{i,(\ell-1)z+1}^r)^2} + \varepsilon},$$

gives the main change as the asymptotic behavior of $\Psi_5$, which now satisfies

$$\Psi_5 = \Theta\left( \eta \eta_\ell^3 T \right).$$

The convergence rate is still dominated by $\Psi_1$, $\Psi_2$ as in Corollary 24, which gives $\mathcal{O}(T^{-1/2})$.

## D  FEDERATED BLENDED OPTIMIZATION (GENERAL/FULL FORM OF FEDADA$^2$)

In federated blended optimization, we distribute local optimizer strategies during the subsampling process which may be formalized as functions that take as input the availability of client resources, and outputs the number of local epochs, $K(O_l^i)$, as well as additional hyperparameters such as delay step size $z$ or preconditioner initialization. These may be chosen to streamline model training based on a variety of factors, such as straggler mitigation or dynamically restricted availability of local resources.

In the general formulation of FedAda$^2$, blended optimization allows the trainer to utilize the unique strengths of each individual optimizer, balancing resource constraints and client noise. Each client has the option to run different optimizer strategies as the training rounds progress, depending on varying individual resource constraints or distribution shift in the local data stream. This faithfully corresponds to real-world settings where the availability of local resources are actively dynamic. Future work will provide empirical results on the performance of blended optimization, including identifying the settings in which mixing optimizer strategies are advantageous for distributed learning. The following theorem shows that under certain non-restrictive conditions, blended optimization still allows for convergence of the global gradient objective.

---

**Algorithm 5** Server-side ADAGRAD and client-side optimizer mixture ($\texttt{FedAda}^2$)

---

**Require:** Local optimizer strategies $O_1, \ldots, O_{Op}$ (e.g. Adam, AdaGrad, SGD...)
**Require:** Initializations $x_0, \widetilde{v}_0 \geq \tau^2$ and $\widetilde{m}_0 \leftarrow 0$
**Require:** Global decay parameter $\widetilde{\beta}_1 \in [0, 1)$
 1: **for** $t = 1, \ldots, T$ **do**
 2:     Sample participating client multiset $S_l^t$ for each optimizer strategy $l \in [Op]$
 3:     **for** each sampled client collection $l \in [Op]$ (in parallel) **do**
 4:         **for** each client $i \in S_l^t$ (in parallel) **do**
 5:             $x_{i,0}^{t,l} \leftarrow x_{t-1}$
 6:             $x_{i,K(O_l^i)}^{t,l} \leftarrow \text{Optimize}(O_l, i, x_{i,0}^{t,l}, Clip = \text{True})$
 7:             $\Delta_i^{t,l} = w(O_l)\left(x_{i,K(O_l^i)}^{t,l} - x_{t-1}\right)$
 8:         **end for**
 9:     **end for**
10:     $S \leftarrow \sum_{l \in [Op]} |S_l^t|$
11:     $\Delta_t = \frac{1}{S} \sum_{l \in [Op]} \sum_{i \in S_l^t} \Delta_i^{t,l}$
12:     $\widetilde{m}_t = \widetilde{\beta}_1 \widetilde{m}_{t-1} + (1 - \widetilde{\beta}_1)\Delta_t$
13:     $\widetilde{v}_t = \widetilde{v}_{t-1} + \Delta_t^2$
14:     $x_t = x_{t-1} + \eta \frac{\widetilde{m}_t}{\sqrt{\widetilde{v}_t} + \tau}$
15: **end for**

---

**Theorem 25.** *Given client $i \in [N]$, strategy $l \in [Op]$, global timestep $r$, and local timestep $p$, assume that the optimizer strategies satisfy the parameter update rule*

$$x_{i,p}^{r,l} = x_{i,p-1}^{r,l} - \eta_\ell \sum_{\ell=1}^p \frac{a_{i,\ell}^{r,l} g_{i,\ell}^{r,l}}{\vartheta_{i,\ell}^{r,l}(g_{i,1}^{r,l}, \ldots, g_{i,\ell}^{r,l})}$$

*where*

$$0 < m_l \leq \vartheta_{i,\ell}^{r,l}(g_{i,1}^{r,l}, \ldots, g_{i,\ell}^{r,l}) \leq M_l \quad and \quad 0 < a_l \leq a_{i,\ell}^{r,l} \leq A_l$$

*for all possible values of $i, \ell, r, l$. If $1 \leq K(O_l^i) \leq K$ and $0 < \Xi^- < w(O_l^i) < \Xi^+$, then Algorithm 5 admits an identical convergence bound as Theorem 20, with $\Psi_3$, $\Psi_4$ replaced by*

$$\Psi_3 = (1 - \widetilde{\beta}_1^T)\eta\eta_\ell CT\widetilde{L}\|\Phi_1^K\|^2,$$
$$\Psi_4 = (1 - \widetilde{\beta}_1)\eta\eta_\ell CTLc(\widetilde{\beta}_1)\|\Phi_2^K\|^2,$$
$$C = \frac{(\Xi^+)^2 K(K+1)(\max_{l \in [Op]} A_l^2)}{2\widetilde{\alpha}_1 \tau \min_{l \in [Op]} m_l^2}.$$

*The intermediary $\widetilde{\gamma}_1, \widetilde{\alpha}_1$ values are defined as*

$$\widetilde{\gamma}_1 := \eta_\ell \frac{\Xi^- \min_{l \in [Op]} a_l}{\max_{l \in [Op]} M_l}, \quad \widetilde{\alpha}_1 := \frac{\Xi^- \min_{l \in [Op]} a_l}{K(K+1) \max_{l \in [Op]} M_l}.$$

We have opted to provide a looser bound for expository purposes, and the proof straightforwardly generalizes to finer bounds that depend on the individual characteristics of the optimizer strategy (e.g. $m_l, M_l, A_l$, etc). The extension to server-side Adam updates follows analogous steps to Section C.2.

It is easy to show that under the bounded gradient assumption (Assumption 2), Adam, AdaGrad, and SGD (including under SM3 for the former two) all satisfy the optimizer condition depicted in Theorem 25. In Appendix E and F, we materialize two realizations of this framework as additional examples, using client-side Adam and AdaGrad with delayed preconditioner updates. Note that delayed updates require the debiasing term in Adam to be adjusted accordingly. To prove Theorem 25, we begin with the following lemma.

**Lemma 26.** *Under Algorithm 5, $|\Delta_i^{t,l}|$ is bounded by*

$$\Phi_1^K := \eta_\ell \Xi^+ \frac{K(K+1)\max_{l\in[Op]} A_l G}{2\min_{l\in[Op]} m_l},$$

*and the server-side pseudogradient is bounded in absolute value by*

$$\Phi_2^K := \min\left\{ \eta\sqrt{(1-\widetilde{\beta}_1)(1-\widetilde{\beta}_1^{2t})}, \frac{\eta}{\tau}\Phi_1^K \right\}.$$

*Proof.* Unraveling the definition of $\Delta_i^{t,l}$, we have

$$\Delta_i^{t,l} := -\eta_\ell w(O_l)\left( \sum_{p=1}^{K(O_l^i)} \sum_{\ell=1}^{p} \frac{a_{i,\ell}^{r,l} g_{i,\ell}^{r,l}}{\vartheta_{i,\ell}^{r,l}(g_{i,1}^{r,l}, \ldots, g_{i,\ell}^{r,l})} \right),$$

which immediately gives

$$|\Delta_i^{t,l}| \le \eta_\ell \Xi^+ \left( \sum_{p=1}^{K} \sum_{\ell=1}^{p} \frac{A_l G}{m_l} \right) = \eta_\ell \Xi^+ \frac{K(K+1) A_l G}{2 m_l}.$$

For the server bound, the proof is identical to Lemma 16. $\qquad\square$

We are now ready to prove Theorem 25.

*Proof.* As the proof follows a similar structure to Theorem 6, we provide only an outline for repetitive steps while focusing on differing aspects. As before, $L$-smoothness gives that

$$f(x_t) \le f(x_{t-1}) + \eta T_{0,0} + (1-\widetilde{\beta}_1)\eta \sum_{r=1}^{t} T_{0,r} + \frac{\eta^2 L}{2} \left\| \frac{\widetilde{\beta}_1^t \widetilde{m}_0 + (1-\widetilde{\beta}_1)\sum_{r=1}^{t} \widetilde{\beta}_1^{t-r}\Delta_r}{\sqrt{\widetilde{v}_t}+\tau} \right\|^2 \tag{16}$$

where for $r \in [t]$,

$$T_{0,r} = \widetilde{\beta}_1^{t-r} \left\langle \nabla f(x_{t-1}), \frac{\Delta_r}{\sqrt{\widetilde{v}_t}+\tau} \right\rangle \quad \text{and} \quad T_{0,0} = \left\langle \nabla f(x_{t-1}), \frac{\widetilde{\beta}_1^t \widetilde{m}_0}{\sqrt{\widetilde{v}_t}+\tau} \right\rangle.$$

Decomposing $T_{0,r}$ as

$$T_{0,r} = \underbrace{\widetilde{\beta}_1^{t-r} \left\langle \nabla f(x_{t-1}), \frac{\Delta_r}{\sqrt{\widetilde{v}_t}+\tau} - \frac{\Delta_r}{\sqrt{\widetilde{v}_{t-1}}+\tau} \right\rangle}_{T_{1,r}} + \underbrace{\widetilde{\beta}_1^{t-r} \left\langle \nabla f(x_{t-1}), \frac{\Delta_r}{\sqrt{\widetilde{v}_{t-1}}+\tau} \right\rangle}_{T_{2,r}},$$

$T_{1,r}$ is bounded by

$$T_{1,r} \le \frac{\|\Phi_1^K\| G \widetilde{\beta}_1^{t-r}}{\tau} \sum_{j=1}^{d} \left[ \frac{\Delta_t^2}{\widetilde{v}_t} \right]_j.$$

For $T_{2,r}$, we aim to apply a further decomposition for $\gamma_r > 0$,

$$T_{2,r} = \underbrace{\widetilde{\beta}_1^{t-r} \left\langle \frac{\nabla f(x_{t-1})}{\sqrt{\widetilde{v}_{t-1}}+\tau}, \Delta_r + \gamma_r \nabla f(x_{t-1}) \right\rangle}_{T_{2,r}^1} - \gamma_r \widetilde{\beta}_1^{t-r} \left\| \frac{\nabla f(x_{t-1})}{\sqrt{\sqrt{\widetilde{v}_{t-1}}+\tau}} \right\|^2.$$

Unraveling the definition of $\Delta_r$ gives

$$\Delta_r = \frac{1}{\sum_{l\in[Op]}|S_l^r|} \sum_{l\in[Op]} \sum_{i\in S_l^r} \Delta_i^{r,l} = \frac{-\eta_\ell}{\sum_{l\in[Op]}|S_l^r|} \sum_{l\in[Op]} \sum_{i\in S_l^r} \sum_{p=1}^{K(O_l^i)} \sum_{\ell=1}^{p} \frac{w(O_l) a_{i,\ell}^{r,l} g_{i,\ell}^{r,l}}{\vartheta_{i,\ell}^{r,l}(g_{i,1}^{r,l}, \ldots, g_{i,\ell}^{r,l})},$$

which induces the following value

$$\gamma_r := \frac{\eta_\ell}{\sum_{l\in[Op]}|S_l^t|} \sum_{l\in[Op]} \sum_{i\in S_l^t} \sum_{p=1}^{K(O_l^i)} \sum_{\ell=1}^{p} \frac{w(O_l)a_{i,\ell}^{r,l}}{\vartheta_{i,\ell}^{r,l}(g_{i,1}^{r,l},\ldots,g_{i,\ell}^{r,l})} = \sum_{l\in[Op]} \gamma_r^l.$$

For the purposes of the proof, we shall consider a local device to have been dropped and unsampled if any runs less than 1 epoch. Then, we have

$$\gamma_r \in [\widetilde{\gamma}_1, \widetilde{\gamma}_2] := \left[\eta_\ell \frac{\Xi^- \min_{l\in[Op]} a_l}{\max_{l\in[Op]} M_l}, \eta_\ell \frac{\Xi^+ K(K+1)\max_{l\in[Op]} a_l}{2\min_{l\in[Op]} M_l}\right].$$

Expanding $T_{2,r}^1$ for $\alpha_r^l > 0$ to be fixed,

$$\widetilde{\beta}_1^{t-r} \left\langle \frac{\nabla f(x_{t-1})}{\sqrt{\widetilde{v}_{t-1}} + \tau}, \Delta_r + \gamma_r \nabla f(x_{t-1}) \right\rangle$$

$$= \frac{\widetilde{\beta}_1^{t-r}}{\sum_{l\in[Op]}|S_l^r|} \sum_{l\in[Op]} \sum_{i\in S_l^r} \sum_{p=1}^{K(O_l^i)} \sum_{\ell=1}^{p} \left\langle \frac{\nabla f(x_{t-1})}{\sqrt{\widetilde{v}_{t-1}} + \tau}, \frac{\eta_\ell w(O_l)a_{i,\ell}^{r,l}(\nabla f(x_{t-1}) - g_{i,\ell}^{r,l})}{\vartheta_{i,\ell}^{r,l}(g_{i,1}^{r,l},\ldots,g_{i,\ell}^{r,l})} \right\rangle$$

$$\leq \frac{\eta_\ell\widetilde{\beta}_1^{t-r}}{4\sum_{l\in[Op]}|S_l^r|} \sum_{l\in[Op]} \alpha_r^l \sum_{i\in\mathcal{S}_l^r} K(O_l^i)(K(O_l^i)+1) \left\| \frac{\nabla f(x_{t-1})}{\sqrt{\sqrt{\widetilde{v}_{t-1}} + \tau}} \right\|^2$$

$$+ \frac{\eta_\ell\widetilde{\beta}_1^{t-r}}{2\sum_{l\in[Op]}|S_l^r|} \sum_{l\in[Op]} \frac{1}{\alpha_r^l} \sum_{i\in S_l^r} \sum_{p=1}^{K(O_l^i)} \sum_{\ell=1}^{p} \left\| \frac{w(O_l)a_{i,\ell}^{r,l}\left(\nabla f(x_{t-1}) - \nabla F_i(x_{i,\ell-1}^{r,l})\right)}{\vartheta_{i,\ell}^{r,l}(g_{i,1}^{r,l},\ldots,g_{i,\ell}^{r,l})\sqrt{\sqrt{\widetilde{v}_{t-1}} + \tau}} \right\|^2$$

$$\leq \frac{\eta_\ell\widetilde{\beta}_1^{t-r}\max_{l\in[Op]}\alpha_r^l K(K+1)}{4} \left\| \frac{\nabla f(x_{t-1})}{\sqrt{\sqrt{\widetilde{v}_{t-1}} + \tau}} \right\|^2$$

$$+ \frac{\eta_\ell\widetilde{\beta}_1^{t-r}(\Xi^+)^2}{2\tau\sum_{l\in[Op]}|S_l^r|} \sum_{l\in[Op]} \frac{A_l^2}{\alpha_r^l m_l^2} \sum_{i\in S_l^r} \sum_{p=1}^{K(O_l^i)} \sum_{\ell=1}^{p} \left\| \nabla f(x_{t-1}) - \nabla F_i(x_{i,\ell-1}^{r,l}) \right\|^2$$

We aim to control the first term by setting for all $l \in [Op]$

$$\alpha_r^l = \frac{\gamma_r}{\eta_\ell K(K+1)} \in [\widetilde{\alpha}_1, \widetilde{\alpha}_2] := \left[\frac{\Xi^- \min_{l\in[Op]} a_l}{K(K+1)\max_{l\in[Op]} M_l}, \frac{\Xi^+ K(K+1)\max_{l\in[Op]} a_l}{2K(K+1)\min_{l\in[Op]} M_l}\right].$$

Via gradient clipping as before, we have

$$\left\| \nabla f(x_{t-1}) - \nabla F_i(x_{i,\ell-1}^{r,l}) \right\|^2 \leq 2L(t-r)^2\|\Phi_2^K\|^2 + 2\widetilde{L}\|\Phi_1^K\|^2.$$

Noting that

$$\frac{\eta_\ell\widetilde{\beta}_1^{t-r}(\Xi^+)^2}{2\tau\sum_{l\in[Op]}|S_l^r|} \sum_{l\in[Op]} \frac{A_l^2}{\alpha_r^l m_l^2} \sum_{i\in S_l^r} \sum_{p=1}^{K(O_l^i)} \sum_{\ell=1}^{p} \left\| \nabla f(x_{t-1}) - \nabla F_i(x_{i,\ell-1}^{r,l}) \right\|^2$$

$$\leq \frac{\eta_\ell(\Xi^+)^2 K(K+1)(\max_{l\in[Op]} A_l^2)}{2\widetilde{\alpha}_1\tau\min_{l\in[Op]} m_l^2} \left(L\widetilde{\beta}_1^{t-r}(t-r)^2\|\Phi_2^K\|^2 + \widetilde{L}\widetilde{\beta}_1^{t-r}\|\Phi_1^K\|^2\right),$$

collecting terms into equation (16) gives that

$$f(x_t) \leq f(x_{t-1}) + \eta T_{0,0} + \eta^2 L \left\| \frac{\widetilde{\beta}_1^t \widetilde{m}_0}{\sqrt{\widetilde{v}_t} + \tau} \right\|^2 + \frac{\eta^2 L d \|\Phi_1^K\|^2}{\tau^2} + (1 - \widetilde{\beta}_1)\eta \sum_{r=1}^{t} \left( \frac{\|\Phi_1^K\| G \widetilde{\beta}_1^{t-r}}{\tau} \sum_{j=1}^{d} \left[ \frac{\Delta_t^2}{\widetilde{v}_t} \right]_j \right)$$

$$+ (1 - \widetilde{\beta}_1)\eta\eta_\ell \sum_{r=1}^{t} \underbrace{\frac{(\Xi^+)^2 K(K+1)(\max_{l \in [Op]} A_l^2)}{2\widetilde{\alpha}_1 \tau \min_{l \in [Op]} m_l^2}}_{C} \left( L \widetilde{\beta}_1^{t-r}(t-r)^2 \|\Phi_2^K\|^2 + \widetilde{L} \widetilde{\beta}_1^{t-r} \|\Phi_1^K\|^2 \right)$$

$$+ (1 - \widetilde{\beta}_1)\eta \sum_{r=1}^{t} \left( -\frac{3\gamma_r \widetilde{\beta}_1^{t-r}}{4} \left\| \frac{\nabla f(x_{t-1})}{\sqrt{\sqrt{\widetilde{v}_{t-1}} + \tau}} \right\|^2 \right). \tag{17}$$

By initializing $\widetilde{m}_0 \leftarrow 0$ and enhancing the upper bound by substituting $\widetilde{\gamma}_1$ into $\gamma_r$, telescoping gives

$$\frac{3(1 - \widetilde{\beta}_1)\eta\widetilde{\gamma}_1}{4} \sum_{t=1}^{T} \sum_{r=1}^{t} \widetilde{\beta}_1^{t-r} \left\| \frac{\nabla f(x_{t-1})}{\sqrt{\sqrt{\widetilde{v}_{t-1}} + \tau}} \right\|^2 \leq f(x_0) - f(x^*) + \frac{(1 - \widetilde{\beta}_1)\eta\|\Phi_1^K\| G}{\tau} \sum_{t=1}^{T} \sum_{r=1}^{t} \sum_{j=1}^{d} \widetilde{\beta}_1^{t-r} \left[ \frac{\Delta_t^2}{\widetilde{v}_t} \right]_j$$

$$+ \frac{\eta^2 LTd\|\Phi_1^K\|^2}{\tau^2} + (1 - \widetilde{\beta}_1)\eta\eta_\ell C \sum_{t=1}^{T} \sum_{r=1}^{t} \left( L \widetilde{\beta}_1^{t-r}(t-r)^2 \|\Phi_2^K\|^2 + \widetilde{L} \widetilde{\beta}_1^{t-r} \|\Phi_1^K\|^2 \right). \tag{18}$$

Again by noting that

$$\frac{3(1 - \widetilde{\beta}_1)\eta\widetilde{\gamma}_1}{4} \sum_{t=1}^{T} \sum_{r=1}^{t} \widetilde{\beta}_1^{t-r} \left\| \frac{\nabla f(x_{t-1})}{\sqrt{\sqrt{\widetilde{v}_{t-1}} + \tau}} \right\|^2 \geq \frac{3(1 - \widetilde{\beta}_1)\eta\widetilde{\gamma}_1 T}{4 \left( \sqrt{T\|\Phi_1^K\|^2 + \widetilde{v}_0} + \tau \right)} \min_{t \in [T]} \|\nabla f(x_{t-1})\|^2,$$

Lemmas 21 and 22 give that

$$\frac{3(1 - \widetilde{\beta}_1)\eta\widetilde{\gamma}_1 T}{4 \left( \sqrt{T\|\Phi_1^K\|^2 + \widetilde{v}_0} + \tau \right)} \min_{t \in [T]} \|\nabla f(x_{t-1})\|^2 \leq f(x_0) - f(x^*) + \frac{\eta^2 LTd\|\Phi_1^K\|^2}{\tau^2}$$

$$+ (1 - \widetilde{\beta}_1^T)\eta\eta_\ell CT\widetilde{L}\|\Phi_1^K\|^2 + (1 - \widetilde{\beta}_1)\eta\eta_\ell CTLc(\widetilde{\beta}_1)\|\Phi_2^K\|^2$$

$$+ \frac{\eta d\|\Phi_1^K\| G \left( 1 - \widetilde{\beta}_1 + \log \left( 1 + \frac{T\|\Phi_1^K\|^2}{\tau^2} \right) \right)}{\tau}.$$

This implies that

$$\min_{t \in [T]} \|\nabla f(x_{t-1})\|^2 \leq \frac{\Psi_1 + \Psi_2 + \Psi_3 + \Psi_4 + \Psi_5}{\Psi_6},$$

where

$$\Psi_1 = f(x_0) - f(x^*),$$
$$\Psi_2 = \frac{\eta^2 LTd\|\Phi_1^K\|^2}{\tau^2},$$
$$\Psi_3 = (1 - \widetilde{\beta}_1^T)\eta\eta_\ell CT\widetilde{L}\|\Phi_1^K\|^2,$$
$$\Psi_4 = (1 - \widetilde{\beta}_1)\eta\eta_\ell CTLc(\widetilde{\beta}_1)\|\Phi_2^K\|^2,$$
$$\Psi_5 = \frac{\eta d\|\Phi_1^K\| G \left( 1 - \widetilde{\beta}_1 + \log \left( 1 + \frac{T\|\Phi_1^K\|^2}{\tau^2} \right) \right)}{\tau},$$
$$\Psi_6 = \frac{3(1 - \widetilde{\beta}_1)\eta\widetilde{\gamma}_1 T}{4 \left( \sqrt{T\|\Phi_1^K\|^2 + \widetilde{v}_0} + \tau \right)},$$
$$C = \frac{(\Xi^+)^2 K(K+1)(\max_{l \in [Op]} A_l^2)}{2\widetilde{\alpha}_1 \tau \min_{l \in [Op]} m_l^2}.$$

The intermediary $\widetilde{\gamma}_1, \widetilde{\alpha}_1$ values are defined as

$$\widetilde{\gamma}_1 := \eta_\ell \frac{\Xi^- \min_{l \in [Op]} a_l}{\max_{l \in [Op]} M_l}, \quad \widetilde{\alpha}_1 := \frac{\Xi^- \min_{l \in [Op]} a_l}{K(K+1) \max_{l \in [Op]} M_l}.$$

$\square$

## E    ADAM DELAYED MOMENT UPDATES (ADMU)

We begin with a brief description of ADAM (Kingma & Ba, 2015).

---

**Algorithm 6** Adam Optimization Algorithm

---

**Require:** $\eta_\ell$: Step size
**Require:** $\beta_1, \beta_2 \in [0, 1)$: Exponential decay rates for the moment estimates
**Require:** $f(x)$: Stochastic objective function with parameters $x$
**Require:** $\varepsilon > 0$: Smoothing term
**Require:** $x_0$: Initial parameter vector
 1: Initialize $m_0 \leftarrow 0$ (1st moment vector)
 2: Initialize $v_0 \leftarrow 0$ (2nd moment vector)
 3: Initialize $t \leftarrow 0$ (Timestep)
 4: **while** not converged **do**
 5:     $t \leftarrow t + 1$
 6:     $g_t \leftarrow \nabla_x f_t(x_{t-1})$
 7:     $m_t \leftarrow \beta_1 \cdot m_{t-1} + (1 - \beta_1) \cdot g_t$
 8:     $v_t \leftarrow \beta_2 \cdot v_{t-1} + (1 - \beta_2) \cdot g_t^2$
 9:     $\hat{m}_t \leftarrow m_t / (1 - \beta_1^t)$
10:     $\hat{v}_t \leftarrow v_t / (1 - \beta_2^t)$
11:     $x_t \leftarrow x_{t-1} - \eta_\ell \cdot \hat{m}_t / (\sqrt{\hat{v}_t} + \varepsilon)$
12: **end while**
13: **return** $x_t$

---

Considering client-side resource constraints in the federated setting, we propose an adapted version of Adam with delayed precondtioner updates aimed at relieving the cost of moment estimate computation in Algorithm 7 which we call ADMU.

Following Kingma & Ba (2015), we provide an intuitive justification for the initialization bias correction employed in ADMU. Recall that the motivation for adaptive step-size in ADAM is updating the parameters via empirical estimates of the pseudo-gradient $\mathbb{E}[g]/\sqrt{\mathbb{E}[g^2]}$, which allows for both momentum and autonomous annealing near steady states. The square root is taken in the denominator to homogenize the degree of the gradient. Bias correction for ADMU adheres to the same principle, while requiring an additional assumption of gradient stabilization during the $z$-step preconditioner update delay. An equivalent formulation of the moment estimates in Algorithm 7 for general $t$ is given

$$m_t = m_0 \beta_1^t + (1 - \beta_1) \sum_{r=1}^{t} \beta_1^{t-r} \cdot g_r,$$

$$v_t = v_0 \beta_2^{\lfloor \frac{t-1}{z} \rfloor + 1} + (1 - \beta_2) \sum_{r=1}^{t} \beta_2^{\lfloor \frac{t-1}{z} \rfloor + 1 - \lceil \frac{r}{z} \rceil} \cdot g_{\lceil \frac{r}{z} \rceil z - z + 1} \odot g_{\lceil \frac{r}{z} \rceil z - z + 1} \cdot \chi_{\left\{ \frac{r-1}{z} \in \mathbb{Z}_{\geq 0} \right\}}$$

$$= v_0 \beta_2^{\lfloor \frac{t-1}{z} \rfloor + 1} + (1 - \beta_2) \sum_{r=1}^{\lceil \frac{t}{z} \rceil} \beta_2^{\lceil \frac{t}{z} \rceil - r} g_{(r-1)z+1} \odot g_{(r-1)z+1}. \tag{19}$$

We work with $v_t$ as the proof for $m_t$ is analogous with $z = 1$. Assume that the gradients $g_1, \ldots, g_t$ are drawn from a latent gradient distribution $g_i \sim \widetilde{\mathcal{D}}(g_i)$. We aim to extract a relation between the expected delayed exponential moving average of the second moment $\mathbb{E}[v_t]$ and the true gradient

---

**Algorithm 7** Adam with Delayed Moment Updates (ADMU)

---

**Require:** $\eta_\ell$: Step size
**Require:** $z \in \mathbb{Z}_{\geq 1}$: Step delay for second moment estimate updates (where $z = 1$ gives no delay)
**Require:** $\beta_1, \beta_2 \in [0, 1)$: Exponential decay rates for the moment estimates
**Require:** $f(x)$: Stochastic objective function with parameters $x$
**Require:** $x_0$: Initial parameter vector
**Require:** $\varepsilon > 0$: Smoothing term
 1: Initialize $m_0 \leftarrow 0$ (1st moment vector)
 2: Initialize $v_0 \leftarrow 0$ (2nd moment vector)
 3: Initialize $t \leftarrow 0$ (Timestep)
 4: **while** not converged **do**
 5:     $t \leftarrow t + 1$
 6:     $g_t \leftarrow \nabla_x f_t(x_{t-1})$
 7:     $m_t \leftarrow \beta_1 \cdot m_{t-1} + (1 - \beta_1) \cdot g_t$
 8:     $\hat{m}_t \leftarrow m_t / (1 - \beta_1^t)$
 9:     **if** $(t-1)/z \in \mathbb{Z}$ **then**
10:         $v_t \leftarrow \beta_2 \cdot v_{t-1} + (1 - \beta_2) \cdot g_t^2$
11:         $\hat{v}_t \leftarrow v_t / (1 - \beta_2^{\lfloor \frac{t-1}{z} \rfloor + 1})$
12:     **else**
13:         $\hat{v}_t \leftarrow \hat{v}_{t-1}$
14:     **end if**
15:     $x_t \leftarrow x_{t-1} - \eta_\ell \cdot \hat{m}_t / (\sqrt{\hat{v}_t} + \varepsilon)$
16: **end while**
17: **return** $x_t$

---

expectation $\mathbb{E}[g_t^2]$. Taking expectation of both sides in equation (19),

$$\mathbb{E}[v_t] = v_0 \beta_1^{\lfloor \frac{t-1}{z} \rfloor + 1} + (1 - \beta_2) \sum_{r=1}^{\lceil \frac{t}{z} \rceil} \beta_2^{\lceil \frac{t}{z} \rceil - r} \mathbb{E}\left[g_{(r-1)z+1}^2\right]$$

$$\approx \zeta + (1 - \beta_2) \mathbb{E}\left[g_t^2\right] \sum_{r=1}^{\lceil \frac{t}{z} \rceil} \beta_2^{\lceil \frac{t}{z} \rceil - r}$$

$$\approx \mathbb{E}[g_t^2] \left(1 - \beta_1^{\lfloor \frac{t-1}{z} \rfloor + 1}\right).$$

Here, we have used zero initialization for the first moment estimate, while accumulating any error terms in $\zeta$. Several assumptions can lead to small $\zeta$. As in Kingma & Ba (2015), we assume that $\beta_1$ is chosen small enough that the exponential moving average decay undermines the influence of non-recent gradients $g_i$ for $i < \lceil \frac{t}{z} \rceil z - z + 1$. A second assumption is that the latent gradient distribution remains stable during the $z$-step delay as training progresses, allowing the approximation $\mathbb{E}[g_t] \approx \mathbb{E}[g_{\lceil \frac{t}{z} \rceil z - z + 1}]$. This leaves the residual scaling of the true gradient second moment of the form $1 - \beta^\varphi$, which is caused by (zero) initialization as setting $v_0 = \mathbb{E}[g_t^2]$ eliminates $\beta^\varphi$. Therefore, bias correction is enforced by scaling the empirical $v_t$ estimate by the inverse. We note that $v_0$ need not be initialized to 0, in which case we should additionally translate $v_t$ by $-v_0 \beta_1^{\lfloor \frac{t-1}{z} \rfloor + 1}$ prior to the inverse scaling.

### E.1   NON-CONVEX CONVERGENCE ANALYSIS

A description of FedAdaAdam is given as Algorithm 8. A few remarks are in order. Firstly, to allow for straggler mitigation, we allow the number of client $i$ epochs $\overline{K}_i^t$ at timestep $t$ to vary among the clients $i \in \mathcal{S}_i$. Although Algorithm 8 sets a schedule for client epochs and pseudogradient weights for clarity of exposition, dynamic allocation still allows the convergence proof to go through, as long as the schedule weights are bounded. By default, we set $\overline{K}^t = K$ and $\Xi^t = B = 1$ to avoid tuning a large number of hyperparameters or having to sample from a client epoch count distribution for the client subsampling case.

**Algorithm 8** Adaptive server-side ADAGRAD and client-side ADAM (FedAdaAdam)

**Require:** Update delay step size $z \in \mathbb{Z}_{\geq 1}$, initializations $x_0, \widetilde{v}_0 \geq \tau^2$ and $\widetilde{m}_0 \leftarrow 0$
**Require:** Global and local decay parameters $\widetilde{\beta}_1, \widetilde{\beta}_2, \beta_1, \beta_2 \in [0, 1)$
**Require:** Pseudogradient weighting schedule $\Xi^1 \times \cdots \times \Xi^T \in \mathbb{R}^{|\mathcal{S}^1|} \times \cdots \times \mathbb{R}^{|\mathcal{S}^T|}$ for $\|\Xi^t\|_\infty \leq B$
**Require:** Client epoch schedule $\overline{K}^1 \times \cdots \times \overline{K}^T \in \mathbb{Z}_{\geq 1}^{|\mathcal{S}^1|} \times \cdots \times \mathbb{Z}_{\geq 1}^{|\mathcal{S}^T|}$ for $\|\overline{K}^t\|_\infty \leq K, \forall t \in [T]$
**Require:** Local epsilon smoothing term $\varepsilon_s > 0$
 1: **for** $t = 1, \dots, T$ **do**
 2:      Sample subset $\mathcal{S}^t \subset [N]$ of clients
 3:      **for** each client $i \in \mathcal{S}^t$ (in parallel) **do**
 4:          $x_{i,0}^t \leftarrow x_{t-1}$
 5:          Initialize $m_0, v_0 \geq 0$ with default values $m_0, v_0 \leftarrow 0$
 6:          **for** $k = 1, \dots, \overline{K}_i^t$ **do**
 7:              Draw stochastic gradient $g_{i,k}^t \sim \mathcal{D}(x_{i,k-1}^t)$ with mean $\nabla F_i(x_{i,k-1}^t) \in \mathbb{R}^d$
 8:              $m_k \leftarrow \beta_1 \cdot m_{k-1} + (1 - \beta_1) \cdot g_{i,k}^t$
 9:              $\hat{m}_k \leftarrow m_k/(1 - \beta_1^k)$
10:              **if** $(k - 1)/z \in \mathbb{Z}$ **then**
11:                  $v_k \leftarrow \beta_2 \cdot v_{k-1} + (1 - \beta_2) \cdot g_{i,k}^t \odot g_{i,k}^t$
12:                  $\hat{v}_k \leftarrow v_k/(1 - \beta_2^{\lfloor \frac{k-1}{z} \rfloor + 1})$
13:              **else**
14:                  $v_k \leftarrow v_{k-1}$
15:              **end if**
16:              **if** $0 < \|\hat{m}_k/(\sqrt{\hat{v}_k} + \epsilon)\| < \varepsilon_s$ **then**
17:                  $m_k \leftarrow 0$
18:              **end if**
19:              $x_{i,k}^t \leftarrow x_{i,k-1}^t - \eta_\ell \cdot \hat{m}_k/(\sqrt{\hat{v}_k} + \epsilon)$
20:          **end for**
21:          $\Delta_i^t = \Xi_i^t \left( x_{i,\overline{K}_i^t}^t - x_{t-1} \right)$
22:      **end for**
23:      $\Delta_t = \frac{1}{|\mathcal{S}^t|} \sum_{i \in \mathcal{S}^t} \Delta_i^t$
24:      $\widetilde{m}_t = \widetilde{\beta}_1 \widetilde{m}_{t-1} + (1 - \widetilde{\beta}_1)\Delta_t$
25:      $\widetilde{v}_t = \widetilde{v}_{t-1} + \Delta_t^2$
26:      $x_t = x_{t-1} + \eta \frac{\widetilde{m}_t}{\sqrt{\widetilde{v}_t} + \tau}$
27: **end for**

Secondly, for the purposes of the proof we shall consider a local device to have been dropped and unsampled if any runs less than 1 epoch. We also enforce that pseudogradient weights are bounded positively from below, i.e. $\Xi_i^t > \varepsilon_w > 0$. We now provide a convergence bound for the general, non-convex case which holds for both full and partial client participation.

**Corollary 27.** *For Algorithm 8, we have an identical bound to Theorem 6 with $\Psi_3, \Psi_4$ replaced by*

$$\Psi_3 = \frac{(1 - \widetilde{\beta}_1^T)\eta\eta_\ell(1 - \beta_1^{2K})K\widetilde{L}B^2 T\|\Phi_1^K\|^2}{2\widetilde{\alpha}_1\tau\varepsilon^2},$$

$$\Psi_4 = \frac{(1 - \widetilde{\beta}_1)\eta\eta_\ell(1 - \beta_1^{2K})KLTB^2 c(\widetilde{\beta}_1)\|\Phi_2^K\|^2}{2\widetilde{\alpha}_1\tau\varepsilon^2}.$$

*Here, the intermediary $\widetilde{\gamma}_1, \widetilde{\alpha}_1$ values are defined for $K^- := \min_{i,t} \overline{K}_i^t \geq 1$ as*

$$\widetilde{\gamma}_1 := \eta_\ell\varepsilon_w \sum_{p=1}^{K^-} \frac{1 - \beta_1^p}{G\sqrt{1 - \beta_2^{\lceil \frac{p}{z} \rceil}} + \varepsilon}, \quad \widetilde{\alpha}_1 := \sum_{p=1}^{K^-} \frac{\varepsilon_w (1 - \beta_1^p)}{\left(G\sqrt{1 - \beta_2^{\lceil \frac{p}{z} \rceil}} + \varepsilon\right)(K+1)^2}.$$

The proof is subsumed by or analogous to Theorems 6 and 25, with changes summarized in the following lemma.

**Lemma 28.** *Under Algorithm 8, $|\Delta_i^t|$ is bounded by*

$$|\Delta_i^t| \leq \Phi_1^{\overline{K}_i^t} := |\Xi_i^t| \cdot \left(\eta_\ell\overline{K}_i^t \sqrt{\left(\sum_{r=1}^{\lceil \frac{\overline{K}_i^t}{z} \rceil} \frac{\beta_1^{2\lceil \frac{\overline{K}_i^t}{z} \rceil - 2r}}{\beta_2^{\lceil \frac{\overline{K}_i^t}{z} \rceil - r}}\right)} + \Phi_0^{\overline{K}_i^t}\right)$$

*where*

$$\Phi_0^{\overline{K}_i^t} := \frac{\overline{K}_i^t G\eta_\ell(1 - \beta_1^{\overline{K}_i^t})}{\varepsilon}.$$

*Proof.* Recall that $\Delta_t = 1/|\mathcal{S}^t| \sum_{i \in \mathcal{S}^t} \Delta_i^t$ and $\Delta_i^t = \Xi_i^t \left(x_{i,\overline{K}_i^t}^t - x_{i,0}^t\right)$. By telescoping for $\overline{K}_i^t$ local steps and the definition of gradient updates in ADMU, we obtain

$$\Delta_i^t = \sum_{p=1}^{\overline{K}_i^t} -\eta_\ell\Xi_i^t \frac{\hat{m}_p}{\sqrt{\hat{v}_p} + \varepsilon} = -\eta_\ell\Xi_i^t \sum_{p=1}^{\overline{K}_i^t} \frac{m_0\beta_1^p + (1 - \beta_1)\sum_{r=1}^p \beta_1^{p-r} \cdot g_{i,r}^t}{\sqrt{v_0\beta_2^{\lfloor \frac{p-1}{z} \rfloor + 1} + (1 - \beta_2)\sum_{r=1}^{\lceil \frac{p}{z} \rceil} \beta_2^{\lceil \frac{p}{z} \rceil - r}(g_{i,(r-1)z+1}^t)^2} + \varepsilon}$$

We assume $m_0, v_0 \leftarrow 0$ for expository purposes, although $v_0 > 0$ also suffices for the analysis (ending in a slightly different $\Phi_1^{\overline{K}_i^t}$). This gives that

$$\Delta_i^t = -\eta_\ell\Xi_i^t \sum_{p=1}^{\overline{K}_i^t} \frac{(1 - \beta_1)\sum_{r=1}^p \beta_1^{p-r} \cdot g_{i,r}^t}{\sqrt{(1 - \beta_2)\sum_{r=1}^{\lceil \frac{p}{z} \rceil} \beta_2^{\lceil \frac{p}{z} \rceil - r}(g_{i,(r-1)z+1}^t)^2} + \varepsilon}$$

$$= -\eta_\ell\Xi_i^t \sum_{p=1}^{\overline{K}_i^t} \frac{(1 - \beta_1)\sum_{r=1}^{\lceil \frac{p}{z} \rceil} \beta_1^{\lceil \frac{p}{z} \rceil - r} \cdot g_{i,(r-1)z+1}^t}{\sqrt{(1 - \beta_2)\sum_{r=1}^{\lceil \frac{p}{z} \rceil} \beta_2^{\lceil \frac{p}{z} \rceil - r}(g_{i,(r-1)z+1}^t)^2} + \varepsilon}$$

$$- \eta_\ell\Xi_i^t \sum_{p=1}^{\overline{K}_i^t} \frac{(1 - \beta_1)\sum_{r=1}^p \beta_1^{p-r} \cdot g_{i,r}^t \cdot \chi_{\left\{\frac{p-1}{z} \notin \mathbb{Z}\right\}}}{\sqrt{(1 - \beta_2)\sum_{r=1}^{\lceil \frac{p}{z} \rceil} \beta_2^{\lceil \frac{p}{z} \rceil - r}(g_{i,(r-1)z+1}^t)^2} + \varepsilon}.$$

To obtain a deterministic bound, we cannot ignore the worst-case stochastic realization that $g^t_{i,(r-1)z+1} = 0$ for $\forall r \in [\lceil \frac{p}{z} \rceil]$. Therefore, we form the intermediary upper bound

$$\left| \Delta^t_i \right| \leq \eta_\ell |\Xi^t_i| \sum_{p=1}^{\overline{K}^t_i} \frac{(1-\beta_1) \sum_{r=1}^{\lceil \frac{p}{z} \rceil} \beta_1^{\lceil \frac{p}{z} \rceil - r} \cdot \left| g^t_{i,(r-1)z+1} \right|}{\sqrt{(1-\beta_2) \sum_{r=1}^{\lceil \frac{p}{z} \rceil} \beta_2^{\lceil \frac{p}{z} \rceil - r} (g^t_{i,(r-1)z+1})^2 + \varepsilon}}$$

$$+ \frac{\eta_\ell |\Xi^t_i|(1-\beta_1)}{\varepsilon} \left( \sum_{p=1}^{\overline{K}^t_i} \sum_{r=1}^{p} \beta_1^{p-r} \cdot \left| g^t_{i,r} \right| \cdot \chi_{\left\{ \frac{p-1}{z} \notin \mathbb{Z} \right\}} \right). \tag{20}$$

Note that the first term is 0 in the worst-case scenario above, which implies that any non-negative upper bound is trivially satisfied. Therefore, we may assume without loss of generality that at least one sampled gradient $g^t_{i,(r-1)z+1}$ is nontrivial and remove $\varepsilon$ from the denominator to obtain an upper bound. By Cauchy-Schwartz, we have

$$\left( \sum_{r=1}^{\lceil \frac{p}{z} \rceil} \beta_2^{\lceil \frac{p}{z} \rceil - r} (g^t_{i,(r-1)z+1})^2 \right) \left( \sum_{r=1}^{\lceil \frac{p}{z} \rceil} \frac{\beta_1^{2\lceil \frac{p}{z} \rceil - 2r}}{\beta_2^{\lceil \frac{p}{z} \rceil - r}} \right) \geq \left( \sum_{r=1}^{\lceil \frac{p}{z} \rceil} \beta_1^{\lceil \frac{p}{z} \rceil - r} \cdot \left| g^t_{i,(r-1)z+1} \right| \right)^2$$

which implies

$$\left| \Delta^t_i \right| \leq \eta_\ell |\Xi^t_i| \sum_{p=1}^{\overline{K}^t_i} \sqrt{\left( \sum_{r=1}^{\lceil \frac{p}{z} \rceil} \frac{\beta_1^{2\lceil \frac{p}{z} \rceil - 2r}}{\beta_2^{\lceil \frac{p}{z} \rceil - r}} \right)} + \frac{\eta_\ell |\Xi^t_i|(1-\beta_1)}{\varepsilon} \left( \sum_{p=1}^{\overline{K}^t_i} \sum_{r=1}^{p} \beta_1^{p-r} \cdot \left| g^t_{i,r} \right| \cdot \chi_{\left\{ \frac{p-1}{z} \notin \mathbb{Z} \right\}} \right)$$

$$\leq \eta_\ell |\Xi^t_i| \sum_{p=1}^{\overline{K}^t_i} \sqrt{\left( \sum_{r=1}^{\lceil \frac{p}{z} \rceil} \frac{\beta_1^{2\lceil \frac{p}{z} \rceil - 2r}}{\beta_2^{\lceil \frac{p}{z} \rceil - r}} \right)} + \frac{\overline{K}^t_i G \eta_\ell |\Xi^t_i|(1-\beta_1)}{\varepsilon} \cdot \frac{(1-\beta_1^{\overline{K}^t_i})}{(1-\beta_1)}$$

$$\leq \eta_\ell |\Xi^t_i| \overline{K}^t_i \sqrt{\left( \sum_{r=1}^{\lceil \frac{\overline{K}^t_i}{z} \rceil} \frac{\beta_1^{2\lceil \frac{\overline{K}^t_i}{z} \rceil - 2r}}{\beta_2^{\lceil \frac{\overline{K}^t_i}{z} \rceil - r}} \right)} + \frac{\overline{K}^t_i G \eta_\ell |\Xi^t_i|(1-\beta_1^{\overline{K}^t_i})}{\varepsilon}.$$

$$\square$$

It can be shown that case of no update delay $z = 1$ allows for $\Phi_0^{\overline{K}^t_i} = 0$, following a similar proof to the one given above. Note that $\Phi_0^{\overline{K}^t_i}$ handles the superfluous gradient terms cemented by delaying preconditioner updates for the second moment, while moving averaging is performed for the first moment estimate. It also follows that $\Delta_t$ is also upper bounded by the identical bound scaled by $\max_t \|\Xi^t\|_\infty \leq B$, as the average of the $\Delta^t_i$.

# F ADAGRAD WITH DELAYED UPDATES (AGDU)

We present AdaGrad with delayed preconditioner as Algorithm 9 for completeness.

Note that due to delayed updates, local gradient updates are not necessarily elementwise bounded in absolute value by $\eta_\ell$. We may expand the delayed updates for $v_t$ as

$$v_t = v_0 + \sum_{r=1}^{\lceil \frac{t}{z} \rceil} g_{(r-1)z+1} \odot g_{(r-1)z+1}.$$

We have the following convergence bound.

**Corollary 29.** *Let* $K^- := \min_{i,t} \overline{K}^t_i \geq 1$ *and*

$$\widetilde{\gamma}_1 := \eta_\ell \varepsilon_w \sum_{p=1}^{K^-} \frac{1}{\sqrt{v_0 + \lceil \frac{K}{z} \rceil G^2} + \varepsilon}, \quad \widetilde{\alpha}_1 := \frac{\varepsilon_w K^-}{2K \left( \sqrt{v_0 + \lceil \frac{K}{z} \rceil G^2} + \varepsilon \right)}.$$

*Then Algorithm 10 has an identical convergence bound to Theorem 6.*

---

**Algorithm 9** AdaGrad with Delayed Updates (AGDU)

---

**Require:** $\eta_\ell$: Step size
**Require:** $z \in \mathbb{Z}_{\geq 1}$: Step delay for second moment estimate updates (where $z = 1$ gives no delay)
**Require:** $f(x)$: Stochastic objective function with parameters $x$
**Require:** $x_0$: Initial parameter vector
**Require:** $\varepsilon > 0$: Smoothing term
1:  Initialize $v_0 \leftarrow 0$ (2nd moment vector)
2:  Initialize $t \leftarrow 0$ (Timestep)
3:  **while** not converged **do**
4:      $t \leftarrow t + 1$
5:      $g_t \leftarrow \nabla_x f_t(x_{t-1})$
6:      **if** $(t-1)/z \in \mathbb{Z}$ **then**
7:          $v_t \leftarrow v_{t-1} + g_t^2$
8:      **else**
9:          $v_t \leftarrow v_{t-1}$
10:     **end if**
11:     $x_t \leftarrow x_{t-1} - \eta_\ell \cdot g_t/(\sqrt{v_t} + \varepsilon)$
12: **end while**
13: **return** $x_t$

---

**Algorithm 10** Adaptive server and client-side ADAGRAD (FedAdaAdagrad)

---

**Require:** Update delay step size $z \in \mathbb{Z}_{\geq 1}$, initializations $x_0, \widetilde{v}_0 \geq \tau^2$ and $\widetilde{m}_0 \leftarrow 0$
**Require:** Global decay parameter $\widetilde{\beta}_1 \in [0, 1)$
**Require:** Pseudogradient weighting schedule $\Xi^1 \times \cdots \times \Xi^T \in \mathbb{R}^{|\mathcal{S}^1|} \times \cdots \times \mathbb{R}^{|\mathcal{S}^T|}$ for $\|\Xi^t\|_\infty \leq B$
**Require:** Client epoch schedule $\overline{K}^1 \times \cdots \times \overline{K}^T \in \mathbb{Z}_{\geq 1}^{|\mathcal{S}^1|} \times \cdots \times \mathbb{Z}_{\geq 1}^{|\mathcal{S}^T|}$ for $\|\overline{K}^t\|_\infty \leq K, \forall t \in [T]$
**Require:** Local epsilon smoothing term $\varepsilon_s > 0$, global smoothing term $\tau > 0$
1:  **for** $t = 1, \ldots, T$ **do**
2:      Sample subset $\mathcal{S}^t \subset [N]$ of clients
3:      **for** each client $i \in \mathcal{S}^t$ (in parallel) **do**
4:          $x_{i,0}^t \leftarrow x_{t-1}$
5:          Initialize $v_0 \geq 0$ with default value $v_0 \leftarrow 0$ (what if use $\tau$ here?)
6:          **for** $k = 1, \ldots, \overline{K}_i^t$ **do**
7:              Draw stochastic gradient $g_{i,k}^t \sim \mathcal{D}(x_{i,k-1}^t)$ with mean $\nabla F_i(x_{i,k-1}^t) \in \mathbb{R}^d$
8:              $m_k \leftarrow g_{i,k}^t$
9:              **if** $(k-1)/z \in \mathbb{Z}$ **then**
10:                 $v_k \leftarrow v_{k-1} + g_{i,k}^t \odot g_{i,k}^t$
11:             **else**
12:                 $v_k \leftarrow v_{k-1}$
13:             **end if**
14:             **if** $0 < \|m_k/(\sqrt{v_k} + \epsilon)\| < \varepsilon_s$ **then**
15:                 $m_k \leftarrow 0$
16:             **end if**
17:             $x_{i,k}^t \leftarrow x_{i,k-1}^t - \eta_\ell \cdot m_k/(\sqrt{v_k} + \epsilon)$
18:         **end for**
19:         $\Delta_i^t = \Xi_i^t \left( x_{i,\overline{K}_i^t}^t - x_{t-1} \right)$
20:     **end for**
21:     $\Delta_t = \frac{1}{|\mathcal{S}^t|} \sum_{i \in \mathcal{S}^t} \Delta_i^t$
22:     $\widetilde{m}_t = \widetilde{\beta}_1 \widetilde{m}_{t-1} + (1 - \widetilde{\beta}_1)\Delta_t$
23:     $\widetilde{v}_t = \widetilde{v}_{t-1} + \Delta_t^2$
24:     $x_t = x_{t-1} + \eta \frac{\widetilde{m}_t}{\sqrt{\widetilde{v}_t} + \tau}$
25: **end for**

---

Similar to delayed Adam, the proof is analogous to Theorem 6 with changes summarized in the following lemma.

**Lemma 30.** *Under Algorithm 10, $|\Delta_i^t|$ is bounded by*

$$|\Delta_i^t| \leq \Phi_1^K := \eta_\ell B \left( \left\lfloor \frac{K-1}{z} \right\rfloor + 1 + \frac{KG}{\sqrt{v_0} + \varepsilon} \right).$$

*Proof.* Recall that $\Delta_t = 1/|\mathcal{S}^t| \sum_{i \in \mathcal{S}^t} \Delta_i^t$ and $\Delta_i^t = \Xi_i^t \left( x_{i,\overline{K}_i^t}^t - x_{i,0}^t \right)$. By telescoping for $\overline{K}_i^t$ local steps and the definition of gradient updates in FedAdaAdagrad, we obtain

$$\Delta_i^t = \sum_{p=1}^{\overline{K}_i^t} -\eta_\ell \Xi_i^t \frac{m_p}{\sqrt{v_p} + \varepsilon} = -\eta_\ell \Xi_i^t \sum_{p=1}^{\overline{K}_i^t} \frac{g_{i,p}^t}{\sqrt{v_0 + \sum_{r=1}^{\lceil \frac{p}{z} \rceil} (g_{i,(r-1)z+1}^t)^2} + \varepsilon}$$

For $\mathcal{F} = \{0, 1, \ldots, \lfloor (\overline{K}_i^t - 1)/z \rfloor\} z + 1$, we thus have that

$$\Delta_i^t = -\eta_\ell \Xi_i^t \sum_{p \in \mathcal{F}} \frac{g_{i,p}^t}{\sqrt{v_0 + \sum_{r=1}^{\lceil \frac{p}{z} \rceil} (g_{i,(r-1)z+1}^t)^2} + \varepsilon}$$

$$- \eta_\ell \Xi_i^t \sum_{p \in [\overline{K}_i^t] \setminus \mathcal{F}} \frac{g_{i,p}^t}{\sqrt{v_0 + \sum_{r=1}^{\lceil \frac{p}{z} \rceil} (g_{i,(r-1)z+1}^t)^2} + \varepsilon}.$$

To obtain a deterministic bound, we cannot ignore the worst-case stochastic realization that $g_{i,(r-1)z+1}^t = 0$ for $\forall r \in [\lceil \frac{p}{z} \rceil]$. Therefore, we form the upper bound

$$\left| \Delta_i^t \right| \leq \eta_\ell |\Xi_i^t| \sum_{p \in \mathcal{F}} \frac{|g_{i,p}^t|}{\sqrt{v_0 + |g_{i,p}^t|^2 + \sum_{r=1}^{\lceil \frac{p}{z} \rceil - 1} (g_{i,(r-1)z+1}^t)^2} + \varepsilon}$$

$$+ \frac{\eta_\ell |\Xi_i^t|}{\sqrt{v_0} + \varepsilon} \left( \sum_{p \in [\overline{K}_i^t] \setminus \mathcal{F}} |g_{i,p}^t| \right) \tag{21}$$

$$\leq \eta_\ell |\Xi_i^t| \left( \left\lfloor \frac{K-1}{z} \right\rfloor + 1 \right) + \frac{\eta_\ell |\Xi_i^t| KG}{\sqrt{v_0} + \varepsilon}$$

where the last line uses that the local epoch schedules are upper bounded by $K$. Noting that $\|\Xi_i^t\|_\infty \leq B$, we are done. $\square$

# G  DATASETS, MODELS, AND BASELINES

Below, we summarize the dataset statistics and provide a more in-depth description.

Table 1: Summary of datasets and models.

| Datasets | # Devices | Non-IID Partition | Model | Tasks |
|---|---|---|---|---|
| StackOverflow (Exchange, 2021) | 400 | Natural | Logistic Regression | 500-Class Tag Classification |
| CIFAR-100 (Krizhevsky, 2009) | 1000 | LDA | ViT-S | 100-Class Image Classification |
| GLD-23K (Weyand et al., 2020) | 233 | Natural | ViT-S | 203-Class Image Classification |
| FEMNIST (Caldas et al., 2018) | 500 | Natural | ViT-S | 62-Class Image Classification |

## G.1  STACKOVERFLOW DATASET

The StackOverflow dataset (Exchange, 2021) is a language dataset composed of questions and answers extracted from the StackOverflow online community. Each data entry includes associated metadata such as tags (e.g., "python"), the time the post was created, the title of the question, the score assigned to the question, and the type of post (question or answer). The dataset is partitioned

by users, with each client representing an individual user and their collection of posts. This dataset exhibits significant imbalance, with some users contributing only a few posts while others have a much larger number of entries. In this paper, we work with a randomly selected 400-client subset of the full StackOverflow Dataset, with a client participation fraction of $0.1$.

## G.2 GLD-23K DATASET

The GLD-23k dataset is a subset of the GLD-160k dataset introduced in Weyand et al. (2020). It contains 23,080 training images, 203 landmark labels, and 233 clients. Compared to CIFAR-10/100, the landmarks dataset consists of images of far higher quality and resolution, and therefore represents a more challenging learning task. The client particiation fraction for all GLD-23K experiments are set to $0.01$.

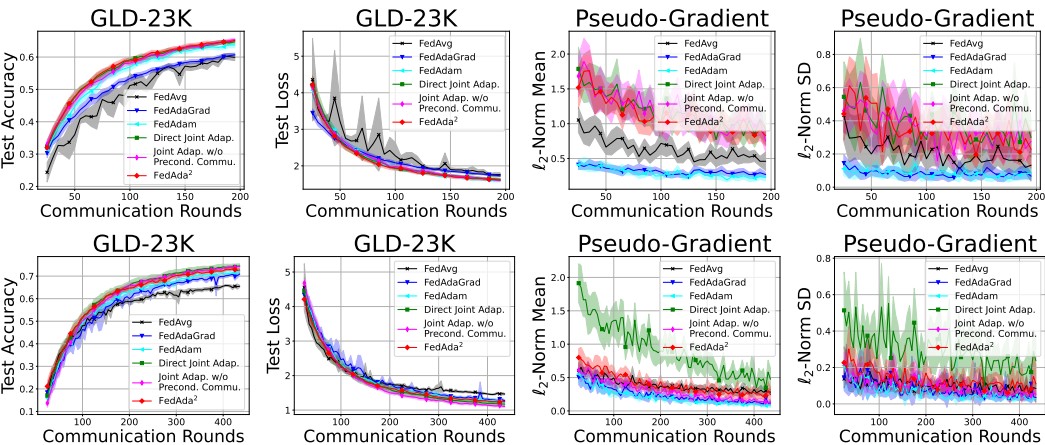

Figure 4: (Top) Additional results for the experiments in Figure 3 (b), where clients train over 5 epochs. (Bottom) Analogous experiments for full fine-tuning, where the entire net is unfrozen after replacing the classification layer. All adaptive optimizers are instantiated with Adam, with the exception of FedAdaGrad where the server-side adaptive optimizer is AdaGrad.

## G.3 CIFAR-100 DATASET

The CIFAR-10/100 datasets (Krizhevsky, 2009) consist of $32 \times 32 \times 3$ images. In the smaller variant CIFAR-10, there are 10 labels, with 50,000 training images and 10,000 test images. The 10 classes represent common objects: airplanes, automobiles, birds, cats, deer, dogs, frogs, horses, ships, and trucks. CIFAR-100 is meant to be an extension of CIFAR-10, consisting of 60,000 color images, but with 100 classes instead of 10. Each class in CIFAR-100 contains 600 images, and the dataset is similarly split into 50,000 training images and 10,000 test images. Unlike CIFAR-10, every class in CIFAR-100 is subsumed by one of 20 superclasses, and each image is provided a fine label and a coarse label that represents the former and latter (super-)class. In this paper, we train and evaluate all algorithms against the fine label. In Figure 5, we show the convergence of $\texttt{FedAda}^2$ as compared to all other adaptive or non-adaptive benchmarks using CIFAR-100.

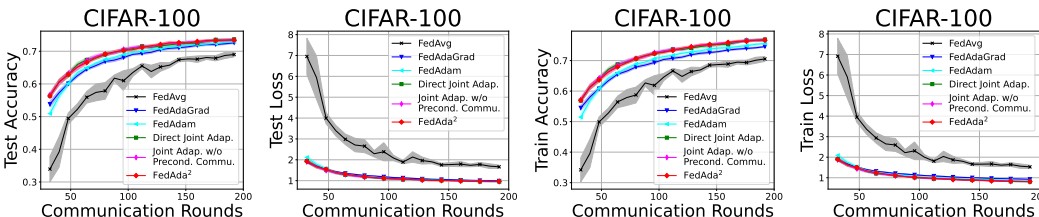

Figure 5: Training and testing accuracies of optimal hyperparameters for CIFAR-100. At each logging step, train/test accuracy and loss evaluation is done over *all* of training and testing data, disjointly, resulting in robust and similar-looking curves. Averaged over 20 random seeds for better convergence. Adaptive optimizer instantiation conventions are identical with Figure 4.

### G.4 FEMNIST DATASET

The FEMNIST dataset (Caldas et al., 2018) extends the MNIST dataset LeCun et al. (1998) to include both digits and letters, comprising 62 unbalanced classes and a total of 805,263 data points. It is specifically designed for federated learning research, featuring a natural, non-IID partitioning of data. Each user in the dataset corresponds to a distinct writer who contributed to the original EMNIST dataset, capturing the individuality of handwriting styles. This user-level segmentation provides a realistic federated learning setting, simulating scenarios where data is distributed heterogeneously across clients. FEMNIST serves as a benchmark for evaluating the performance of federated learning algorithms under non-IID conditions, emphasizing challenges such as personalization and robustness to client heterogeneity.

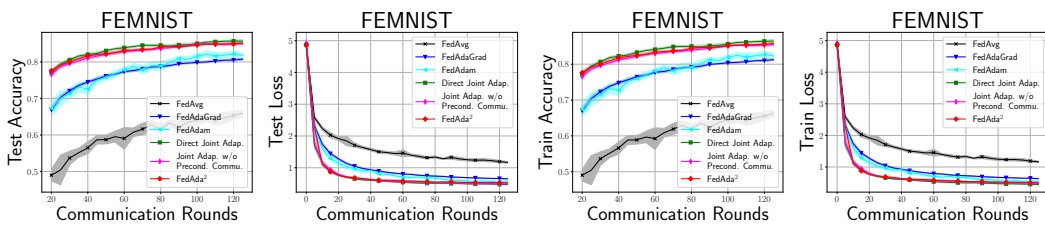

Figure 6: Training and testing accuracies of optimal hyperparameters for FEMNIST, with 0.5% participation (2 clients per round). Averaged over 20 random seeds for clearer convergence. Adaptive optimizer instantiation conventions are identical with Figure 4, where jointly adaptive optimizing paradigms use Adam due to better performance. We see that FedAvg is the least robust, both in terms of stability (i.e., confidence interval region) and final performance. By contrast, adding server-side adaptivity greatly strengthens the performance, and introducing client-side adaptive optimization further enhances the speed of convergence as well as test-time accuracy. We see that removing preconditioner transmission, and compressing client-side gradient statistics to save on-device memory as in `FedAda`$^2$, does not detract from the performance of joint adaptivity.

### G.5 DESCRIPTIONS OF BASELINES

In the original FedAvg algorithm introduced by McMahan et al. (2017), the server-side aggregation is performed without any additional momentum, relying solely on simple averaging. On the other hand, algorithms like FedAdaGrad and FedAdam represent examples of server-only adaptive approaches (Reddi et al., 2021), where the server employs adaptive optimizers such as AdaGrad or Adam instead of vanilla averaging. We note that server-only adaptive frameworks such as FedAdam and FedAdaGrad are optimizer-specific instantiations of FedOpt (Reddi et al., 2021), a competitive framework that has been utilized in recent works to develop leading applications (e.g., by Google Deepmind to develop DiLoCo (Douillard et al., 2024; Liu et al., 2024; Jaghouar et al., 2024)). The concept of 'Direct Joint Adaptivity' (Direct Joint Adap.) refers to a training paradigm where the server's adaptive preconditioners are shared with clients during each communication round. An

example of this is the AdaGrad-AdaGrad setup used as a differential privacy baseline in the Stack-Overflow task, where the server-side AdaGrad preconditioners are applied to client-side AdaGrad optimizers, guiding client model updates.

Alternatively, by eliminating the transmission of server-side preconditioners and initializing client-side preconditioners to zero, we derive the 'Joint Adaptivity without Preconditioner Communication' (Joint Adap. w/o Precond. Commu.) baseline, which is more communication-efficient. Further, compressing local preconditioners to align with client memory constraints leads to the development of $\texttt{FedAda}^2$. Thus, $\texttt{FedAda}^2$ and the various baselines can be viewed as logically motivated extensions, incorporating adaptive updates and memory-efficient strategies. We provide comprehensive evaluations of all 15 algorithms (including 12 jointly adaptive methods tailored to each adaptive optimizer, 2 server-only adaptive methods, and 1 non-adaptive method) in Section 6 and in the Appendix G, I.

Below, we include a table to summarize the communication complexity and memory efficiency of $\texttt{FedAda}^2$ and baselines, compared to alternative adaptive frameworks such as MIME or MIMELite (Karimireddy et al., 2021; Ro et al., 2022) (evaluation not included in paper).

Table 2: Comparison of Baselines versus $\texttt{FedAda}^2$ with AdaGrad instantiations. $d$ denotes the model dimensions.

| Method | Joint Adaptivity | Communication | Computation (#gradient calls) | Memory (client) |
|---|---|---|---|---|
| FedAvg | N | 2d | 1 | d |
| FedAdaGrad | N | 2d | 1 | d |
| MIME/MIMELite | N | 5d / 4d | 3/2 | 4d / 3d |
| DJA | Y | 3d | 1 | 2d |
| $\texttt{FedAda}^2$ | Y | 2d | 1 | $1d \sim 2d$ |

For the ViT model for instance, we require just 0.48% memory to store the second moment EMA compared to the full gradient statistic during preconditioning when using SM3. The variance between 1d and 2d in the $\texttt{FedAda}^2$ 'Memory (client)' column depends on the instantiation of the client-side memory-efficient optimizer.

## H  HYPERPARAMETER SELECTION

### H.1  HYPERPARAMETERS FOR DP STACKOVERFLOW

We use a subsampling rate of 0.1, for a total of 400 clients and 500 communication rounds. We investigate the setting of noise multiplier $\sigma = 1$, which provides a privacy budget of $(\varepsilon, \delta) = (13.1, 0.0025)$ with optimal Rényi-Differential Privacy (RDP) order 2.0. We sweep over the following hyperparameters:

$$c \in \{0.1, 0.5, 1\},$$
$$\eta_l \in \{0.001, 0.01, 0.1, 0.5, 1\},$$
$$\eta_s \in \{0.001, 0.01, 0.1, 0.5, 1\},$$
$$\tau_l \in \{10^{-7}, 10^{-5}, 10^{-3}\},$$
$$\tau_s \in \{10^{-7}, 10^{-5}, 10^{-3}\},$$

where $c$ is the gradient clip value. Here, $\eta_l, \eta_s$ indicates the client and server learning rates, while $\tau_l, \tau_s$ represents their respective adaptivity parameters. In the case of singular adaptivity, we ignore the irrelevant terms (i.e. client adaptivity parameter for FedAdaGrad). For FedAvg only, we select best hyperparameters using the expanded local learning rate grid

$$\eta_l \in \{0.001, 0.01, 0.1, 0.5, 1, 5, 20, 40, 80, 160\}.$$

The optimal hyperparameters are summarized in Table 3, which were chosen based on optimal test accuracy over a running average of the last 10 logged datapoints. In Figure 3 (bottom), we see that adaptive optimization on either the client or server induces varying model training dynamics. Notably, we see in our experiments that for this privacy budget, removing preconditioners from

jointly adaptive systems supercedes the performance of direct joint adaptivity. Compressing client adaptive preconditioning (FedAda$^2$) reduces the performance slightly, but still performs the best among all other baselines.

Table 3: Best performing hyperparameters for DP StackOverflow with $\sigma = 1$

|  | FedAvg | FedAdaGrad | Direct Joint Adap. | Joint Adap. w/o Precond. Commu. | FedAda$^2$ |
|---|---|---|---|---|---|
| $c$ | 1.0 | 0.1 | 0.5 | 0.5 | 0.1 |
| $\eta_s$ | N/A | 1.0 | 1.0 | 1.0 | 1.0 |
| $\eta_l$ | 20.0 | 1.0 | 1.0 | 0.1 | 0.1 |
| $\tau_s$ | N/A | 1e-3 | 1e-3 | 1e-5 | 1e-5 |
| $\tau_l$ | N/A | N/A | 1e-3 | 1e-3 | 1e-3 |

## H.2 HYPERPARAMETERS FOR IMAGE DATASETS

For all ViT experiments, images were resized to $224 \times 224$ pixels, and the client optimizer employed a linear learning rate warm-up, increasing from $0$ to the final value over the first 10 local backpropagation steps. The local batch size was consistently set to 32 across all datasets used in this paper. Due to better empirical performance, Adam was selected as the main optimizer strategy for ViT fine-tuning against the image datasets. We utilized prior work (Reddi et al., 2021) as well as small-scale experiments regarding server-only adaptivity to guide the selection of the momentum parameters $\beta_1 = 0.9$, $\beta_2 = 0.999$ for server Adam. The identical parameters were selected for client Adam, and better choices may exist for either the server or client. In order to determine suitable learning rates and adaptivity parameters, we conduct extensive hyperparameter sweeps using a two-step procedure.

**(Step 1)** The first step involved a symmetric sweep over the values

$$\eta_l \in \{0.001, 0.01, 0.1, 0.5, 1, 5, 20\},$$
$$\eta_s \in \{0.001, 0.01, 0.1, 0.5, 1, 5, 20\},$$
$$\tau_l \in \{10^{-9}, 10^{-7}, 10^{-5}, 10^{-3}\},$$
$$\tau_s \in \{10^{-9}, 10^{-7}, 10^{-5}, 10^{-3}\}.$$

Similar to the StackOverflow case, $\eta_l, \eta_s$ indicates the client and server learning rates, while $\tau_l, \tau_s$ represents their respective adaptivity parameters. For FedAvg only, we probe over the expanded grid

$$\eta_l \in \{0.001, 0.01, 0.1, 0.5, 1, 5, 20, 40, 80, 160, 320\}.$$

**(Step 2)** Based on the sweep results over all 10 algorithm and dataset combinations, a second asymmetric search was launched over the most promising hyperparameter regions, which probed over the following:

$$\eta_l \in \{10^{-6}, 10^{-5}, 10^{-4}, 10^{-3}, 10^{-2}, 10^{-1}\},$$
$$\eta_s \in \{10^{-7}, 10^{-6}, 10^{-5}, 10^{-4}, 10^{-3}, 10^{-2}\},$$
$$\tau_l \in \{10^{-7}, 10^{-5}, 10^{-3}, 10^{-1}, 1\},$$
$$\tau_s \in \{10^{-12}, 10^{-11}, 10^{-10}, 10^{-9}, 10^{-5}\}.$$

Afterwards, the best performing hyperparameters were selected. For FedAvg only, the final grid increased additively by $10^{-3}$ from $10^{-3}$ to $10^{-2}$, then by $10^{-2}$ onward until the largest value $10^{-1}$. That is, we sweep over the following:

$$\eta_l \in \{0.001, 0.002, 0.003, \ldots, 0.009, 0.01, 0.02, \ldots, 0.09, 0.1\}.$$

For server-only adaptivity or FedAvg, any irrelevant hyperparameters were ignored during the sweep. In Tables 4 and 5, we summarize the best performing learning rates and adaptivity parameters. In this subsection, any notion of adaptivity in jointly adaptive systems refers to the Adam optimizer, and 5 local epochs were taken prior to server synchronization. Full fine-tuning indicates that the entire net was unfrozen after replacement of the linear classification layer. For FedAdaGrad, full fine-tuning, Step 2 utilized an expanded hyperparameter grid search due to poor performance.

Table 4: Server/Client Learning Rates $\eta_s/\eta_l$

|  | FedAvg | FedAdaGrad | FedAdam | Direct Joint Adap. | Joint Adap. w/o Precond. Commu. | FedAda$^2$ |
|---|---|---|---|---|---|---|
| FEMNIST | N/A / 8e-3 | 1e-4 / 1e-3 | 1e-4 / 1e-3 | 1e-3 / 1e-3 | 1e-3 / 1e-3 | 1e-3 / 1e-3 |
| CIFAR-100 | N/A / 1e-1 | 1e-2 / 1e-5 | 1e-3 / 1e-3 | 1e-3 / 1e-2 | 1e-3 / 1e-2 | 1e-3 / 1e-2 |
| GLD-23K | N/A / 0.04 | 1e-2 / 1e-2 | 1e-3 / 1e-2 | 1e-3 / 1e-2 | 1e-3 / 1e-2 | 1e-3 / 1e-2 |
| GLD-23K (Full) | N/A / 0.02 | 1e-4 / 1e-2 | 1e-4 / 1e-2 | 1e-4 / 1e-4 | 1e-4 / 1e-2 | 1e-4 / 1e-4 |

Table 5: Server/Client Adaptivity Parameters $\tau_s/\tau_l$

|  | FedAvg | FedAdaGrad | FedAdam | Direct Joint Adap. | Joint Adap. w/o Precond. Commu. | FedAda$^2$ |
|---|---|---|---|---|---|---|
| FEMNIST | N/A / N/A | 1e-7 / N/A | 1e-7 / N/A | 1e-5 / 1e-7 | 1e-5 / 1e-7 | 1e-5 / 1e-7 |
| CIFAR-100 | N/A / N/A | 1e-10 / N/A | 1e-5 / N/A | 1e-5 / 1.0 | 1e-5 / 1.0 | 1e-5 / 1.0 |
| GLD-23K | N/A / N/A | 1e-5 / N/A | 1e-5 / N/A | 1e-5 / 0.1 | 1e-5 / 0.1 | 1e-5 / 0.1 |
| GLD-23K (Full) | N/A / N/A | 1e-2 / N/A | 1e-5 / N/A | 1e-5 / 1e-3 | 1e-5 / 1 | 1e-5 / 1e-3 |

**Hyperparameter Sweep for FEMNIST.** The setup was almost analogous to above. The only difference is that due to limited resources in **(Steps 1-2)**, we swept over the grid

$$\eta_l \in \left\{10^{-4}, 10^{-3}, 10^{-2}, 10^{-1}\right\},$$
$$\eta_s \in \left\{10^{-4}, 10^{-3}, 10^{-2}, 10^{-1}\right\},$$
$$\tau_l \in \left\{10^{-7}, 10^{-5}, 10^{-3}\right\},$$
$$\tau_s \in \left\{10^{-7}, 10^{-5}, 10^{-3}\right\}.$$

For FedAvg only, we utilized the expanded learning rate grid

$$\eta_l \in \left\{0.001, 0.002, 0.003, \ldots, 0.009, 0.01, 0.02, \ldots, 0.09, 0.1\right\}.$$

**Hyperparameters for varying client resources, GLD-23K.** Analogous sweeps as in **(Step 1)** above for the limited and sufficient client resource settings (locally training over 1, 20 local epochs prior to server synchronization) were taken. For the constrained setting, there were no changes to the **(Step 2)** grid. In the abundant setting, the modified final search space for adaptive methods was

$$\eta_l \in \left\{10^{-6}, 10^{-5}, 10^{-4}, 10^{-3}, 10^{-2}, 10^{-1}\right\},$$
$$\eta_s \in \left\{10^{-3}, 10^{-2}, 10^{-1}, 1, 4, 16, 32\right\},$$
$$\tau_l \in \left\{10^{-7}, 10^{-5}, 10^{-3}, 10^{-1}, 1\right\},$$
$$\tau_s \in \left\{10^{-12}, 10^{-11}, 10^{-10}, 10^{-9}, 10^{-5}\right\},$$

and the optimal hyperparameters are summarized in Table 6.

Table 6: Hyperparameters for GLD-23K under restricted/sufficient client resource settings

|  | FedAvg | FedAdaGrad | FedAdam | Direct Joint Adap. | Joint Adap. w/o Precond. Commu. | FedAda$^2$ |
|---|---|---|---|---|---|---|
| $\eta_s$ | N/A / N/A | 1e-2 / 1e-2 | 1e-3 / 1e-3 | 1e-3 / 1e-3 | 1e-3 / 1e-3 | 1e-3 / 1e-3 |
| $\eta_l$ | 7e-2 / 1e-2 | 1e-2 / 1e-2 | 1e-1 / 1e-2 | 1e-2 / 1e-3 | 1e-2 / 1e-3 | 1e-1 / 1e-3 |
| $\tau_s$ | N/A / N/A | 1e-9 / 1e-7 | 1e-5 / 1e-7 | 1e-5 / 1e-7 | 1e-5 / 1e-7 | 1e-5 / 1e-7 |
| $\tau_l$ | N/A / N/A | N/A / N/A | N/A / N/A | 1e-3 / 1e-1 | 1e-3 / 1e-1 | 1e-1 / 1e-1 |

## H.3 COMPUTE RESOURCES

Experiments were performed on a computing cluster managed by Slurm, consisting of nodes with various configurations. The cluster includes nodes with multiple GPU types, including NVIDIA RTX 2080 Ti, A40, and H100 GPUs. The total compute utilized for this project, including preliminary experiments, amounted to approximately 6 GPU-years.

## I ADDITIONAL EXPERIMENTS

### I.1 DYNAMICS OF HETEROGENEOUS CLIENT-SERVER ADAPTIVITY

In Figure 7, we display the effects of heterogeneous client-server adaptivity in the setting of ViT fine-tuning over GLD-23K. All hyperparameter sweeps were done over the following grid:

$$
\begin{aligned}
\eta_l &\in \left\{ 10^{-4}, 10^{-3}, 10^{-2}, 10^{-1} \right\}, \\
\eta_s &\in \left\{ 10^{-4}, 10^{-3}, 10^{-2}, 10^{-1} \right\}, \\
\tau_l &\in \left\{ 10^{-7}, 10^{-5}, 10^{-3}, 10^{-1}, 1 \right\}, \\
\tau_s &\in \left\{ 10^{-7}, 10^{-5}, 10^{-3}, 10^{-1}, 1 \right\}.
\end{aligned}
\tag{22}
$$

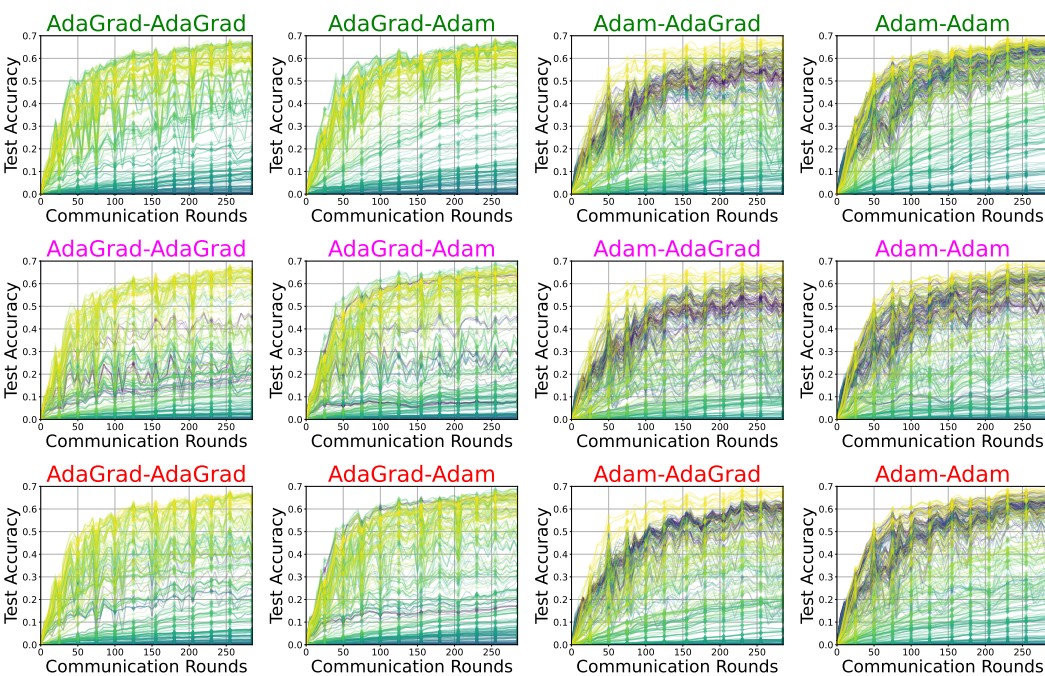

Figure 7: Each test accuracy is color-coded and ranked based on the final test loss, and lighter colors indicate lower loss. Algorithm title colors are also consistent with labels; green for Direct Joint Adaptivity (top), magenta for Joint Adaptivity without Preconditioner Transmission (middle), and red for FedAda² (bottom). Title ordering indicates server- and client-side optimizers, respectively; i.e. AdaGrad-Adam uses server AdaGrad and client Adam. In the case of Direct Joint Adaptivity with heterogeneous client-server optimizers, we transmit the *mismatched* server-side preconditioner to the client, which to our surprise demonstrates considerable performance. For FedAda², we add SM3 compression to the client-side optimizer after zero initialization of the local preconditioner.

### I.2 EFFECT OF DELAYED UPDATES

Similar to Figure 7, we demonstrate the effects of delayed updates in Figure 8. Hyperparameter configuration for delayed updates is identical to Figure 3 (b), except that client-side preconditioner updates are delayed. Hyperparameter sweeps were done over the following grid:

$$
\begin{aligned}
\eta_l &\in \left\{ 10^{-4}, 10^{-3}, 10^{-2}, 10^{-1} \right\}, \\
\eta_s &\in \left\{ 10^{-4}, 10^{-3}, 10^{-2}, 10^{-1} \right\}, \\
\tau_l &\in \left\{ 10^{-3}, 10^{-1}, 1 \right\}, \\
\tau_s &\in \left\{ 10^{-5}, 10^{-3}, 10^{-1} \right\}.
\end{aligned}
$$

We see that delaying the computation of the preconditioners does not significantly degrade the performance.

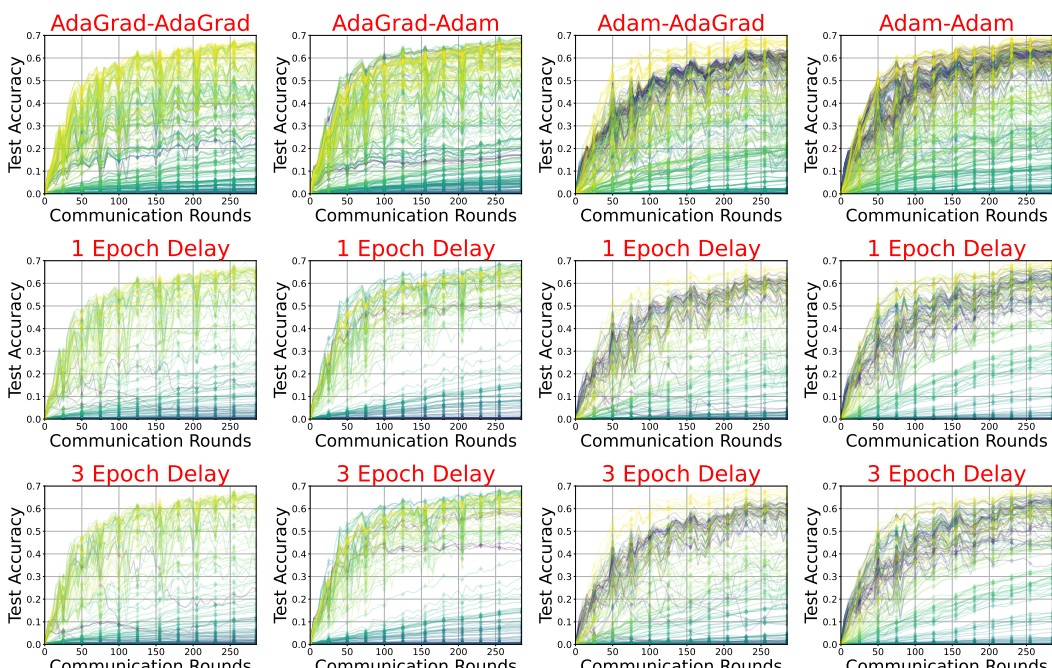

Figure 8: After updating preconditioners per every local backpropagation step for the first client epoch, preconditioners are periodically frozen for the next 1 (middle), 3 (bottom) epochs, respectively, for each communication round. Algorithms are consistent across columns, and the top row is identical to the `FedAda`$^2$ results in Figure 7 with hyperparameter sweep (22).

