# OpenReview forum: "Efficient Adaptive Federated Optimization"
_ICLR.cc/2025/Conference — Submitted to ICLR 2025_

### Official Review · Reviewer_gLqi · 2024-10-23

**Soundness:** 3
**Presentation:** 2
**Contribution:** 3
**Rating:** 6
**Confidence:** 3

**Summary:**

This work considers efficient adaptive optimization in FL and proposes FedAda2 to optimize communication efficiency and memory costs. The theoretical and empirical results demonstrate the benefits of FedAda2.

**Strengths:**

1) This work proposes the algorithm FedAda2 to avoid extra communication costs and reduce the memory cost on device.

2) This work provides comprehensive convergence analysis to show the importance of client adaptivity and also proves the convergence of FedAda2 for general non-convex function.

**Weaknesses:**

1) The appendix is missing, and it is hard to check the experimental setup and convergence analyses.

2) The theoretical analysis in Section 3 is mainly for online strongly-convex functions, which limits their contribution for most real-world applications.

3) The current manuscript lacks a comparison (possibly a table) of the communication costs and memory costs between FedAda2 and existing algorithms, which can significantly increase the readability of this paper.

4) The baselines in experiments seem to be confusing; it is better to distinguish existing methods and FedAda2 with ablations.

**Questions:**

1) As the abstract mentioned that this algorithm is designed for large-scale, cross-device federated environments, I wonder what the number of clients in total and per round in the experiments are.

2) On DP stackoverflow task, FedAda2 achieves worse performance than Joint Adap. w/o Precond. Commu. What are comparisons between their (and also other baselines') memory costs?

---

> ### Author Response · Authors · 2024-11-20
>
> We thank reviewer gLqi for their feedback.
>
> **[Missing Appendix?]** In the original submission, we manually split the main paper and the appendix, and submitted the full paper along *with the appendix and source code* as a single supplementary zip file, per ICLR guidelines. We understand that this has led to the impression of a missing appendix, and would like to gently ask for a re-evaluation of the work given this context. In the revision, we directly uploaded the full paper as well as the appendix without splitting them for reviewers’ convenience.
>
> **[Convexity Assumptions]** We use convex, online objectives in Section 3 only for motivation –but for the convergence of the proposed algorithm $FedAda^2$, we use the more general L-smoothness setting in Section 5.
>
> **[Table of $FedAda^2$ and Baselines + Ablations]** The comparison table is given in Appendix G.5, with additional discussions. Regarding ablations, $FedAda^2$ and baselines can be viewed as natural extensions or ablations of one another, which incorporate local adaptive updates and memory-efficient strategies. That is, Direct Joint Adaptivity refers to a training paradigm where the server’s adaptive preconditioners are shared with clients during each communication round. Eliminating the transmission of server-side preconditioners and initializing client side preconditioners to zero gives ‘Joint Adaptivity without Preconditioner Communication’, which is more communication-efficient. Further compressing local preconditioners (e.g., SM3) to align with client memory constraints leads to the development of $FedAda^2$.
>
>
> **[Questions.]**
>
> **[Q1. Empirical Setup]** The empirical setup, including client count, synchronization rounds, and hyperparameter selection, is detailed in Appendix G, H (e.g., Table 1). Generally, we chose the device number to be large in relation to the compute we had available. For Stackoverflow (G.1), there are 400 devices, where each device is a site user. For CIFAR-100 (G.3), we used 1000 Devices, where highly non-IID client data distributions were induced via LDA partitioning (a topic modeling method [1]) with $\alpha = 0.001$. GLD-23K (G.2) consists of 233 real-world photographers, taking pictures of landmarks. The participation fractions were quite low, e.g., 0.01 for GLD-23K (only 2 clients per round) to model challenging, real-world non-IID settings.
>
> **[Q2. $FedAda^2$ and Baseline Comparisons]** The communication/memory comparisons between $FedAda^2$ and other baselines are provided in Appendix G.5 in the original submission. For convenience, we also provide it here.
>
> **Table 1: Comparison of Baselines versus $FedAda^2$ with AdaGrad Instantiations.**
>
> | **Method**           | **Joint Adaptivity** | **Communication** | **Computation (#gradient calls)** | **Memory (client)** |
> |----------------------|----------------------|-------------------|-----------------------------------|---------------------|
> | FedAvg               | N                    | $2d$              | 1                                 | $d$                 |
> | FedAdaGrad           | N                    | $2d$              | 1                                 | $d$                 |
> | MIME/MIMELite        | N                    | $5d / 4d$          | $3/2$                             | $4d / 3d$           |
> | DJA                  | Y                    | $3d$              | 1                                 | $2d$                |
> | $FedAda^2$  | Y                    | $2d$              | 1                                 | $1d \sim 2d$                |
>
> For the ViT model for instance, we require just 0.48% memory to store the second moment EMA compared to the full gradient statistic during preconditioning, when SM3 is used. The variance between 1d and 2d in the $FedAda^2$ “Memory (client)” column depends on the instantiation of the client-side memory-efficient optimizer. Here, $d$ denotes the model dimensions.
>
> [1] Blei et al., Latent dirichlet allocation. JMLR, 2003.

---

> > ### Comment · Reviewer_gLqi · 2024-11-23
> >
> > I have rechecked the appendix and most of my questions are solved. I have raised my score accordingly.

---

### Official Review · Reviewer_3XLN · 2024-10-24

**Soundness:** 1
**Presentation:** 3
**Contribution:** 1
**Rating:** 3
**Confidence:** 4

**Summary:**

This paper highlights the importance of adaptivity in the FL framework, and propose a FedAda2 paradigm to implement the advantages of both global and local adaptivity, reduce communication bits through Zero Local Preconditioner Initialization, while allowing local clients to use different adaptive optimizers. The author provides both theoretical analysis and empirical research to validate the effectiveness of the framework.

After the review, several issues remain, including conflicts in the assumptions made in the analysis, the relatively low technical contribution of the proposed methods, and the potential training divergence caused by naive aggregation. Additionally, the experimental results are not presented clearly, making it difficult to see the performance improvements of the framework in the chosen settings.

**Strengths:**

1. Discussing the advantages of adaptive methods in long-tailed data is very interesting, as it highlights why we should use adaptive learning rates in FL.

2. The paper is well organized, making the reading relatively easy.

**Weaknesses:**

1. Some statements in the introduction are inaccurate. For example, "To the very best of our knowledge, there are no known convergence results on joint adaptive federated optimization in the general convex or non-convex setting". Is the joint adaptive federated optimization a special setup? Based on the analysis of Reddi et al., most papers seem to provide convergence analysis in either convex or non-convex settings. I don’t understand what scenario this refers to, as there is no corresponding explanatory content in the context.

2. After proposing the gradient condition under the long-tailed data distribution in Definition 1, the authors later adopt the assumption of Lipschitz continuity to constrain the bounded gradient in subsequent proofs. These two assumptions seem to conflict with each other. The authors do not carefully address the inconsistency between the assumptions. This could lead to another issue: whether an function satisfying Lipschitz continuity would still experience divergence during training with long-tailed data? I believe this is important. The current version lacks discussions about their relationships, which fails to demonstrate the advantages and significance of local adaptivity.

3. Zero Local Preconditioner Initialization is proposed to save communication bits. However, previous works seem to have already achieved this without the need to additional communications. [1] has been pointed out that using local historical preconditioner can also achieve good convergence and performance in experiments. This approach does not require knowledge of the global preconditioner. If the proposed technique is merely intended to reduce communication, its contribution is insignificant.

[1] Chen, Xiangyi, Xiaoyun Li, and Ping Li. "Toward communication efficient adaptive gradient method." Proceedings of the 2020 ACM-IMS on Foundations of Data Science Conference. 2020.

4. [1] also mentions a critical issue: when using a local adaptive optimizer, performing adaptive learning rate based on the local preconditioner might lead to divergence. They provide an example to illustrate the issue of divergence, attributing the cause of the divergence to the inconsistency of the local preconditioner. It is worth noting that pure global adaptivity does not have this issue, as local nodes use simple SGD. However, since the FedAda2 method proposed in this paper introduces local adaptivity, computing a simple average at the global server may encounter divergence issues. Specifically, the calculation of $\Delta_t$ could potentially approach infinity in some bad case. This leads to the conclusion that the framework presented in this paper seems unreliable.

5. The experimental improvements appear to be minimal. I do not see any significant enhancement in accuracy reflected in the figures. Can authors provide the table comparison of the test accuracy? In certain results, the performance of FedAda2 seems even worse. In most cases, it overlaps with other baselines, making it difficult for me to accurately discern its experimental effectiveness.

**Questions:**

The discussion on long-tailed data of the paper is quite interesting. However, the latter part regarding the proposed solution and its explanatory analysis seems to have issues; for details, please refer to the weaknesses section. In the experimental part, it is challenging to differentiate the performance improvements of the proposed framework. Currently, the main issues include the feasibility and effectiveness of the proposed solution. The current solution may lead to divergence due to its naive aggregation approach, and the techniques aimed at saving communication in the analysis do not seem very significant.

---

> ### Author Response · Authors · 2024-11-20
>
> We thank reviewer 3XLN for their review.
>
> **[Inaccurate Statements in Introduction]** We respectfully disagree with the reviewer’s statement. The “explanatory context” the reviewer requests is given in Section 1, lines 033-055, Section 6, lines 374-387, etc. To summarize, there is a server-side optimizer and client-side optimizer in federated learning. Adaptivity can be employed in either the server-side or the client-side, where joint adaptivity (consisting of global and local adaptive updates) has been shown to play a pivotal role in accelerating convergence and enhancing accuracy in prior works. However, to our knowledge, there are no known convergence results in the jointly adaptive setup. The vast majority of work, including the work by Reddi et al that the reviewer cites, include proofs for server or client-only adaptive setups, e.g., FedAdam, FedAdaGrad, which we have implemented as baselines. In joint adaptivity settings, significant complications arise from the interaction between server- and client-side adaptive updates in the proof, which both leverages historical local gradient information. We would be happy to elaborate further upon request.
>
> **[On “Conflicting” Lipschitz Assumption]** We respectfully disagree that the assumptions the reviewer refers to are conflicting, i.e., the Lipschitzness of the objectives (Section 5) do not conflict with the heavy-tailedness of the stochastic gradients (Section 3). This is because the Lipschitzness is imposed on the expected objective, which is defined as $F_i(x)=\mathbb{E}_{z \sim \mathcal{D}_i}\left[F_i(x, z)\right]$ (line 202). The heavy-tailedness of the gradients are modeled by $z$, which is integrated out. Therefore, an objective satisfying Lipschitzness can indeed “experience divergence during training” due to heavy tailed noise, as $z \sim \mathcal{D}$ is orthogonal to $F_i(x)$ after the expectation. Furthermore, Section 3 was included in order to theoretically study intriguing questions regarding the desirability of adaptive client-side optimization, something another reviewer has asked about. In contrast, the analysis in Section 5 aims to establish standard convergence results in the context of bounded gradients and L-smoothness as standard assumptions (e.g., [2-4]).
>
> **[Using Local Historical Preconditioner as in [1]]** Using local historical preconditioners throughout training is not a viable approach for several real-world, cross-device settings, as noted in Section 4, lines 206-208. One determining property of cross-device federated settings is that the clients are not able to store or maintain ‘states’ across communication rounds [5]. For example, it is unclear that a client will participate in the training more than once, and it is unreasonable for resource-strapped clients (e.g., mobile phones) to maintain optimizer states in memory when not partaking in local updates. Additionally, server-side adaptivity is critical. Thus, we propose the $FedAda^2$ framework for communication and memory-efficient optimization, with rigorous convergence guarantees. We respectfully disagree with reviewer 3XLN that our work is “insignificant”, given the practical nature of this problem and the lack of theoretical convergence results in joint adaptive settings.
>
> [2] Xie et al., Local adaalter: Communication efficient stochastic gradient descent with adaptive learning rates. Workshop on Optimization for Machine Learning, 2020.
>
> [3] Wang et al., Tackling the objective inconsistency problem in heterogeneous federated optimization. NeurIPS, 2020.
>
> [4] Li et al., On the convergence of FedAvg on non-iid data. ICLR, 2020.
>
> [5] Kairouz et al., Advances and open problems in federated learning. Foundations and trends in machine learning, 2021.

---

> ### Author Response · Authors · 2024-11-20
>
> **[Divergence of Client-Side Adaptivity in [1]]** Firstly, the reviewer notes that as $FedAda^2$ uses local adaptivity, “computing a simple average at the global server may encounter divergence issues.” But in fact, FedAda^2 is performing globally adaptive updates, as opposed to “computing a simple average” at the server as the reviewer states, as described in Algorithm 1 (step 12). Hence, the issues with the client-only adaptive optimization the reviewer mentions do not necessarily hold in our setting. Our experiment results (Figures 1~3) in Section 6 validate the performance of $FedAda^2$, and prior works (e.g., [6]) have also shown the performance enhancement of client-side adaptive optimization.
>
> **[Experimental Improvements]** Our goal is to show that joint adaptivity is practicable in real-world cross-device environments via efficient techniques. Therefore, we do not expect memory-efficient optimization to do even better than full-information optimization. The goal of this work is not to achieve the highest accuracy on all datasets, rather, it is to show that the communication- and memory-efficient $FedAda^2$ framework (a) converges theoretically, and (b) does not degrade the performance of expensive jointly adaptive optimization despite information compression. However, despite using approximation techniques, we found that $FedAda^2$ is (a) competitive compared with much more expensive baselines, and (b) sometimes does even better than the more expensive jointly adaptive baseline, "Direct Joint Adaptivity" (e.g., DP-Stackoverflow, Figure 1). When we report accuracies versus bits communicated, $FedAda^2$ widens its lead (Figure 2).
>
> [6] Zhou et al., FedCAda: Adaptive Client-Side Optimization for Accelerated and Stable Federated Learning. Arxiv, 2024

---

> ### Comment · Reviewer_3XLN · 2024-11-20
>
> Thanks for the response and I believe my question has not been effectively resolved for the following perspectives.
>
> **About the assumption**: I greatly appreciate your analysis in sec.3, as it establishes the necessity of adaptive learning rates under label-inconsistent heterogeneity. If you could provide a convergence analysis under a long-tailed data distribution and highlight the advantages of using a dual-adaptive optimizer, I believe it would be a significant contribution. Currently, the theory can only be seen as an extension of FedAdam, and the convergence rate you’ve proven cannot surpass that of FedAdam because it already represents the optimal rate under general assumptions.
>
> **About the assumption conflicts**: Authors use the assumption 2, that the gradient coordinates are bounded. It means the stochastic gradient $g$ is always bounded on each sample $z$, i.e. $\vert \nabla [F_i(x,z)(j)] \vert \leq G$. Therefore, $\Vert g \Vert^2 = \sum_{j=1}^d g_j^2 \leq dG^2$ is bounded. Then we consider $\mathbb{E}\Vert g - \nabla F(x) \Vert^\alpha$. We can take a sample as $\alpha=2$. $\mathbb{E}\Vert g - \nabla F(x) \Vert^2 = \mathbb{E}\Vert g\Vert^2 + \mathbb{E}\Vert \nabla F(x)\Vert^2 - 2\mathbb{E}\langle g, \nabla F(x)\rangle = \mathbb{E}\Vert g\Vert^2 - \Vert \nabla F(x)\Vert^2 \leq \mathbb{E}\Vert g\Vert^2 \leq dG^2$. In fact, it is bounded for all constant $\alpha$. This is the question I raised. These two assumptions seem to conflict. Can the author help me answer this question?
>
> **About the Zero Local Preconditioner Initialization**: I emphasize that it is trivial because past work has already known relevant techniques. As mentioned above, [1] proposed that a local second-order momentum could be re-used. [2] proposed using a constant pre-conditioner as the adaptivity state at each initialization of local rounds (details seen in page 3 in their paper). Obviously, authors only set the constant in [2] to 0 and restart the local optimization state, and there is nothing new about it. Given that the adaptation is similar to the diagonalization estimation of the Fisher matrix, I do not think that the zero initialization in this paper can be regarded as an novel technique.
>
> [1] Toward communication efficient adaptive gradient method
>
> [2] Local Adaptivity in Federated Learning: Convergence and Consistency
>
> **Divergence of the naive global average:** The problem pointed out by [1] is that if naive averaging is used, the merged direction $\Delta$ may be opposite to the gradient. which will cause training divergence. The author's answer is that adaptive correction is used globally, so this problem can be avoided. I would like to ask the author a question: can authors use the reverse gradient to train models? When the direction of the merged $\Delta$ is opposite to the gradient, is it still effective to use adaptive learning rate? My intuition for asking this question comes from the fact that I think the answer to both of the above questions is no. If the author believes that adaptation can correct the wrong direction, can provide me with more intuition or examples to explain why it can work. I am also very interested in this issue and happy to discuss it.
>
>
> I understand the rate of this iclr are generally low, but I hope the author can read my questions carefully and then reply. I will seriously reply to your rebuttal during the whole discussion phase. At present, I think the concerns yet have not been well solved.

---

> ### Author Response · Authors · 2024-11-21
>
> Thank you for your questions.
>
> **[Extending from FedAdam]** We appreciate the reviewer’s enthusiasm on Section 3. We also respectfully argue that the theory cannot be seen as a simple extension of FedAdam, as FedAdam employs SGD on the client-side. Moving to the jointly adaptive client setting introduces significant theoretical challenges, as now expectations do not act linearly on the client gradients due to dependencies on the entire historical gradients of the client, rendering existing proof techniques inapplicable. Additionally, we incorporate memory-efficient local adaptive optimizers in the proof, such as SM3 (e.g., lines 1235-1347), which is significantly different from local SGD that FedAdam uses. The reviewer enquires about a proof of convergence of joint adaptivity under heavy-tailed noise; a step toward being able to achieve this is to first prove the convergence of joint adaptivity under the conventional L-smoothness (Assumption 1) and Bounded Gradients (Assumption 2) assumptions, which has not been done before and is by itself a significant complication in the convergence proof. Extending the proof to infinite-variance heavy-tailed distributions is beyond the scope of this paper, an entirely new paper altogether.
>
> **[About the Assumption Conflicts]** In Section 3, we study infinite-variance heavy-tailed noise to model transformer training dynamics and elucidate the theoretical importance of client-side adaptivity. In Section 5, we assume bounded gradients (Assumption 2) to provide standard convergence results. It is straightforward to see that bounded gradient distributions *cannot* have infinite variance, and these two obviously “conflict”. But our understanding was that the reviewer’s original question was about the Lipschitzness of the gradients of the *expected* objective $F_i(x)$ (Assumption 1), which does not conflict with the heavy-tailed assumption. As noted in our response above, Section 3 was included in order to theoretically study intriguing questions regarding the desirability of adaptive client-side optimization, something another reviewer has asked about. In contrast, the analysis in Section 5 aims to establish standard convergence results in the context of bounded gradients and L-smoothness as standard assumptions (e.g., [3-4]), which in our view are essential to theoretically establishing the practicability of a proposed framework such as $FedAda^2$. Therefore, Sections 3 and 5 clearly serve different purposes, as stated in our paper. We hope the confusion has been clarified.
>
> **[About Zero Local Preconditioner Initialization]** We respectfully disagree that our paper is "trivial" for the following reasons.
>
> **(a)** The work [1] proposes that local preconditioners can be re-used and accumulated across communication rounds. As mentioned in our rebuttal above to reviewer 3XLN, maintaining local historical preconditioners throughout training is not a viable approach for several real-world, cross-device settings, as noted in Section 4, lines 206-208. One determining property of cross-device federated settings is that the clients are not able to store or maintain ‘states’ across communication rounds [5]. For example, it is unclear that a client will participate in the training more than once, and thus it is unreasonable for resource-strapped clients (e.g., mobile phones) to maintain optimizer states in memory when not partaking in local updates. Additionally, [1] studies only the cross-silo setting, where the maintenance of client preconditioners is not an issue. Therefore, we respectfully would like to argue that [1] does not render our work “trivial” or “insignificant”.
>
> **(b)** We are aware that the paper [2], which we have cited, empirically uses zero preconditioner initialization. However, [2] does not have a proof for convergence of jointly adaptive setups under this setting. In Theorems 6, 20, and 25, we not only provide convergence bounds for server-adaptive and client SM3-adaptive setups under zero preconditioner initialization, but we also extend these results to provide a rigorous convergence guarantee for the $FedAda^2$ framework in the fullest generality, including a class of adaptive methods and using client-specific optimizers (Appendix D, Theorem 25).
>
> **(c)** Additionally, zero preconditioner initialization is only one of the multiple techniques we utilize for $FedAda^2$. Although we have relegated details to the Appendix for interested readers due its technicality, the elements of SM3 (e.g., lines 1235-1347) and delayed preconditioner updates (e.g., lines 1269-1273, 1300-1361, and Appendix I.2) needed to be incorporated into the proof, which was non-trivial.

---

> ### Author Response · Authors · 2024-11-21
>
> **[Comparisons Between Our Work and [1]]** The reviewer cites [1] closely, and we compare [1] with ours.
>
> **(a)** We remove the need for preconditioner maintenance on the client-side, in contrast to [1]. For example, in our experiments for the FEMNIST dataset, only 2 writers out of the 500 clients participate in training each round, which is done for 131 global synchronization rounds until convergence. It is unlikely that any given client will participate more than a few times in such realistic, non-IID settings, thus it is not reasonable for clients to to maintain preconditioners when not directly participating in updates.
>
> **(b)** We show convergence for general joint adaptive methods with local updates, and we remove the aggregation of preconditioners on the server-side for communication efficiency. The algorithm in [1] must transmit preconditioners for aggregation and incurs significant communication complexity, and additionally, [1] does not use joint adaptivity.
>
> **(c)** We incorporate local memory-efficient compression, and study the more general cross-device setting in which only a fraction of the clients participate.
>
> **[Benefits of Adaptive Optimizers on the Client-Side]** The reviewer has expressed concerns about naive local AMSGrad (an algorithm proposed in [1]), because the local preconditioners may lead to divergence. We agree this can be true in specifically constructed negative examples (e.g., using AMSGrad locally and setting $\beta_1=0$, $\beta_2 = 0.5$ with just 1 client update, and choosing an initial condition adversarial to a particularly designed 1-dimensional objective as in [1]). In light of such potential challenges of local adaptivity, our goal is to come up with a theoretical framework that avoids divergence while enjoying the benefits of joint adaptivity, in an efficient way. We establish convergence results of $FedAda^2$ in Section 5, which also covers scenarios in which participating clients use different local optimizer instantiations (hence the name “Blended Optimization” in Appendix D), which would help to mitigate adversarially constructed negative examples. Our work is different to the negative algorithm presented in [1], and one cannot directly attribute the divergence to $FedAda^2$. Our experiment results (Figures 1~3) in Section 6 validate the effectiveness of $FedAda^2$, and prior works (e.g., [6]) have also shown the performance enhancement of client-side adaptive optimization.
>
> We welcome any additional questions or feedback.
>
> [1] Chen et al., Toward communication efficient adaptive gradient method. Proceedings of the 2020 ACM-IMS on Foundations of Data Science Conference, 2020.
>
> [2] Wang et al., Local Adaptivity in Federated Learning: Convergence and Consistency. Arxiv, 2021.
>
> [3] Xie et al., Local adaalter: Communication efficient stochastic gradient descent with adaptive learning rates. Workshop on Optimization for Machine Learning, 2020.
>
> [4] Wang et al., Tackling the objective inconsistency problem in heterogeneous federated optimization. NeurIPS, 2020.
>
> [5] Kairouz et al., Advances and open problems in federated learning. Foundations and trends in machine learning, 2021.
>
> [6] Zhou et al., FedCAda: Adaptive Client-Side Optimization for Accelerated and Stable Federated Learning. Arxiv, 2024

---

> > ### Comment · Reviewer_3XLN · 2024-11-21
> >
> > Thanks for the authors' further rebuttal. After reading the feedback, I believe the two main issues at present may not be resolved by the current submission.
> >
> > **About the assumption conflicts**: It seems I have consistently pointed out that the first assumptions about long-tailed data conflict with the Lipschitz continuity of objective $F$, i.e. the bounded gradient assumption. I have not discussed the smoothness assumption, i.e., the Lipschitz continuity of the gradient $\nabla F$. Given that the authors' response acknowledges the conflict between assumption 2 and the previously constructed long-tailed data assumption, I have no further questions on this matter.
> >
> > **About the naive average**: The purpose of referencing paper [1] is not to let authors compare the proposed FedAda2 with their proposed method, but rather to highlight that the paper provides a sufficient evidence of the naive averaging's unreliability. This means that aggregating without any modifications carries significant risks. Given that the authors' response confirms my previous intuition that using adaptation on the reverse gradients may still hinder training, I have no further questions on this point.
> >
> > The two techniques claimed in the paper to reduce communication—zero initialization and naive aggregation—are either already existing techniques or carry significant risks. In fact, the main reason why researchers avoid naive aggregation from local adaptivity is because it is unreliable. I am unsure why the authors chose to design and propose this solution despite being aware of its unreliability. Therefore, the core contribution of this paper has some flaws. Since assumption 2 conflicts the validity of the long-tailed data assumption, the subsequent theoretical analysis no longer holds the analytical advantage for addressing heterogeneious long-tailed data. Additionally, I hope the authors do not present zero initialization as a technique proposed in this paper, as it is not an original contribution. The authors mentioned a series of variants of such schemes, and I believe that if the authors' innovation lies in the design of these variants, they should emphasize and present these variants in the main text, rather than focusing on the zero initialization technique. Based on the current discussion, I will maintain my current score.

---

> > > ### Author Response · Authors · 2024-11-23
> > >
> > > Thanks for your comments. We are glad to hear that some confusion has been addressed, and we provide some responses below.
> > >
> > > It is possible that “naive aggregation” alone may not perform well for adaptive optimization, which can in fact be one of the contributions of our work. We come up with a simple and effective framework that addresses such challenges both theoretically and empirically. The proposed $FedAda^2$ framework is a different algorithm from the naive aggregation in [1]. Our key message is that joint adaptivity can be effectively realized while greatly reducing the communication bottleneck and client resource requirements which have conventionally plagued jointly adaptive optimization. We prove that $FedAda^2$ converges under a broad class of adaptive methods, and empirically show that it has better tradeoffs between systems efficiency (communication and memory costs) and model utilities.
> > >
> > > Bounded gradient distributions clearly cannot have infinite (heavy-tailed) variance. We use different assumptions to prove different theorems that are of interest in various aspects of the studied problem. We never claim to show federated optimization convergence under heavy-tailed gradient noise in this work; that is not the focus of our work, and would be an entirely new paper altogether. Our work is on establishing communication-efficient and on-device memory-compressed federated learning. Section 3 provides a regret analysis for local adaptive methods under heavy-tailed gradients just to motivate the problem (e.g., potential benefits of local adaptivity). Section 5 presents the main convergence bounds for $FedAda^2$ under the standard assumptions of smoothness and bounded gradients, which is standard in other works when proposing new frameworks. As stated in our paper, these two sections serve different purposes.
> > >
> > > As noted in our responses above, we have cited prior works that explore zero preconditioner initialization in the original submission. However, existing literature lacks comprehensive theoretical understandings for this setting (e.g., convergence behavior of *efficient* joint adaptive optimization). We provide rigorous guarantees for a class of optimizer instantiations in the $FedAda^2$ framework, while allowing for client on-device memory compression (SM3) and compute efficiency techniques (delayed updates) in the proof. Empirically, our extensive experiment results (Figures 1~3) in Section 6 validate the effectiveness of $FedAda^2$ compared with strong baselines, in terms of both performance and efficiency.
> > >
> > > [1] Chen, Xiangyi, Xiaoyun Li, and Ping Li. "Toward communication efficient adaptive gradient method." Proceedings of the 2020 ACM-IMS on Foundations of Data Science Conference. 2020.

---

> > > > ### Comment · Reviewer_3XLN · 2024-11-23
> > > >
> > > > Thanks for the author's responses and summary. I believe I have gained a deep understanding of this work. Personally, I appreciate the content of sec.3 and look forward to theoretical extensions based on this setup. I believe this work has the potential to attract significant attention. Current assumptions conflicts logically make me feel that there are vulnerabilities in this presentation. For instance, sec.3 emphasizes the advantages of adaptivity in training on long-tailed datasets, which is great. However, by assuming bounded gradients, the training on any dataset will be guaranteed to converge. Despite the authors emphasizing that the analyses are independent and sec.3 is merely intended to inspire interests in adaptivity, when I review these two conclusions, I naturally get that gradient clipping can resolve training divergence on long-tailed data (since convergence is already guaranteed), and the local adaptivity is not strictly necessary. This significantly weakens the motivation behind the paper and the contribution of sec.3, which makes this paper looks like a incremental idea.
> > > >
> > > > As for the subsequent algorithm design and theoretical proofs in the current manuscript, they possess a certain degree of originality, but the level of novelty is not particularly outstanding. Without the risks posed by naive aggregation, I would rate the entire paper as borderline. However, my current assessment of this risk remains pessimistic. Although the experiments in this paper show promising results, existing counterexamples consistently highlight this potential for divergence. Actually, a series of works on local adaptivity in FL have all independently highlighted that the aggregation step for second-moment terms might be indispensable. Some even demonstrated that similar dual-adaptive forms like FedAda2 do not always perform well in certain experimental setups. I have reviewed some of the issues raised by other reviewers and will carry these concerns into the next discussion phase with the AC and other reviewers.

---

> > > > > ### Author Response · Authors · 2024-12-03
> > > > >
> > > > > We thank the reviewer for their final comments, and appreciate the opportunity to address their concerns as our discussion concludes. Below, we leave a final response.
> > > > >
> > > > > **[Section 3 and Bounded Gradients]** We appreciate the reviewer’s acknowledgement of our contribution in Section 3. However, we respectfully contest the assertion that “by assuming bounded gradients, the training on any dataset will be guaranteed to converge”. The bounded gradient assumption is standard in the related literature (e.g., [3–5]) , and we do not assume both bounded gradients and heavy-tailed noise at the same time. As the reviewer mentioned, our goal in writing Section 3 was to provide interesting motivations to make our paper more engaging and dynamic. We do not intend to distract from the central message of our work, which is establishing the communication- and memory-efficient jointly adaptive optimization framework $FedAda^2$.
> > > > >
> > > > > **[Algorithm Design and Theoretical Proofs]** We concur that our proofs, which simultaneously allow for various communication- and memory-efficiency techniques within a joint adaptive framework, are original contributions to the field. Additionally, our experimental results effectively validate our theoretical findings with extensively-tuned hyperparameters and repetitive runs with 20 random seeds (Appendix H) across various datasets and models. Notably, our approach, $FedAda^2$, maintains strong performance without the performance degradation typically associated by compressing expensive, jointly adaptive baselines for communication and memory efficiency.
> > > > >
> > > > > **[Counterexamples and Divergence]** The reviewer mentioned that some works have shown the "aggregation step for second-moment terms might be indispensable" (e.g., [1] and [2]). The previous stylized, special counterexamples may not satisfy the assumptions or algorithm design in our work. For example, the toy problem in [1] is not smooth and has several client objectives that are unbounded from below, and the toy problem in [2] employs a one-shot aggregation method where learning rates cannot be decayed, which differs fundamentally from $FedAda^2$. As mentioned in our previous responses, the issues of other *different* algorithms observed in some toy problems cannot be attributed to $FedAda^2$. Such challenges in fact motivate us to design and propose $FedAda^2$ with provable convergence guarantees.
> > > > >
> > > > > Thank you.
> > > > >
> > > > > References:
> > > > >
> > > > > [1] Chen, Xiangyi, Xiaoyun Li, and Ping Li. "Toward communication efficient adaptive gradient method." Proceedings of the 2020 ACM-IMS on Foundations of Data Science Conference. 2020.
> > > > >
> > > > > [2] Wang et al., Local Adaptivity in Federated Learning: Convergence and Consistency. Arxiv, 2021.
> > > > >
> > > > > [3] Reddi et al., Adaptive Federated Optimization. ICLR, 2021
> > > > >
> > > > > [4] Zaheer et al., Adaptive Methods for Nonconvex Optimization. NeurIPS, 2018.
> > > > >
> > > > > [5] Streeter et al., Less Regret via Online Conditioning. Arxiv, 2010.

---

### Official Review · Reviewer_EvCH · 2024-11-02

**Soundness:** 2
**Presentation:** 2
**Contribution:** 2
**Rating:** 3
**Confidence:** 3

**Summary:**

The paper first discusses the importance of adaptive methods on the client side of Federated Learning in terms of heavy-tail. Then it proposes a method that does not require a communication Preconditioner and gives its convergence guarantee in the non-convex case. Experimentally it is shown that their method has a little saving on communication overhead.

**Strengths:**

1.The paper provides the importance of adaptive methods on the client side of the argument.

2.The paper provides the FedAda framework to avoid passing Preconditioner and use more memory-efficient adaptive methods on the client side and provides a proof of convergence.

**Weaknesses:**

1.The thesis is not well structured around the core contribution points **AVOID extra communication cost** and **REDUCE on-device memory** for discussion and experimentation, as shown below:

(1) What exactly is the main impact of Section 3 on the main contribution of the paper? Section 3 seems to only show that client-side adaptation is important and has little impact on the topic of the paper, please explain to Section 3 what specific impact or direct effect HEAVY-TAILED DISTRIBUTION has on the additional communication overhead and on-device memory?

(2) The theory is not analyzed around the core contribution points.Convergence analysis is necessary to some extent, but not very critical in terms of communication overhead and MEMORY. The paper uses SM3 to reduce the memory required in training, and should theoretically or experimentally analyze how much the communication overhead and memory can be reduced relative to other algorithms?

2. The paper doesn't seem to propose a substantial solution, what is the difference between this method and using the adaptive method directly on the client side if the preconditioner is not transmitted during the communication process? It appears that the adaptive optimization method is simply switched to save training memory overhead on the client side. Can the authors provide a theory to refute this?

3. Does using an adaptive optimization update on the client side just drop faster, and what is the difference between using a simple SGD on the client side for more rounds of training?

**Questions:**

Based on the weakness, I have following questions/suggestions.

1. How the heavy-tailed analysis motivates or informs the specific techniques you propose to reduce communication and memory usage.

2. Could you please  provide quantitative comparisons of communication overhead and memory usage between your method and baselines, either theoretically or through additional experiments.

3. Could you please clearly articulate the novelty of your approach compared to simply using adaptive methods on the client side without preconditioner transmission. Specifically, you can highlight any unique aspects of your method beyond just switching optimizers, and explain how these contribute to the overall goals of reducing communication and memory costs.

4. Could you please provide empirical comparisons between your method and SGD with more training rounds, focusing on both convergence speed and final performance, and give an analysis of the trade-offs between using adaptive methods versus more rounds of SGD in terms of communication costs and memory usage?

---

> ### Author Response · Authors · 2024-11-20
>
> We thank reviewer EvCH for their review.
>
> **[Impact of Section 3]** Section 3 is included to answer the other questions the reviewer has asked, such as: Why don’t we just use “a simple SGD on the client side for more rounds of training”? As discussed in Section 3 (lines 114-191), the discussion on heavy-tails justifies the importance of client-side adaptive optimization. In summary, we use Section 3 to motivate the studied problem (joint adaptivity), not the proposed solution. After analyzing the importance of client-side adaptivity, we propose efficient FL frameworks to mitigate the heightened resources induced by adaptive local optimizers, which is $FedAda^2$. We have clarified this further in the revision.
>
> **[On Core Contributions]**
>
>  - Necessity of convergence analysis: We respectfully disagree with the reviewer, and argue that proving convergence via a rigorous convergence analysis (Section 5 and Appendix B~D) is both challenging and essential to theoretically establishing the practicability of a proposed framework.
>
>  - Reduction in communication and memory: We do analyze how communication and memory is reduced relative to other algorithms both theoretically and empirically. For example, in Appendix G.5, we provide a table summarizing communication, computation, and memory complexity across $FedAda^2$ and baselines, with detailed discussions. We provide the table below for convenience. We also refer to Figure 2  (c.f., performance of Direct Joint Adaptivity in Figure 1), which plots accuracies versus bits communicated. There, empirical results show $FedAda^2$ attains significant performance advantages in accuracy and convergence. For additional details, we also provide extensive discussions in Appendix B, lines 1232-1288, Appendix C, 1300-1345, and Section 6, 374-387, etc.
>
> **Table 1: Comparison of Baselines versus $FedAda^2$ with AdaGrad Instantiations.**
>
> | **Method**           | **Joint Adaptivity** | **Communication** | **Computation (#gradient calls)** | **Memory (client)** |
> |----------------------|----------------------|-------------------|-----------------------------------|---------------------|
> | FedAvg               | N                    | $2d$              | 1                                 | $d$                 |
> | FedAdaGrad           | N                    | $2d$              | 1                                 | $d$                 |
> | MIME/MIMELite        | N                    | $5d / 4d$          | $3/2$                             | $4d / 3d$           |
> | DJA                  | Y                    | $3d$              | 1                                 | $2d$                |
> | $FedAda^2$  | Y                    | $2d$              | 1                                 | $1d \sim 2d$                |
>
> For instance, for the ViT model, when using SM3, we require only 0.48% more memory to store the second moment estimates for preconditioning, as opposed to 1x more memory without SM3. The variance between 1d and 2d in the $FedAda^2$ “Memory (client)” column depends on the instantiation of the client-side memory-efficient optimizer and model architectures. Here, $d$ denotes the model dimensions.
>
> **[Difference Between $FedAda^2$ & Adaptive Methods on Client-Side]** We have discussed these differences in Section 6, Appendix B, etc of the original submission. To summarize, ‘Direct Joint Adaptivity’ refers to a training paradigm where the server’s adaptive preconditioners are shared with clients for adaptive optimization during each communication round. By eliminating the transmission of server-side preconditioners and initializing client-side preconditioners to zero, we derive the ’Joint Adaptivity without Preconditioner Communication’ baseline. Further compressing local preconditioners to align with limited client memory constraints leads to the development of $FedAda^2$. Therefore, our framework is quite naturally motivated. However, the convergence analysis (Theorems 6, 20) is very challenging due to the generality of $FedAda^2$. We present convergence proofs for both the SM3-specific instantiation, as well as the fully general version, in Appendix C, D.

---

> ### Author Response · Authors · 2024-11-20
>
> **[Does Adaptive Optimization “Drop” Faster?]** Adaptive optimizers, specifically for training transformer-based models, indeed demonstrate enhanced performance (e.g., [1]). This is why we have included Section 3 to analyze this theoretically in the distributed training setting. For differences between using “SGD... for more rounds” or adaptive optimizers, we refer to Section 3 (lines 113-191), which explains this extensively. For more details, please refer to Appendix A, lines 702-1230 where a self-contained exposition is provided. For experimental results confirming the benefits of using adaptive client-side optimizers, we point to Figures 1~3, c.f., comparison of jointly adaptive paradigms versus baselines.
>
> **[Novelty of Approach]** We note that our work targets joint adaptivity on both the server side and client side. It is motivated by the fact that joint adaptivity may require additional communication cost and “simply using adaptive methods on the client side without preconditioner transmission” will induce significant memory limitations in resource-constrained clients (Section 1, lines 041-055, Appendix G). To mitigate these issues, we propose to simply avoid preconditioner communication, use various techniques for on-device compute/memory-compression, and successfully provide a novel convergence guarantee for such an efficient framework ($FedAda^2$) in the fullest generality. Empirically, we show better tradeoffs between systems efficiency (communication and memory costs) and model utilities.
>
> **[Quantitative Comparisons with SGD & Baselines]** We refer to Figures 1~3 in the main text, where FedAvg (distributed local SGD) has indeed been included as a baseline. In all setups, FedAvg does far poorly in terms of convergence speed and final performance than its adaptive counterparts. For the theoretical analysis behind why this may be the case, it is for this reason that we have included Section 3 that the reviewer asked about. We also give an exposition in terms of communication costs/memory advantage of $FedAda^2$ versus baselines in various portions throughout the text, e.g., Appendix G.5, Section 6, lines 374-387, etc. We also refer to Figure 2 (main text), which plots accuracies versus bits communicated.
>
> We welcome any additional questions or requests for clarifications.
>
> [1] Zhou et al., FedCAda: Adaptive Client-Side Optimization for Accelerated and Stable Federated Learning. Arxiv, 2024

---

> > ### Comment · Reviewer_EvCH · 2024-11-25
> > **Thanks for your response**
> >
> > I appreciate the authors' detailed response and acknowledge their contributions to the theory of joint adaptive convergence.
> > I suggest the authors delve deeper into the convergence results of joint adaptive federated optimization in general convex or non-convex settings, analyzing both theoretical and experimental aspects.
> > Regrettably, I still believe that the paper has not convincingly addressed the scenario of not transmitting optimizer states. Additionally, the experimental setup may not have effectively demonstrated the potential performance of transmitting preconditioners. I'm considering to increase my score to 4.

---

> > > ### Author Response · Authors · 2024-11-27
> > >
> > > Thank you for your questions and comments, and for acknowledging our contributions around joint adaptivity convergence.
> > >
> > > **[Theoretical/Empirical Study of Joint Adaptive Federated Optimization]** We would like to clarify that we do analyze the convergence and performance, from both theoretical and experimental aspects–in Section 5, we use various techniques for on-device compute/memory-compression, and provide a novel convergence guarantee for such an efficient framework ($FedAda^2$) in the fullest generality. In Sections 5, 6 and Appendix G, H, I, we empirically and theoretically evaluate the performance of $FedAda^2$ in terms of metrics such as accuracy, communication costs, memory advantage on datasets Stack Overflow, GLD-23K, CIFAR-100, and EMNIST, under various different settings. Our evaluations also include federated learning under differential privacy. If the reviewer has remaining questions about the experiment results or proof techniques, we are happy to provide additional details.
> > >
> > > **[Potential Performance of Transmitting Preconditioners]** In the empirical results in Section 6, full preconditioner transmission (Direct Joint Adaptivity) consistently shows the best performance compared with non-jointly adaptive baselines, but is the most costly in terms of both communication and on-device memory. We provide detailed discussions regarding the empirical setup in Section 6 (lines 357-387), and Appendix G, H (e.g., lines 2691-2807). Part of our contributions is developing a theoretically principled framework that retains the benefits of direct joint adaptivity, while avoiding extra costs. We are also happy to provide additional discussions.
> > >
> > > Thank you for your time.

---

> ### Author Response · Authors · 2024-11-24
>
> Dear reviewer EvCH,
>
> Thank you for your time in writing your review. We are wondering if our responses have addressed the reviewer's concerns or answered clarification questions. We are happy to provide additional information and discussions.
>
> (In case the Appendix was missed during the initial review, we would like to note that it was submitted as supplementary materials before, and now it has been re-uploaded together with the main submission.)
>
> Thank you.

---

### Official Review · Reviewer_2y9A · 2024-11-07

**Soundness:** 3
**Presentation:** 3
**Contribution:** 3
**Rating:** 8
**Confidence:** 3

**Summary:**

The paper proposes the algorithm FedAda^2 as a strategy for jointly training adaptive optimizers for both the client and server, but also saving on memory and communication costs via a modification in  standard client and server optimization. Usually the preconditioners for the adaptive optimizers are communicated between the clients and server (server aggregates a bunch of client preconditioner matrices, server sends them back). Under FedAda, the server does not send the preconditioners back to the client. Authors have determined that not sending the preconditioners to the client results in very little degradation of performance, but better memory costs and communication costs (don't need to store preconditioners on clients/no need to send them)/ The preconditioners are initialized on the client for local epochs. The authors provide convergence guarantees and demonstrate that their method is superior on CIFAR-100, DP-StackOverflow, and GLD-23K against FedAdaGrad, FedAvg, FedAdam, and other baselines.

**Strengths:**

-Writing is easily understandable and well cited

-Figures are relevant and well-explained.

-Strong related works and literature survey.

-Convergence is in $\mathcal{O}(1/\sqrt{T})$ which is respectable and requires non-aggressive assumptions.

-Clear motivation behind the algorithm, and appears to be relatively novel.

-Good results against baselines (the authors run the same type of experimental settings as [1])

[1] Reddi, S., Charles, Z., Zaheer, M., Garrett, Z., Rush, K., Konečný, J., ... & McMahan, H. B. (2020). Adaptive federated optimization. arXiv preprint arXiv:2003.00295.

**Weaknesses:**

-Experimental results feel like there could be a little more, the authors run CIFAR-100, GLD-23-K, DP Stackoverflow (logistic regression), but their chief comparison [1] runs Stackoverflow NWP, Shakespeare, some EMNIST stuff (all classification tasks). I would be curious to see how FedAda^2 compares on these untested datasets.
.
-The authors mention several other adaptive federated optimization frameworks (MIME, FedOPT) that seem to be doing something similar, but no mention or comparison afterwards. The authors could expand and compare their results against these techniques. It is unclear if this work is a significant improvement in the wider landscape of adaptive federated optimization.

- No empirical evidence or experiments for memory reductions (they use SM3 algorithm which demonstrates excellent performance in [2], and rely on their theoretical work to show it is an improved memory cost). I would personally like to see more concrete memory metrics.

[1] Reddi, S., Charles, Z., Zaheer, M., Garrett, Z., Rush, K., Konečný, J., ... & McMahan, H. B. (2020). Adaptive federated optimization. arXiv preprint arXiv:2003.00295.

[2] Anil, R., Gupta, V., Koren, T., & Singer, Y. (2019). Memory efficient adaptive optimization. Advances in Neural Information Processing Systems, 32.

**Questions:**

See weaknesses

---

> ### Author Response · Authors · 2024-11-20
>
> We thank reviewer 2y9A for their review, and for their constructive comments. We would like to mention two quick notes of clarification:
>
> **[Comparison with FedOPT, MIME]** We have incorporated instantiations of FedOPT into our baselines (such as FedAdam, FedAdaGrad), and evaluated their performance against $FedAda^2$ in Section 6. Regarding MIME, their algorithm is not using joint adaptivity and can be significantly more expensive in terms of both communication and on-device memory.  Thus we opted for the “Direct Joint Adaptivity” baseline instead. We refer to Table 2 in Appendix G.5 for more detailed explanation and comparisons.
>
> **[More Datasets from Reddi et al.,]** Thanks for the suggestion. Our experiments do share some datasets with Adaptive Federated Optimization, Reddi et al., e.g., CIFAR-100, StackOverflow LR (specifically, these are in Figures 1, 3 in their work). We also use more challenging models or datasets such as ViT models and the GLD-23K dataset. GLD-23K is a real-world 203-class dataset where each client is a real-life photographer, and thus represents a more interesting task than CIFAR-10/100, or EMNIST classification (62-class). But it would indeed be interesting to see how $FedAda^2$ compares against datasets such as EMNIST as well. We have been able to implement these experiments on $FedAda^2$ and baselines, and we summarize their results in a table below. Convergence figures are added to the revision in Appendix section G.4 (highlighted in blue).
>
> | Metric          | FedAvg | FedAdaGrad | FedAdam | Direct Joint Adap. | Joint Adap. w/o Precond. Commu. | FedAda$^2$ |
> |------------------|--------|------------|---------|---------------------|---------------------------------|------------|
> | Test Accuracy    | 65.95% | 80.77%     | 81.79%  | 85.72%              | 85.01%                          | 85.11%     |
> | Train Accuracy   | 66.26%  | 81.25%      | 82.30%   | 86.43%               | 85.41%                           | 85.76%      |
>
> In summary, we observe that jointly adaptive approaches (Direct Joint Adap., Joint Adap. w/o Prec. Trans., and $FedAda^2$) outperform non-jointly adaptive methods by ~4% on the EMNIST dataset. Additionally, removing preconditioner transmission as well as compressing client-side gradient statistics for efficiency ($FedAda^2$) does not degrade the performance of direct joint adaptivity, while being more efficient. We refer to Appendix G.4 in Page 50 of the revision for more detailed discussions.
>
> Thank you for your feedback.

---

### Author Response · Authors · 2024-11-20

We appreciate the time all reviewers have dedicated to evaluating our work, and we would like to briefly provide a summary of our contributions, as well as the structure of the paper.

**[Contributions Summary]** Our goal is to show benefits and convergence of efficient joint adaptive optimization in FL, where both the server-side and client-side optimizers use adaptive optimizers. In Section 1, we explain the bottleneck behind deploying joint adaptive optimization, namely, resource-limited client-side memory increase and client-server communication complexity. In Section 3, we further motivate our problem by studying the necessities of client-side optimization. Afterwards, we propose our solution in Section 4, the $FedAda^2$ framework, to efficiently deploy joint adaptivity while cutting the communication and on-device memory costs. One key contribution of our work is the severing of preconditioners transmission between clients and the server for *joint* adaptive optimization, and proving that such a framework converges under a broad class of adaptive methods, which has not been shown before. In Section 6, we demonstrate empirically that $FedAda^2$ is more efficient than naive joint adaptive methods, and achieves better accuracy than non-joint adaptive baselines. All proofs are given in full in the Appendix.

**[On Appendix]** It seems the Appendix was missed by some reviewers as some of the questions are directly answered there. In our original submission, **the Appendix *was included* in the supplementary material** alongside complete code for all experiments as a single zip file per ICLR guidelines. We have updated the main text without manually splitting the Appendix for the reviewers’ convenience during the discussion phase.

Thank you for your time and feedback.

---

### Meta-Review · Area_Chair_35Y2 · 2024-12-22

**Metareview:**

The paper presents results on adaptive optimization in the FL context, including the value of adaptive methods at the clients and efficient communication, say by not communicating the pre-conditioner, and also ideas on reducing on device memory. There is general agreement among the reviewers that the work is interesting and could potentially advance aspects of FL. There are however some concerns regarding the work, including convincingly demonstrating the benefits of proposed ideas, such as avoiding preconditiners vs full or layer wise preconditioning, limited novelty of memory saving ideas, more extensive empirical evidence, and alternative baselines.  To their credit, the authors have responded to all questions in detail and addressed some of the concerns, but certain concerns remain even after the discussion phase.

**Additional Comments On Reviewer Discussion:**

The reviewers engaged with the authors and there were discussions which brought clarity to aspects of the work.

---

### Decision · Program_Chairs · 2025-01-22

Reject